# Tracking Drift: Variation-Aware Entropy Scheduling for Non-Stationary Reinforcement Learning

**Tongxi Wang** [1]  **Zhuoyang Xia** [1]  **Xinran Chen** [1]  **Shan Liu** [2]

## Abstract

Real-world reinforcement learning often faces environment drift, but most existing methods rely on static entropy coefficients/target entropy, causing over-exploration during stable periods and under-exploration after drift, and leaving unanswered the principled question of how exploration intensity should scale with drift magnitude. We show that, under standard assumptions, entropy scheduling in non-stationary maximum-entropy RL can be cast as the dynamic-regret trade-off between tracking a drifting comparator and stabilizing updates, yielding a square-root scaling rule for the entropy weight in terms of a online non-stationarity proxy. Building on this, we propose AES–Adaptive Entropy Scheduling–which adaptively adjusts the entropy coefficient/temperature online using observable drift proxies during training, requiring almost no structural changes and incurring minimal overhead. Across 4 algorithm variants, 12 tasks, and 4 drift modes, AES significantly reduces the fraction of performance degradation caused by drift and accelerates recovery after abrupt changes.

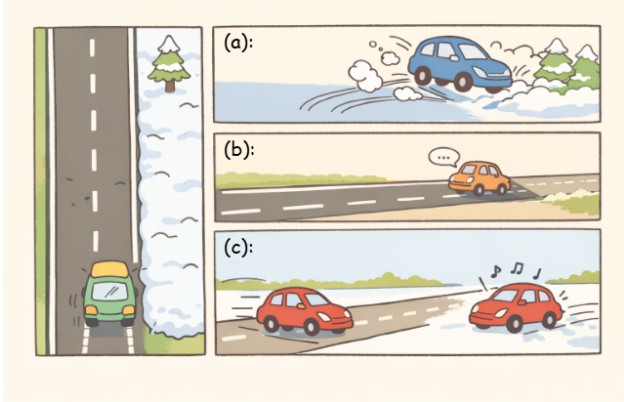

*Figure 1.* **Adaptive exploration for non-stationary RL.** Left: environment dynamics/objectives drift over time. Right: (a) too little exploration slows recovery after abrupt changes; (b) too much exploration harms performance in stable phases; (c) drift-aware scheduling adapts exploration to recover quickly while avoiding unnecessary randomness.

## 1. Introduction

Reinforcement learning (RL) has achieved remarkable success across a wide range of domains (Sutton & Barto, 1998; Mnih et al., 2015). However, most existing theory and many practical algorithms are developed under a **stationarity assumption**, namely that the reward function and environment dynamics remain fixed over time. In real-world applications, this assumption is often violated:

robots encounter changing physical conditions, autonomous driving systems must cope with evolving traffic patterns, and recommender systems are required to continuously adapt to shifting user preferences. In such scenarios, an agent effectively interacts not with a single stationary Markov decision process (MDP), but with a **sequence of time-varying MDPs** (Garcia & Rachelson, 2013; Besbes et al., 2014).

The central challenge of non-stationary reinforcement learning lies not only in learning an optimal policy, but also in **continuously adapting** to a changing environment while maintaining a balance between exploitation and exploration. Modern RL,such as maximum-entropy reinforcement learning methods make this balance explicit through entropy regularization, where the entropy coefficient (or temperature parameter) serves as the primary control knob for exploration. In practice, however, this parameter is typically treated as stationary: it is either fixed or adjusted to match a target entropy designed for stationary objectives, as in standard Soft Actor-Critic (SAC) style algorithms (Haarnoja et al., 2018). This practice has gradually fostered an implicit assumption that entropy control is largely

[1]School of Future Technology, Southeast University, Nanjing, China [2]School of Automation, Southeast University, Nanjing, China. Correspondence to: Shan Liu <liushan22@seu.edu.cn>.

*Proceedings of the 43rd International Conference on Machine Learning*, Seoul, South Korea. PMLR 306, 2026. Copyright 2026 by the author(s).

orthogonal to non-stationarity. Yet this assumption breaks down precisely when the environment changes: excessive exploration wastes samples during stable periods, while insufficient exploration significantly slows policy recovery after changes occur.

This work focuses on a seemingly simple yet crucial question: **how should exploration intensity scale with the magnitude of environmental change?** Existing research on non-stationary reinforcement learning partially alleviates adaptation challenges, but does not directly answer this question. Change-point detection methods can provide theoretical guarantees (Mellor & Shapiro, 2013), but often introduce additional computational and design complexity that makes them difficult to integrate into real-time continuous-control systems. Sliding-window strategies rely on manually chosen window sizes and lack principled guidance in high-dimensional reinforcement learning pipelines (Besbes et al., 2014). Meta-learning approaches can accelerate adaptation across tasks (Bing et al., 2023), yet they do not explicitly characterize the relationship between the **rate of environmental variation** and the **optimal entropy scheduling strategy**. Meanwhile, theoretical analyses of entropy regularization have largely focused on stationary settings, providing limited guidance on whether—and how—exploration strength should increase as non-stationarity intensifies. As a result, entropy coefficients in practice are still commonly tuned heuristically, with performance that is highly environment-dependent.

Our key observation is that the necessity of entropy scheduling is **conceptually independent** of the specific algorithmic details of reinforcement learning. Once non-stationarity induces drift in the optimal comparator, the learner faces a fundamental per-round trade-off: (i) adapting sufficiently fast to track the drifting optimum, versus (ii) avoiding unnecessary randomness when the environment is stable. This perspective suggests that entropy control can be **framed as a one-dimensional trade-off** that is sensitive to non-stationarity.

Based on this principle, we propose **Adaptive Entropy Scheduling (AES)**, a unified theoretical framework for non-stationary environments that adjusts the entropy coefficient according to observable signals. Our analysis yields a scaling implication under standard regularity assumptions: dynamic performance loss is governed by the interaction between the time horizon and the cumulative variation budget (Besbes et al., 2014; Cheung et al., 2020). More importantly, when applying this framework to deep reinforcement learning with function approximation and off-policy learning, we are able to explicitly distinguish the **dominant non-stationarity term** from **additional interface error terms**, thereby clarifying which factors must be controlled in practical training pipelines.

We instantiate AES as a plug-in entropy-scheduling mechanism for maximum-entropy reinforcement learning. Across all carriers, the underlying learning algorithm is kept unchanged, and we only replace the fixed entropy weight (temperature $\alpha$ or entropy-bonus coefficient) with the scheduled value produced by AES, without modifying the core architecture of different methods. Entropy scheduling is driven by online signals already available during training. We evaluate AES on three task families: Toy 2D multi-goal environment, MuJoCo continuous-control benchmarks, and large-scale Isaac Gym tasks under 4 drift modes, including abrupt, linear, periodic, and mixed changes induced by controlled goal, dynamics, or physics drifts. Across carriers and task suites, AES consistently mitigates performance degradation caused by non-stationarity and accelerates recovery after environmental changes compared to standard entropy-control baselines.

Overall, this work identifies entropy strength as a tractable control variable for non-stationary RL and shows how to schedule it in a variation-aware manner. We hope this perspective will move entropy scheduling beyond heuristic tuning toward variation-aware system design, and that it will integrate naturally with broader adaptive mechanisms in future research.

## 2. Related Work

### 2.1. Variation-Aware Non-Stationary RL

Non-stationary reinforcement learning is commonly formalized through variation budgets and dynamic regret, where rewards or transition dynamics evolve over time (Besbes et al., 2014). Recent work develops algorithms with guarantees that explicitly scale with the cumulative amount of non-stationarity. Model-free approaches show how policy updates should react to bounded variation in rewards and transitions (Mao et al., 2025; Peng & Papadimitriou, 2024), while complementary analyses extend these results to general function approximation and restart-based schemes (Feng et al., 2024). Empirical studies further confirm that fixed adaptation or exploration strategies are systematically suboptimal in drifting environments (Dal Toe et al., 2024; Hamadanian et al., 2025; Duan et al., 2025; Salaorni et al., 2025).

Together, these works establish a clear principle: optimal learning dynamics must explicitly depend on the rate of environmental change. However, existing variation-aware analyses largely remain decoupled from modern maximum-entropy deep RL. In particular, they do not address how the entropy regularization coefficient—the primary exploration control knob in maximum-entropy methods—should be scheduled as a function of online non-stationarity. Our work directly targets this gap

by casting entropy control under non-stationarity into a one-dimensional, variation-sensitive trade-off derived from dynamic online optimization, and by instantiating this principle in entropy-regularized deep RL with explicit separation between the dominant non-stationarity term and additional interface errors.

Recent offline-RL studies further show that adaptation can fail not only because the environment drifts, but also because the learning interface itself becomes unstable. Qiao et al. (2026a) identify harmful TD cross-covariance induced by standard squared-error TD objectives and mitigate it through clustered replay and corrective gradient penalties. Qiao et al. (2026b) analyze offline TD learning as a feedback system and show how optimizer dynamics can trigger or suppress critic collapse. These mechanisms are orthogonal to AES: they regularize the critic, replay, or optimizer dynamics, whereas AES regulates the global entropy level according to online non-stationarity. A natural combination is therefore to use such stability controls to reduce interface errors while AES adapts the exploration to environmental drift.

### 2.2. Exploration Control: Entropy Scheduling vs. State-Dependent Adaptation

Entropy-regularized RL, most notably Soft Actor-Critic (SAC) (Haarnoja et al., 2018), introduces an explicit entropy coefficient to balance exploration and exploitation. Classical analyses of entropy regularization focus primarily on stationary settings, characterizing how entropy decay or smoothing affects convergence and policy diversity, often yielding time-based schedules such as $\mathcal{O}(t^{-1/2})$ (Szpruch et al., 2022; Jhaveri et al., 2025).

More recent work explores adaptive entropy mechanisms in complex domains, including distributed multi-agent RL (Hu et al., 2024) and large language model RL (Zhang et al., 2025). These approaches adjust entropy coefficients based on optimization dynamics or task difficulty, but they remain largely heuristic: the entropy parameter is tuned as an internal algorithmic quantity, without a principled link to the magnitude of environmental non-stationarity.

A partially orthogonal line of work addresses adaptation through *state- or transition-dependent* intrinsic rewards. Curiosity- and novelty-based methods such as ICM and RND (Pathak et al., 2017; Burda et al., 2019) encourage exploration of poorly predicted or previously unseen states, while reactive exploration methods for lifelong or non-stationary RL explicitly increase intrinsic reward when the environment becomes harder to model locally (Steinparz et al., 2022; Xie et al., 2021). These methods primarily change *which* states or transitions are preferred during exploration.

AES addresses a different control question: not which

states are locally novel, but how random the *overall policy distribution* should be when the environment drifts. In a non-stationary MDP, reward or dynamics drift can substantially change the optimal policy without necessarily producing highly novel states, while novelty can also rise in ways that do not imply a comparator drift of comparable magnitude. For this reason, intrinsic-reward methods and AES should be viewed as complementary rather than mutually exclusive: the former modulate state preference, whereas the latter modulates global exploration intensity.

An alternative line of research addresses non-stationarity through explicit change-point detection and model or policy switching. Statistical testing and Bayesian schemes detect shifts in rewards, transitions, or value functions and trigger restarts or model reassignment (Alami et al., 2023; Chartouny et al., 2025; Li et al., 2025). While such methods can precisely localize abrupt changes and achieve strong regret guarantees, they are often computationally heavy and treat exploration as secondary—typically fixed or adjusted implicitly via restarts—rather than continuously modulated in proportion to detected drift.

Meta-reinforcement learning and continual RL provide yet another adaptation paradigm, training agents to infer latent task structure or rapidly adapt across evolving environments (Bing et al., 2023; Qi et al., 2025; Khetarpal et al., 2022). Although effective at improving adaptability, these methods usually rely on fixed entropy regularization or ad-hoc noise injection, leaving exploration strength loosely coupled to the actual rate of environmental change.

In contrast to these approaches, our method provides a principled, lightweight mechanism for exploration control under non-stationarity. The entropy coefficient is explicitly driven by an online non-stationarity proxy through a transparent per-round trade-off of the form $C_1 \xi_t / \lambda_t + C_2 \lambda_t$, yielding a square-root-type scaling rule. This makes exploration strength increase when sustained drift is detected and decrease during stable phases, embedding adaptivity directly into the core learning rule rather than relying on heuristics, restarts, or learned adaptation alone.

## 3. Theory

This section isolates the part of the argument that determines how exploration should scale with non-stationarity. The main point is simple: the entropy strength $\lambda_t$ acts as a single control variable. When the environment changes rapidly, a larger $\lambda_t$ helps the learner avoid over-committing to an outdated solution and re-track the new optimum; when the environment is stable, a smaller $\lambda_t$ avoids unnecessary randomness and preserves performance.

We first derive the scheduling principle in a surrogate OCO layer, where the trade-off between tracking drift and

preserving stability is easiest to isolate. We then transfer the same structure to non-stationary soft RL, where the comparator becomes the drifting soft-optimal policy. In this view, $u_t$ denotes a time-varying target solution; in the RL bridge below, it becomes the soft-optimal comparator.

### 3.1. Main Theoretical Claims

We begin by separating the three roles of the main results. Theorem 3.2 gives the structural claim: in the OCO surrogate, dynamic regret reduces to the per-round trade-off $C_1 \frac{\xi_t}{\lambda_t} + C_2 \lambda_t$, which identifies $\lambda_t$ as the single knob balancing tracking and stability. Theorem 3.5 gives the online claim: replacing the unobservable drift $\xi_t$ by a conservative observable proxy $\widehat{\xi}_t$ yields a fully online AES schedule with regret controlled by the cumulative proxy budget. Theorem 3.6 gives the RL-bridge claim: in non-stationary maximum-entropy RL, MDP variation induces drift of the soft-optimal comparator, so the same schedule yields a planning-version principal-term dynamic-regret bound, with approximation, sampling, and occupancy effects kept as explicit residuals.

Together, Theorems 3.2, 3.5 and 3.6 define the progression of this section: from a clean surrogate trade-off, to an implementable online rule, to its soft-RL interpretation. Technical details are deferred to Appendices A.3–A.5, with learning-side residual terms made explicit in Appendices A.6–A.9.

### 3.2. Dynamic mirror descent yields a per-round $\lambda$-trade-off

We start from non-stationary online convex optimization (OCO) on the simplex $\Pi = \Delta_K = \{x \in \mathbb{R}_+^K : \sum_{i=1}^K x_i = 1\}$. At round $t$, the learner plays $x_t \in \Pi$ and suffers a convex differentiable loss $f_t$. A *dynamic comparator* is a sequence $\{u_t\}_{t=1}^T \subset \Pi$ with per-round comparator drift $\xi_t := \|u_t - u_{t-1}\|_1$ (set $u_0 = u_1$), and dynamic regret

$$\operatorname{Reg}_T^{\mathrm{dyn}}(u_{1:T}) := \sum_{t=1}^T \big(f_t(x_t) - f_t(u_t)\big). \tag{1}$$

**Entropy mirror descent.** Let $\Psi(x) := \sum_{i=1}^K x_i \log x_i$ be negative entropy. Define

$$D_\Psi(x, y) := \Psi(x) - \Psi(y) - \langle \nabla \Psi(y), x - y \rangle, \tag{2}$$

and the mirror-descent update

$$x_{t+1} = \arg\min_{x \in \Pi} \Big\{ \eta_t \langle g_t, x \rangle + D_\Psi(x, x_t) \Big\}. \tag{3}$$

We use the $\ell_1/\ell_\infty$ pairing: $\|\cdot\| = \|\cdot\|_1$ and $\|\cdot\|_* = \|\cdot\|_\infty$.

**Lemma 3.1** (Dynamic MD inequality). *Let $\{u_t\}_{t=1}^T \subset \Pi$ be any comparator sequence and set $u_0 = u_1$. Assume*

*bounded mirror gradients: there exists $G_\Psi < \infty$ such that*

$$\|\nabla \Psi(z)\|_* \le G_\Psi, \qquad \forall z \in \Pi, \tag{4}$$

*and assume $\eta_t$ is nondecreasing. Then the iterates generated by* (3) *satisfy*

$$\sum_{t=1}^T \langle g_t, x_t - u_t \rangle \le \frac{D_\Psi(u_1, x_1)}{\eta_1} + \sum_{t=1}^T \frac{\eta_t}{2} \|g_t\|_*^2 + \sum_{t=2}^T \frac{2G_\Psi}{\eta_t} \|u_t - u_{t-1}\|_1. \tag{5}$$

Appendix A.14 provides a complete proof. This lemma is the main "non-stationary switch": it isolates drift through $\sum_t \|u_t - u_{t-1}\|_1 / \eta_t$.

**Entropy regularization and a master dynamic-regret inequality.** Assume bounded gradients for the base losses:

$$\|\nabla f_t(x)\|_\infty \le G, \qquad \forall x \in \Pi. \tag{6}$$

AES uses the time-varying regularized losses

$$\ell_t(x) := f_t(x) + \lambda_t \Psi(x), \tag{7}$$

and runs (3) with $g_t = \nabla \ell_t(x_t)$. Convexity gives

$$\ell_t(x_t) - \ell_t(u_t) \le \langle \nabla \ell_t(x_t), x_t - u_t \rangle. \tag{8}$$

Combining (8) with Lemma 3.1 yields

$$\sum_{t=1}^T \big(\ell_t(x_t) - \ell_t(u_t)\big) \le \frac{D_\Psi(u_1, x_1)}{\eta_1} + \sum_{t=1}^T \frac{\eta_t}{2} \|\nabla \ell_t(x_t)\|_*^2 + \sum_{t=2}^T \frac{2G_\Psi}{\eta_t} \|u_t - u_{t-1}\|_1. \tag{9}$$

To remove regularization, use

$$f_t(x_t) - f_t(u_t) = \big(\ell_t(x_t) - \ell_t(u_t)\big) - \lambda_t \big(\Psi(x_t) - \Psi(u_t)\big), \tag{10}$$

and the entropy bound $|\Psi(x)| \le \log K$ (Appendix A.15), which implies

$$-\lambda_t \big(\Psi(x_t) - \Psi(u_t)\big) \le 2\lambda_t \log K. \tag{11}$$

Altogether,

$$\operatorname{Reg^{dyn}}_T(u_{1:T}) \le \frac{D_\Psi(u_1, x_1)}{\eta_1} + \sum_{t=1}^T \frac{\eta_t}{2} \|\nabla \ell_t(x_t)\|_*^2 + \sum_{t=2}^T \frac{2G_\Psi}{\eta_t} \|u_t - u_{t-1}\|_1 + 2\log K \sum_{t=1}^T \lambda_t. \tag{12}$$

**Coupling $\eta_t$ and $\lambda_t$ produces a per-round trade-off.** AES couples the learning rate and entropy strength:

$$\eta_t = c\,\lambda_t, \qquad c > 0, \tag{13}$$

with $\lambda_t \in [\lambda_{\min}, \lambda_{\max}]$ and $\eta_t$ nondecreasing. Moreover, since $\nabla \ell_t = \nabla f_t + \lambda_t \nabla \Psi$,

$$\|\nabla \ell_t(x_t)\|_\infty \leq G + \lambda_t G_\Psi. \qquad (14)$$

For negative entropy, $G_\Psi$ is bounded on a truncated simplex $\Delta_{K,\varepsilon}$ (Appendix A.15):

$$G_\Psi = \sup_{x \in \Delta_{K,\varepsilon}} \|\nabla \Psi(x)\|_\infty \leq 1 + |\log \varepsilon|. \qquad (15)$$

Substituting (13) and (14) into (12) and absorbing higher-order dependence through $\lambda_t \leq \lambda_{\max}$ (Appendix A.16) yields:

**Theorem 3.2** ($\lambda$-trade-off dynamic regret bound). *Run mirror descent* (3) *on* $\ell_t$ *with* (13). *Then there exist constants* $C_0, C_1, C_2 > 0$ *independent of* $T$ *such that for any comparator sequence* $u_{1:T} \subset \Pi$,

$$\text{Reg}_T^{\text{dyn}}(u_{1:T}) \leq C_0 + \sum_{t=2}^T \left( C_1 \frac{\xi_t}{\lambda_t} + C_2 \lambda_t \right),$$

$$\xi_t := \|u_t - u_{t-1}\|_1. \qquad (16)$$

*Equivalently, each round contributes*

$$\varphi_t(\lambda) := C_1 \frac{\xi_t}{\lambda} + C_2 \lambda, \qquad \lambda > 0. \qquad (17)$$

Equation (16) exposes the operative per-round trade-off in entropy scheduling. The term $C_1 \xi_t / \lambda_t$ is the tracking cost: after larger comparator drift, choosing $\lambda_t$ too small slows re-tracking. The term $C_2 \lambda_t$ is the stability cost: choosing $\lambda_t$ too large keeps the iterate unnecessarily diffuse even in stable phases. After coupling $\eta_t = c\lambda_t$, the remaining higher-order dependence is absorbed into constants, so (16) is the structural form that drives the schedule.

**Oracle scale.**

**Theorem 3.3** (Per-round oracle optimal $\lambda_t$). *For each* $t \geq 2$, *define*

$$\lambda_t^\star = \arg \min_{\lambda > 0} \varphi_t(\lambda) = \sqrt{\frac{C_1}{C_2} \xi_t}. \qquad (18)$$

*Then*

$$\text{Reg}_T^{\text{dyn}}(u_{1:T}) \leq C_0 + 2\sqrt{C_1 C_2} \sum_{t=2}^T \sqrt{\xi_t}$$

$$\leq C_0 + 2\sqrt{C_1 C_2} \sqrt{T \sum_{t=2}^T \xi_t}. \qquad (19)$$

Thus the oracle entropy scale is $\lambda_t^\star \asymp \sqrt{\xi_t}$ at the dominant-order level, with higher-order terms absorbed into constants.

### 3.3. A fully-online schedule from an observable drift proxy

The oracle $\lambda_t^\star$ depends on $\xi_t$, which is usually unobservable. AES instead assumes an observable proxy sequence $\widehat{\xi}_t$ satisfying

$$\xi_t \leq \widehat{\xi}_t, \qquad (20)$$

*Remark* 3.4. Condition (20) does not require $\hat{\xi}_t$ to be an unbiased estimator of $\xi_t$; it only requires a conservative signal that increases with non-stationarity. Hence the schedule is compatible with noisy, clipped, and smoothed proxies.

Appendix A.7 provides an empirical calibration procedure that constructs a conservative $\widehat{\xi}_t$ from an observable learning signal (our default TD-error proxy), and makes (20) verifiable on injected-drift benchmarks with finite-sample coverage. We use its prefix sum

$$\widehat{A}_t := \sum_{s=1}^t \widehat{\xi}_s. \qquad (21)$$

The resulting online schedule (before clipping) is

$$\lambda_t = \sqrt{\frac{C_1}{C_2}} \sqrt{\frac{\widehat{A}_t}{t}}. \qquad (22)$$

**Theorem 3.5** (Online AES scheduling driven by an observable proxy). *Under the conditions of Theorem 3.2, and assuming* (20), *choosing* $\lambda_t$ *by* (22) *yields*

$$\text{Reg}_T^{\text{dyn}} \leq C_0 \log K + 4\sqrt{C_1 C_2 T \widehat{A}_T}. \qquad (23)$$

In practice we clip $\lambda_t$ to $[\lambda_{\min}, \lambda_{\max}]$:

$$\lambda_t = \text{clip}_{[\lambda_{\min}, \lambda_{\max}]} \left( \sqrt{\frac{C_1}{C_2}} \sqrt{\frac{\widehat{A}_t}{t}} \right), \qquad (24)$$

which introduces only controllable additive compensation terms (Appendix A.19).

### 3.4. Bridge to non-stationary maximum-entropy RL

We now show how MDP drift induces comparator drift, and why **critic drift** is a sensible proxy $\widehat{\xi}_t$ in experiments.

**Non-stationary soft MDPs.** Let $M_t = (\mathcal{S}, \mathcal{A}, P_t, r_t, \rho, \gamma)$ be a sequence of discounted MDPs with bounded rewards. For a policy $\pi$, define the maximum-entropy return $J_t(\pi) = \mathbb{E}[\sum_{h \geq 0} \gamma^h (r_t(s_h, a_h) + \mu H(\pi(\cdot \mid s_h)))]$ and let $\pi_t^\star \in \arg \max_\pi J_t(\pi)$.

**MDP variation controls $Q^\star$ drift.** Define the non-stationarity budget

$$B_T^{\text{MDP}} := \sum_{t=2}^T \left( \Delta_t^r + \gamma V_{\max} \Delta_t^P \right), \qquad (25)$$

where $\Delta_t^r := \sup_{s,a} |r_t(s,a) - r_{t-1}(s,a)|$ and $\Delta_t^P := \sup_{s,a} \|P_t(\cdot \mid s,a) - P_{t-1}(\cdot \mid s,a)\|_1$. Let $Q_t^\star$ be the soft-optimal $Q$-function. A standard fixed-point perturbation argument for the $\gamma$-contractive soft Bellman operator gives (Appendix A.20)

$$\|Q_t^\star - Q_{t-1}^\star\|_\infty \leq \frac{1}{1-\gamma}\Big(\Delta_t^r + \gamma V_{\max}\Delta_t^P\Big). \quad (26)$$

**$Q^\star$ drift controls $\pi^\star$ drift.** For each $s$, $\pi_t^\star(\cdot \mid s) = \mathrm{softmax}(Q_t^\star(s,\cdot)/\mu)$. Using a statewise $\ell_1-\ell_\infty$ Lipschitz bound for softmax (Appendix A.21),

$$\|\pi_t^\star(\cdot \mid s) - \pi_{t-1}^\star(\cdot \mid s)\|_1 \leq \frac{1}{\mu}\|Q_t^\star - Q_{t-1}^\star\|_\infty. \quad (27)$$

Combining (26)–(27) shows that the comparator drift in the OCO view is controlled by $B_T^{\mathrm{MDP}}$ up to $(\mu(1-\gamma))^{-1}$ factors.

**Why this matches the OCO surrogate.** For each state $s$, define the per-state linear-plus-entropy loss

$$f_{t,s}(\pi_s) := -\langle Q_t^\star(s,\cdot), \pi_s\rangle + \mu\,\Psi(\pi_s), \qquad \pi_s \in \Delta(\mathcal{A}). \quad (28)$$

The soft-optimal policy satisfies $\pi_t^\star(\cdot \mid s) = \arg\min_{\pi_s} f_{t,s}(\pi_s)$, and the gap has an explicit KL form:

$$f_{t,s}\big(\pi(\cdot \mid s)\big) - f_{t,s}\big(\pi_t^\star(\cdot \mid s)\big) = \mu\,\mathrm{KL}\big(\pi(\cdot \mid s)\,\|\,\pi_t^\star(\cdot \mid s)\big). \quad (29)$$

Thus, the principal term in soft-RL dynamic regret is an occupancy-weighted sum of per-state OCO regrets with comparator $u_{t,s} = \pi_t^\star(\cdot \mid s)$.

**Overall (planning-version) rate.**

**Theorem 3.6** (AES-RL-Plan: non-stationary soft-RL dynamic regret (principal term)). *In a non-stationary tabular soft-MDP sequence, run AES with the online schedule from Theorem 3.5 using a valid drift proxy. Then there exists a constant $C$ such that*

$$\mathrm{Reg}_T^{\mathrm{RL}} \leq \widetilde{O}\left(\frac{\sqrt{|\mathcal{S}|\log|\mathcal{A}|}}{\mu(1-\gamma)^2}\sqrt{B_T^{\mathrm{MDP}}\,T}\right) + \sum_{t=1}^T \mathrm{Bias}_t, \quad (30)$$

*where $\mathrm{Bias}_t$ collects planning-to-RL interface residuals (made explicit in the appendix).*

**From planning to learning.** Theorem 3.6 is a planning-version principal-term result: it isolates the dominant dependence on MDP variation in non-stationary soft RL. In deep off-policy RL, replacing $Q_t^\star$ and $d_t^{\pi_t^\star}$ by learned surrogates introduces additional approximation and occupancy terms, which are collected into $\mathrm{Bias}_t$ and made explicit in Appendices A.8 and A.9. We therefore view Theorem 3.6 as the structural RL bridge for AES, rather than as a complete finite-sample theorem for deep off-policy learning.

### 3.5. Experiment-Theory Interface

In theory, AES is driven by a conservative proxy for comparator drift. In practice, we instantiate this proxy by the upper quantile of TD errors, which tends to rise when environmental change makes previously accurate value predictions stale. It is also an inherent training parameter, obtainable without extra cost or operation. AES therefore uses the proxy as an online alarm for increasing or relaxing exploration, rather than as an exact estimator of comparator drift. Appendix A.7 gives the calibration procedure, and other calibrated signals (e.g., model disagreement or critic-parameter drift) would also fit the same interface.

Section 4 evaluates the structural prediction of the theory: exploration strength should increase under stronger non-stationarity and relax in stable phases. We therefore report behavioral metrics, rather than treating the experiments as a numerical verification of the regret bounds.

## 4. Experiments

### 4.1. Experiment Setup

**Algorithm carriers.** We evaluate AES on four entropy-regularized RL algorithms: SAC (Haarnoja et al., 2018), PPO (Schulman et al., 2017), SQL (Haarnoja et al., 2017), and MEow (Chao et al., 2024). All methods contain an explicit entropy regularization term (temperature $\alpha$ for SAC/SQL/MEow, entropy bonus $c_{\mathrm{ent}}$ for PPO). AES is integrated without modifying the underlying actor–critic or value-learning structure, covering both off-policy and on-policy learning as well as distinct MaxEnt formulations. For the SAC baseline, we applied automatic temperature control.

AES acts as a plug-in exploration controller that schedules the entropy weight online. At each update, a scalar drift proxy is extracted from critic TD errors (off-policy) or value TD deltas (PPO), smoothed via a window or EMA, and fed into the variation-aware scheduler. AES outputs a scheduled temperature $\alpha_t$ (SAC/SQL/MEow) or entropy coefficient $c_{\mathrm{ent},t}$ (PPO), replacing the fixed entropy weight. Our default proxy is the $q$-quantile of absolute TD errors,

$$\hat{v}_t = \mathrm{Quantile}_q(|\delta|), \qquad q = 0.9, \quad (31)$$

computed on the current update batch. The scheduled entropy weight is clipped to a fixed range for numerical stability. Carrier-specific injection points are summarized in Table 1.

**Tasks and non-stationarity.** We evaluate AES on three task families: **Toy** (2D multi-goal), **MuJoCo** (Todorov et al., 2012) (Hopper, HalfCheetah, Walker2d, Ant, Humanoid), and **Isaac Gym** (Makoviychuk et al., 2021) (Ant, Humanoid, Ingenuity, ANYmal, AllegroHand, FrankaCabinet). Each

*Table 1.* How AES is integrated into each algorithm carrier. We keep the base algorithms unchanged except for replacing the entropy weight with the scheduled value.

| Carrier | Entropy weight | AES output and injection | Default proxy |
|---|---|---|---|
| SAC | temperature $\alpha$ | output $\alpha_t$; replace $\alpha$ in the actor objective term $\alpha_t \log \pi(a|s)$ (and the corresponding soft target/value computation, if applicable) | $\text{Quantile}_{0.9}(|\delta^{Q_1}| \cup |\delta^{Q_2}|) +$ smoothing |
| PPO | entropy coefficient $c_{\text{ent}}$ | output $c_{\text{ent},t}$; replace $c_{\text{ent}}$ in the policy loss term $-c_{\text{ent},t} H(\pi(\cdot|s))$ | $\text{Quantile}_{0.9}(|\delta^V|)$ over rollout batch + smoothing |
| SQL | temperature $\alpha$ | output $\alpha_t$; replace $\alpha$ in soft backup/value computation (soft Bellman target) | $\text{Quantile}_{0.9}(|\delta^{\text{soft}Q}|) +$ smoothing |
| MEow | temperature $\alpha$ | output $\alpha_t$; replace $\alpha$ wherever it appears in the clipped double-$Q$ targets / soft value computation; synchronize online/target modules | $\text{Quantile}_{0.9}(|\delta^{Q_1}| \cup |\delta^{Q_2}|) +$ smoothing (EMA recommended) |

task is trained under five non-stationarity patterns: *Steady*, *Abrupt*, *Linear*, *Periodic*, and *Mixed*. Drifts are injected via goal changes (Toy), dynamics or target variations (MuJoCo), and scalable physics or task perturbations (Isaac Gym). Change points are aligned by *training progress* rather than wall-clock time: all methods encounter the same drift events at the same fractions of total environment interaction steps. This keeps the non-stationarity schedule comparable across methods even when their internal optimization dynamics differ.

**Evaluation protocol.** All results are averaged over multiple random seeds with appropriate dispersion. Methods are aligned by environment interaction steps and evaluated under identical drift schedules and termination criteria. AES incurs negligible overhead, requiring only a scalar statistic of TD errors and a lightweight scheduler update per iteration.

Unlike change-point detection or restart-based methods, AES performs continuous entropy modulation and does not explicitly detect or localize change points.

### 4.2. Evaluation Metrics

We report three complementary metrics that jointly capture overall sample efficiency, robustness to non-stationarity, and direct post-change adaptation speed.

**(1) Normalized Area Under Curve (nAUC).** For each method and task, we compute the area under the return–environment-steps curve

$$\text{AUC} = \int_0^T R(t)\, \mathrm{d}t, \tag{32}$$

where $R(t)$ is the expected return at environment step $t$ and $T$ is the total training steps. To reduce scale differences across tasks, we normalize AUC by the **Steady SAC**

baseline within the same task family $\mathcal{F}$:

$$\text{nAUC}(\tau) = \frac{\text{AUC}(\tau)}{\text{AUC}(\text{SAC, Steady, } \mathcal{F})}. \tag{33}$$

**(2) Performance drop area ratio.** To quantify the proportion of performance loss induced by non-stationarity, we compare the AUC under a non-stationary pattern to the corresponding Steady AUC. Let $\text{AUC}^{\text{ns}}$ be the AUC under a non-stationary pattern and $\text{AUC}^{\text{steady}}$ the AUC under Steady. We define

$$\text{DropRatio} = 1 - \frac{\text{AUC}^{\text{ns}}}{\text{AUC}^{\text{steady}}}. \tag{34}$$

Smaller values indicate less cumulative damage from drift. Note that DropRatio can be negative when $\text{AUC}^{\text{ns}} > \text{AUC}^{\text{steady}}$, meaning the non-stationary run outperformed the steady reference over the horizon.

**(3) Abrupt-change recovery time.** Under abrupt pattern, we measure how quickly a method recovers after a change point. Specifically, recovery time is defined as the number of environment steps from the change occurrence to the first time the performance returns to the pre-change level (using the same evaluation protocol as the learning curves). We report recovery time *normalized* by the total training steps, so the metric is comparable across tasks. Let $\{t_{\text{change}}^{(i)}\}_{i=1}^K$ denote the set of $K$ change points during training, and let $t_{\text{recover}}^{(i)}$ be the corresponding recovery time after the $i$-th change. We define the normalized recovery time as

$$\text{Rec}(\tau) = \frac{1}{T} \sum_{i=1}^K \left( t_{\text{recover}}^{(i)} - t_{\text{change}}^{(i)} \right), \tag{35}$$

Lower values indicate stronger adaptability to abrupt non-stationarity.

*Table 2.* **Overall performance under different non-stationarity patterns.** We report normalized AUC (higher is better) and performance drop area ratio (lower is better) for each method across five change patterns (Steady, Abrupt, Linear, Periodic, Mixed), aggregated within each task family. Normalization is performed relative to the Steady SAC baseline in the same family.

| | nAUC ↑ | | | | | Performance drop area ratio ↓ | | | | |
|---|---|---|---|---|---|---|---|---|---|---|
| Toy | Steady | Abrupt | Linear | Periodic | Mixed | Steady | Abrupt | Linear | Periodic | Mixed |
| SAC | 1.00 | 0.72 | 0.80 | 0.81 | 0.73 | 0.00 | 0.28 | 0.20 | 0.19 | 0.27 |
| PPO | 0.89 | 0.75 | 0.79 | 0.71 | 0.67 | 0.00 | 0.16 | 0.11 | 0.20 | 0.25 |
| SQL | 0.90 | 0.75 | 0.82 | 0.77 | 0.68 | 0.00 | 0.17 | 0.09 | 0.14 | 0.24 |
| MEow | 0.90 | 0.79 | 0.87 | 0.86 | 0.77 | 0.00 | 0.12 | 0.03 | 0.04 | 0.14 |
| SAC+AES | **1.13** | 0.88 | 0.94 | 0.94 | **0.97** | 0.00 | 0.22 | 0.17 | 0.17 | 0.14 |
| PPO+AES | 0.89 | 0.92 | 0.92 | 0.89 | 0.79 | 0.00 | -0.03 | -0.03 | 0.00 | 0.11 |
| SQL+AES | 0.98 | 0.90 | 0.94 | 0.93 | 0.91 | 0.00 | 0.08 | 0.04 | 0.05 | **0.07** |
| MEow+AES | 0.97 | **1.03** | **1.02** | **1.01** | 0.83 | 0.00 | **-0.06** | **-0.05** | **-0.04** | 0.14 |
| MuJoCo | Steady | Abrupt | Linear | Periodic | Mixed | Steady | Abrupt | Linear | Periodic | Mixed |
| SAC | 1.00 | 0.67 | 0.76 | 0.68 | 0.65 | 0.00 | 0.33 | 0.24 | 0.32 | 0.35 |
| PPO | 0.88 | 0.66 | 0.64 | 0.67 | 0.57 | 0.00 | 0.25 | 0.27 | 0.24 | 0.35 |
| SQL | 0.80 | 0.52 | 0.53 | 0.50 | 0.44 | 0.00 | 0.35 | 0.34 | 0.38 | 0.45 |
| MEow | 0.92 | 0.67 | 0.71 | 0.79 | 0.63 | 0.00 | 0.27 | 0.23 | 0.14 | 0.32 |
| SAC+AES | **1.24** | 0.87 | **0.94** | **0.94** | **0.94** | 0.00 | 0.30 | 0.24 | 0.24 | 0.24 |
| PPO+AES | 0.94 | 0.69 | 0.79 | 0.82 | 0.77 | 0.00 | 0.27 | 0.16 | 0.13 | 0.18 |
| SQL+AES | 0.81 | 0.65 | 0.79 | 0.71 | 0.64 | 0.00 | 0.20 | **0.02** | 0.12 | 0.21 |
| MEow+AES | 0.98 | **0.95** | 0.88 | 0.87 | 0.89 | 0.00 | **0.03** | 0.10 | **0.11** | **0.09** |
| Isaac Gym | Steady | Abrupt | Linear | Periodic | Mixed | Steady | Abrupt | Linear | Periodic | Mixed |
| SAC | 1.00 | 0.58 | 0.69 | 0.57 | 0.51 | 0.00 | 0.42 | 0.31 | 0.43 | 0.49 |
| PPO | 0.82 | 0.44 | 0.59 | 0.55 | 0.48 | 0.00 | 0.46 | 0.28 | 0.33 | 0.41 |
| SQL | 0.76 | 0.39 | 0.47 | 0.49 | 0.38 | 0.00 | 0.49 | 0.38 | 0.36 | 0.50 |
| MEow | 0.99 | 0.68 | 0.76 | 0.76 | 0.54 | 0.00 | 0.31 | 0.23 | 0.23 | 0.45 |
| SAC+AES | **1.13** | 0.73 | **0.93** | 0.95 | 0.79 | 0.00 | 0.35 | 0.18 | 0.16 | 0.30 |
| PPO+AES | 0.87 | 0.62 | 0.67 | 0.70 | 0.57 | 0.00 | 0.29 | 0.23 | 0.20 | 0.34 |
| SQL+AES | 0.81 | 0.66 | 0.66 | 0.72 | 0.60 | 0.00 | **0.19** | 0.19 | 0.11 | 0.26 |
| MEow+AES | 1.09 | **0.81** | 0.90 | **0.99** | **0.92** | 0.00 | 0.26 | **0.17** | **0.09** | **0.16** |

## 4.3. Overall Performance Across Non-stationarity Patterns

We first evaluate overall performance under five patterns (Steady, Abrupt, Linear, Periodic, Mixed) across three task families (Toy, MuJoCo, Isaac Gym) and four algorithmic carriers (SAC, PPO, SQL, MEow). Table 2 summarizes **normalized AUC** and **drop-area ratio**.

**Overall trends.** Table 2 reveals a consistent benefit of AES across task families and carriers. Under **non-stationary patterns** (Abrupt/Linear/Periodic/Mixed), AES generally *increases* normalized AUC while *reducing* the drop-area ratio, indicating improved sample efficiency and stronger robustness to changes. Under the **Steady** pattern, we do not observe systematic degradation; in most settings the normalized AUC is comparable or slightly improved. This

suggests that AES is not a one-off patch specialized for change events, but rather a broadly applicable mechanism for *adaptive calibration of exploration strength*.

## 4.4. Abrupt-change Recovery Time

We next focus on adaptation speed under abrupt non-stationarity. Table 3 reports the **normalized abrupt-change recovery time** (percentage of total training steps) across all tasks and carriers.

**Recovery is consistently faster with AES.** Across all tasks and carriers, AES reduces normalized recovery time. Averaged over the 12 tasks, recovery time decreases from $\approx 13.96\%$ to $7.74\%$ for SAC, from $\approx 12.12\%$ to $8.43\%$ for PPO, from $\approx 15.73\%$ to $9.73\%$ for SQL, and from $\approx 11.58\%$ to $6.42\%$ for MEow. Gains are especially

*Table 3.* **Adaptation speed after abrupt environmental changes.** Abrupt-change recovery time is reported as the fraction of total training steps required for performance to return to the pre-change level after each change point. Lower values indicate better adaptability.

| | | Abrupt-change recovery time | | | | | | | |
|---|---|---|---|---|---|---|---|---|---|
| Family | Task | SAC | PPO | SQL | MEow | SAC +AES | PPO +AES | SQL +AES | MEow +AES |
| Toy | 2d multi-goal | 7.8% | 7.6% | 9.4% | 6.1% | 3.7% | 3.7% | 4.6% | 3.2% |
| MuJoCo | Hopper | 12.2% | 9.4% | 12.7% | 10.9% | 6.4% | 6.1% | 6.6% | 4.7% |
| | HalfCheetah | 9.6% | 8.0% | 11.8% | 7.7% | 5.1% | 5.1% | 5.8% | 4.4% |
| | Walker2d | 10.9% | 9.0% | 12.2% | 9.0% | 5.6% | 5.5% | 6.6% | 4.7% |
| | Ant | 11.9% | 10.9% | 16.4% | 11.7% | 6.7% | 6.9% | 8.3% | 5.9% |
| | Humanoid | 14.8% | 13.0% | 16.3% | 15.1% | 8.6% | 10.3% | 10.6% | 7.5% |
| Isaac Gym | Ant | 14.1% | 11.2% | 17.5% | 11.6% | 8.7% | 9.7% | 10.2% | 7.0% |
| | Humanoid | 16.5% | 14.1% | 18.2% | 12.5% | 9.8% | 10.5% | 13.9% | 7.8% |
| | Ingenuity | 15.3% | 14.1% | 16.7% | 12.7% | 8.2% | 11.0% | 11.6% | 7.0% |
| | ANYmal | 13.5% | 13.8% | 17.7% | 12.5% | 9.0% | 8.7% | 10.2% | 6.5% |
| | AllegroHand | 21.5% | 18.4% | 21.3% | 14.6% | 10.6% | 12.1% | 14.6% | 9.8% |
| | FrankaCabinet | 19.4% | 15.9% | 18.5% | 14.6% | 10.5% | 11.6% | 13.7% | 8.5% |

large on high-dimensional tasks (e.g., ALLEGROHAND and FRANKACABINET), where distribution shifts are more disruptive, consistent with the role of increased exploration after change points.

### 4.5. Summary

Combining Table 2 (normalized AUC and drop-area ratio) with Table 3 (abrupt-change recovery time), we observe three key characteristics of AES: **(i) Cross-algorithm consistency:** all four carriers benefit, suggesting that AES functions as a *plug-in exploration-strength control layer*. **(ii) Stronger gains under more severe or composite changes:** improvements are most pronounced under Abrupt and Mixed patterns, where changes are sharper and harder to track, as reflected by larger reductions in drop-area ratio and recovery time. **(iii) Good steady-state behavior:** under Steady training, we do not see systematic regressions, alleviating the common concern that better adaptation to non-stationarity must come at the cost of stable-phase performance. Ablation study is shown in Appendix B.

## 5. Conclusion

This work frames entropy scheduling in non-stationary reinforcement learning as a one-dimensional trade-off and introduces AES, an online method that adapts the entropy coefficient based on observable drift proxies. Empirically, AES improves robustness to non-stationarity and speeds up recovery after abrupt changes across a range of algorithms and tasks. Future directions include improving drift proxy design, analyzing interactions with other adaptation mechanisms, and extending the framework to broader control settings.

## Acknowledgement

This study is supported by the National Natural Science Foundation of China [Grant 72401063], Natural Science Foundation of Jiangsu Province [BK20241285], Zhishan Young Scholarship of Southeast University, and the Young Elite Scientist Sponsorship Program By CAST [YESS20240177].

## Impact Statement

This paper presents work whose goal is to advance the field of Machine Learning. There are many potential societal consequences of our work, none which we feel must be specifically highlighted here.

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

# A. Theory Details and Proofs for Section 3

## A.1. Guide to Appendix A

Appendix A contains the theoretical material behind Section 3. It is organized from the main argument to the supporting proofs. The first part of the appendix develops the full OCO-to-RL chain used in Section 3; the later part records the technical ingredients, deferred proofs, and auxiliary inequalities used along that chain.

A first reading can proceed through Appendix A.2– A.9. These sections contain the main theory path: the unified problem setup, the OCO trade-off, the online schedule, the reduction to non-stationary soft RL, the proxy calibration step, and the planning-to-learning residual terms. Appendix A.10– A.22 can then be consulted locally when a proof, constant, or auxiliary lemma is needed.

More specifically, each subsection serves the following role.

Appendix A.2 fixes the common language for the rest of the theory. It introduces the non-stationary OCO problem, the entropy mirror-descent update, the scheduled entropy coefficient, and the way the same control variable will later be transferred to non-stationary maximum-entropy RL.

Appendix A.3 derives the core per-round trade-off

$$\frac{C_1 \xi_t}{\lambda_t} + C_2 \lambda_t. \tag{36}$$

It starts from dynamic mirror descent, removes the entropy regularization term, and turns the resulting bound into the one-dimensional form used in Theorem 3.2. It also gives the corresponding per-round oracle choice behind Theorem 3.3.

Appendix A.4 turns the oracle rule into a fully online schedule driven by an observable proxy. It derives the prefix-sum schedule, proves the corresponding regret bound, and states the clipped form used in implementation. This subsection is the direct support for Theorem 3.5.

Appendix A.5 transfers the same structure to non-stationary soft RL. It defines the statewise entropy-regularized surrogate, shows how MDP variation controls $Q_t^\star$ drift and then policy drift, and states the planning-version principal-term result in Theorem 3.6.

Appendix A.6 explains how the ideal planning objects are connected to practical RL quantities. It writes the learning-side residual terms explicitly and shows where critic substitution and occupancy mismatch enter the argument.

Appendix A.7 makes the observable-proxy condition operational. It relates comparator drift to an MDP-level drift surrogate, then calibrates an observable signal through a monotone upper envelope with conformal adjustment, yielding a conservative proxy suitable for the online schedule.

Appendix A.8 gives the exact decomposition of the planning-to-learning gap. It introduces the discounted occupancy notation, states the soft performance-difference identity, gives the per-state surrogate representation, and defines the bias and occupancy-mismatch terms formally.

Appendix A.9 turns the decomposition from Appendix A.8 into a usable residual bound. It isolates a directly controllable bias term, bounds the occupancy mismatch, and records an on-policy strengthening that can be invoked when needed.

Appendix A.10 marks the transition from the main theory chain to the technical support material. It explains that the remaining subsections are collected so that the core argument in Appendix A.2–A.9 can be read without interruption.

Appendix A.11 records the basic facts about the simplex, the $\ell_1/\ell_\infty$ pairing, negative entropy, Bregman identities, and the truncated simplex. These facts are used repeatedly in the mirror-descent analysis and in the constant tracking steps.

Appendix A.12 rewrites the AES–OCO update in equivalent forms. It shows the connection between the proximal mirror-descent form and the exponentiated-gradient form, and explains why coupling $\eta_t = c\lambda_t$ yields a single control knob.

Appendix A.13 proves a robust dynamic mirror-descent inequality in a general form. It gives the one-step bound, the exact telescoping decomposition for dynamic comparators, and the summed inequality that underlies Lemma 3.1.

Appendix A.14 specializes the previous subsection to give the full proof of Lemma 3.1. It is the place to read when one wants the complete dynamic-comparator handling used in Section 3.

Appendix A.15 collects the entropy bounds used throughout the appendix. It proves that $|\Psi(x)| \leq \log K$ on the simplex and that $|\nabla\Psi(x)|_\infty$ is controlled on the truncated simplex.

Appendix A.16 tracks the constants in Theorem 3.2. Starting from the intermediate regret inequality, it shows exactly how the drift term, the gradient term, and the entropy term are reorganized into the constants $C_0$, $C_1$, and $C_2$.

Appendix A.17 studies the best offline constant choice of $\lambda$. It derives the minimizer when $\lambda_t$ is held fixed, and relates that offline choice to the per-round oracle scale.

Appendix A.18 proves the two summation inequalities used in the online schedule analysis. These are the square-root potential bound and the bound on $\sum_t \sqrt{\widehat{A}_t/t}$ that appear in the proof of Theorem 3.5.

Appendix A.19 analyzes the effect of clipping the schedule to $[\lambda_{\min}, \lambda_{\max}]$. It shows that clipping changes the bound only through controllable additive compensation terms and explains why clipping is needed in practical RL implementations.

Appendix A.20 proves Eq. (26). It develops the soft Bellman operator, shows that log-sum-exp is 1-Lipschitz under $\|\cdot\|_\infty$, proves the contraction and operator-drift lemmas, and then derives the sensitivity bound for $Q_t^\star$.

Appendix A.21 proves Eq. (27). It gives the Jacobian calculation for softmax, establishes the $\ell_\infty \to \ell_1$ Lipschitz constant, and then transfers $Q^\star$ drift into policy drift in the statewise form used in Section 3.

Appendix A.22 explains how to pass from squared drift terms to the linear variation budget without losing the target rate. It provides an explicit constant $C_Q$ and shows why this step preserves the intended $\widetilde{\mathcal{O}}(\sqrt{B_T^{\mathrm{MDP}}T})$ scaling.

For quick navigation, a reader interested mainly in the main claims can read Appendix A.2–A.5 first, then Appendix A.7–A.9. A reader checking only the OCO derivation may focus on Appendix A.3 together with Appendix A.13–A.19. A reader checking only the RL bridge may focus on Appendix A.5 together with Appendix A.20–A.22.

### A.2. Preliminaries and Problem Setting: a Unified View of OCO and Non-stationary Maximum-Entropy RL

This section studies adaptive exploration–exploitation in **non-stationary** environments. A fixed entropy regularization weight (temperature) may systematically mis-balance adaptation speed and excessive randomness when the environment varies over time. The analysis develops an **Adaptive Entropy Scheduling** (AES) principle: the entropy strength is scheduled online according to the magnitude of non-stationarity, yielding a unified theory chain from Online Convex Optimization (OCO) to non-stationary maximum-entropy reinforcement learning (soft-RL).

#### A.2.1. NON-STATIONARY ONLINE CONVEX OPTIMIZATION (OCO)

The decision set is the $K$-dimensional simplex

$$\Pi = \Delta_K := \left\{x \in \mathbb{R}_+^K : \sum_{i=1}^K x_i = 1\right\}. \tag{37}$$

At round $t = 1, \ldots, T$, the environment reveals a convex differentiable loss $f_t : \Pi \to [0, 1]$. Assume bounded gradients: there exists $G > 0$ such that for all $x \in \Pi$,

$$\|\nabla f_t(x)\|_\infty \leq G. \tag{38}$$

A **dynamic comparator** is any sequence $\{u_t\}_{t=1}^T \subset \Pi$. Define the per-round path increment

$$\Delta_t := \|u_t - u_{t-1}\|_1 \quad (t \geq 2), \qquad \Delta_1 := 0, \tag{39}$$

and the total path length $P_T := \sum_{t=1}^T \Delta_t$. The dynamic regret is

$$\mathrm{Reg}_T^{\mathrm{dyn}} := \sum_{t=1}^T \big(f_t(x_t) - f_t(u_t)\big), \tag{40}$$

where $\{x_t\}$ is the algorithm's decision sequence.

A.2.2. ENTROPY MIRROR DESCENT AND THE ENTROPY COEFFICIENT AS A CONTROL KNOB

The mirror map is the negative entropy

$$\Psi(x) := \sum_{i=1}^{K} x_i \log x_i, \tag{41}$$

with the Bregman divergence

$$D_\Psi(x, y) := \Psi(x) - \Psi(y) - \langle \nabla\Psi(y), x - y \rangle. \tag{42}$$

Negative entropy is strongly convex w.r.t. $\ell_1$ on the simplex and $\Psi$ is bounded on $\Delta_K$ (scale $\log K$). To avoid $\nabla\Psi(x)$ diverging on the boundary, the analysis implicitly works on a standard truncated/smoothed domain (e.g. $x_i \geq \varepsilon$) so that the resulting effects are confined to constants/log-factors.

The key modeling component is an entropy-regularized loss

$$\ell_t(x) := f_t(x) + \lambda_t \Psi(x), \qquad \lambda_t \geq 0, \tag{43}$$

where $\lambda_t$ is the scheduled **entropy coefficient**. Heuristically, larger $\lambda_t$ induces smoother, more randomized decisions (more exploration/robustness), while smaller $\lambda_t$ approaches greedy exploitation.

A.2.3. AES-OCO UPDATE RULE

Given nonnegative sequences $\{\lambda_t\}$ and step sizes $\{\eta_t\}$, AES-OCO performs mirror descent on $\ell_t$:

$$x_{t+1} = \arg\min_{x \in \Delta_K} \left\{ \eta_t \langle \nabla\ell_t(x_t), x \rangle + D_\Psi(x, x_t) \right\}, \tag{44}$$

where

$$\nabla\ell_t(x_t) = \nabla f_t(x_t) + \lambda_t \nabla\Psi(x_t). \tag{45}$$

A core structural choice (formalized below) is the **coupled step size**

$$\eta_t = c\,\lambda_t, \qquad c > 0, \tag{46}$$

which induces the canonical per-round trade-off

$$\text{(tracking cost)} \propto \frac{\text{non-stationarity}}{\lambda_t} \quad + \quad \text{(regularization cost)} \propto \lambda_t. \tag{47}$$

A.2.4. INTERFACE TO NON-STATIONARY MAXIMUM-ENTROPY RL

The end goal concerns adaptive temperature scheduling in non-stationary RL. The analysis uses the maximum-entropy (soft) formulation so that the soft-optimal policy takes a softmax form, enabling Lipschitz connections between value/advantage variations and policy variations. Conceptually:

1. **OCO layer:** derive a dynamic regret bound for AES-OCO with an explicit per-round trade-off, and provide an online rule for choosing $\lambda_t$ driven by an observable non-stationarity proxy;

2. **soft-RL layer:** view each state-wise policy update as an entropy-regularized OCO problem; transmit MDP variation to soft-$Q^\star$ variation via soft Bellman sensitivity, and then to policy path variation via softmax Lipschitzness;

3. **real RL error accounting:** when moving from full-information planning to sampling and function approximation, explicit bias and occupancy-mismatch terms appear; these are separated as additive error terms and paired with computable non-stationarity estimators to drive $\lambda_t$ online.

## A.3. From Dynamic Mirror Descent to a $\lambda$-Trade-off Bound

A foundational inequality couples mirror descent updates with a dynamic comparator. Let $\Pi \subset \mathbb{R}^K$ be a closed convex set (e.g., a truncated simplex). Let $\Psi : \Pi \to \mathbb{R}$ be differentiable and 1-strongly convex w.r.t. a norm $\|\cdot\|$, with dual norm $\|\cdot\|_*$. Define the Bregman divergence

$$D_\Psi(x, y) := \Psi(x) - \Psi(y) - \langle \nabla \Psi(y), x - y \rangle. \tag{48}$$

Given vectors $\{g_t\}$ and step sizes $\{\eta_t\}$, mirror descent performs

$$x_{t+1} = \arg\min_{x \in \Pi} \left\{ \eta_t \langle g_t, x \rangle + D_\Psi(x, x_t) \right\}. \tag{49}$$

**Lemma A.1** (Dynamic MD inequality). *Let $\{u_t\}_{t=1}^T \subset \Pi$ be any comparator sequence and set $u_0 = u_1$. Assume bounded mirror gradients: there exists $G_\Psi < \infty$ such that*

$$\|\nabla \Psi(z)\|_* \le G_\Psi, \qquad \forall z \in \Pi, \tag{50}$$

*and assume the step sizes are nondecreasing ($\eta_t \ge \eta_{t-1}$). Then the iterates generated by* (3) *satisfy*

$$\sum_{t=1}^T \langle g_t, x_t - u_t \rangle \le \frac{D_\Psi(u_1, x_1)}{\eta_1} + \sum_{t=1}^T \frac{\eta_t}{2} \|g_t\|_*^2 + \sum_{t=2}^T \frac{2G_\Psi}{\eta_t} \|u_t - u_{t-1}\|. \tag{51}$$

The inequality in Lemma 3.1 is the central switch: the first two terms match the static-comparator telescoping structure; the last term quantifies the extra cost of comparator drift. A complete proof (with a fully explicit dynamic telescoping argument under the stated conditions) is deferred to Appendix A.14.

### A.3.1. ENTROPY-REGULARIZED OCO AND DYNAMIC REGRET

Assume convex differentiable $f_t$ with bounded gradients

$$\|\nabla f_t(x)\|_\infty \le G, \qquad \forall x \in \Pi. \tag{52}$$

Define the regularized loss

$$\ell_t(x) := f_t(x) + \lambda_t \Psi(x). \tag{53}$$

Run mirror descent (3) on $\ell_t$ with $g_t = \nabla \ell_t(x_t)$. By convexity,

$$\ell_t(x_t) - \ell_t(u_t) \le \langle \nabla \ell_t(x_t), x_t - u_t \rangle. \tag{54}$$

Combining (8) with Lemma 3.1 yields

$$\sum_{t=1}^T \big( \ell_t(x_t) - \ell_t(u_t) \big) \le \frac{D_\Psi(u_1, x_1)}{\eta_1} + \sum_{t=1}^T \frac{\eta_t}{2} \|\nabla \ell_t(x_t)\|_*^2 + \sum_{t=2}^T \frac{2G_\Psi}{\eta_t} \|u_t - u_{t-1}\|. \tag{55}$$

The target is the **unregularized** dynamic regret

$$\mathrm{Reg}_T^{\mathrm{dyn}}(u_{1:T}) := \sum_{t=1}^T \big( f_t(x_t) - f_t(u_t) \big). \tag{56}$$

Using $\ell_t = f_t + \lambda_t \Psi$ gives the identity

$$f_t(x_t) - f_t(u_t) = \big( \ell_t(x_t) - \ell_t(u_t) \big) - \lambda_t \big( \Psi(x_t) - \Psi(u_t) \big). \tag{57}$$

Since $|\Psi(x)| \le \log K$ on the simplex,

$$-\lambda_t \big( \Psi(x_t) - \Psi(u_t) \big) \le 2\lambda_t \log K. \tag{58}$$

Combining (9)–(11) yields the master inequality

$$\mathrm{Reg}_T^{\mathrm{dyn}}(u_{1:T}) \le \frac{D_\Psi(u_1, x_1)}{\eta_1} + \sum_{t=1}^T \frac{\eta_t}{2} \|\nabla \ell_t(x_t)\|_*^2 + \sum_{t=2}^T \frac{2G_\Psi}{\eta_t} \|u_t - u_{t-1}\| + 2\log K \sum_{t=1}^T \lambda_t. \tag{59}$$

A.3.2. COUPLED STEP SIZES AND THE PER-ROUND TRADE-OFF

**(i) Coupled step size $\eta_t = c\lambda_t$.** Choose

$$\eta_t = c\,\lambda_t, \qquad c > 0, \tag{60}$$

and clip $\lambda_t \in [\lambda_{\min}, \lambda_{\max}]$ with $\lambda_{\min} > 0$ for numerical stability. Monotonicity of $\eta_t$ (required by Lemma 3.1) can be ensured by a doubling/epoch construction or by taking a monotone upper envelope.

**(ii) Bounding $\|\nabla\ell_t(x_t)\|_\infty$ and reducing higher-order terms.** In the $\ell_1/\ell_\infty$ pairing, $\|\cdot\|_* = \|\cdot\|_\infty$. Since $\nabla\ell_t = \nabla f_t + \lambda_t \nabla\Psi$, by (6) and (4),

$$\|\nabla\ell_t(x_t)\|_\infty \le G + \lambda_t G_\Psi. \tag{61}$$

Under a truncated simplex $\Delta_{K,\varepsilon}$, negative entropy satisfies

$$G_\Psi = \sup_{x \in \Delta_{K,\varepsilon}} \|\nabla\Psi(x)\|_\infty \le 1 + |\log\varepsilon|. \tag{62}$$

Substituting (13) and (14) into (12), and absorbing higher-order dependence on $\lambda_t$ using $\lambda_t \le \lambda_{\max}$, yields the following per-round one-dimensional trade-off.

**Theorem A.2** ($\lambda$-trade-off dynamic regret bound). *Run mirror descent (3) on $\ell_t$ with the coupled step sizes (13) and $\lambda_t \in [\lambda_{\min}, \lambda_{\max}]$ (with $\eta_t$ nondecreasing). Then there exist constants $C_0, C_1, C_2 > 0$ independent of $T$ (depending on $c, G, G_\Psi, \log K, \lambda_{\min}, \lambda_{\max}$) such that for any comparator sequence $u_{1:T} \subset \Pi$,*

$$\mathrm{Reg}_T^{\mathrm{dyn}}(u_{1:T}) \le C_0 + \sum_{t=2}^{T}\left(C_1\frac{\xi_t}{\lambda_t} + C_2\lambda_t\right), \qquad \xi_t := \|u_t - u_{t-1}\|_1. \tag{63}$$

*Equivalently, each round contributes the convex function*

$$\varphi_t(\lambda) := C_1\frac{\xi_t}{\lambda} + C_2\lambda, \qquad \lambda > 0. \tag{64}$$

The interpretation of (16) is explicit: the first term is the **tracking cost**, while the second term is the **regularization/stability cost**.

**Theorem A.3** (Per-round oracle optimal $\lambda_t$). *For each $t \ge 2$, define*

$$\lambda_t^\star := \arg\min_{\lambda>0} \varphi_t(\lambda) = \sqrt{\frac{C_1}{C_2}\,\xi_t}. \tag{65}$$

*Then*

$$\mathrm{Reg}_T^{\mathrm{dyn}}(u_{1:T}) \le C_0 + 2\sqrt{C_1 C_2}\sum_{t=2}^{T}\sqrt{\xi_t} \le C_0 + 2\sqrt{C_1 C_2}\sqrt{T\sum_{t=2}^{T}\xi_t}. \tag{66}$$

**A.4. AES-OCO: Fully Online Adaptive Scheduling**

This section upgrades the per-round oracle $\lambda_t^\star$ into a fully online schedule driven by an observable non-stationarity proxy. Start from (16) (absorbing constants into $C_0$ as needed) and rewrite it as

$$\mathrm{Reg}_T^{\mathrm{dyn}} \le C_0 \log K + \sum_{t=1}^{T}\left(C_1\frac{\xi_t}{\lambda_t} + C_2\lambda_t\right), \qquad \xi_t := \|u_t - u_{t-1}\|_1. \tag{67}$$

If a constant entropy coefficient $\lambda_t \equiv \lambda$ is used, then

$$\mathrm{Reg}_T^{\mathrm{dyn}} \le C_0 \log K + C_1\frac{A_T}{\lambda} + C_2 T\lambda, \qquad A_T := \sum_{t=1}^{T}\xi_t, \tag{68}$$

whose minimizer is

$$\lambda^{\mathrm{off}} = \sqrt{\frac{C_1 A_T}{C_2 T}}, \qquad \mathrm{Reg}_T^{\mathrm{dyn}} \le C_0 \log K + 2\sqrt{C_1 C_2 A_T T}. \tag{69}$$

Thus the optimal leading term is $\sqrt{A_T T}$ once the total non-stationarity $A_T$ is known.

In general OCO, $\xi_t$ is not directly observable. The subsequent soft-RL reduction will upper bound $\xi_t$ by an **observable/estimable** non-stationarity signal. The following result is therefore stated with a proxy.

Assume a nonnegative online proxy sequence $\{\widehat{\xi}_t\}$ such that for all $t$,

$$\xi_t \le \widehat{\xi}_t, \tag{70}$$

and define its prefix sum

$$\widehat{A}_t := \sum_{s=1}^{t} \widehat{\xi}_s. \tag{71}$$

Consider the no-restart schedule (before clipping)

$$\lambda_t = \sqrt{\frac{C_1}{C_2}} \sqrt{\frac{\widehat{A}_t}{t}}. \tag{72}$$

**Theorem A.4** (Online AES scheduling driven by an observable proxy). *Under the conditions of Theorem 3.2, and assuming the proxy condition (20), choosing $\lambda_t$ by (22) yields*

$$\mathrm{Reg}_T^{\mathrm{dyn}} \le C_0 \log K + 4\sqrt{C_1 C_2 \, T \, \widehat{A}_T}. \tag{73}$$

**Proof.** From (67) and (20),

$$\mathrm{Reg}_T^{\mathrm{dyn}} \le C_0 \log K + \sum_{t=1}^{T} \Big( C_1 \frac{\widehat{\xi}_t}{\lambda_t} + C_2 \lambda_t \Big). \tag{74}$$

Substituting (22) gives

$$\sum_{t=1}^{T} \Big( C_1 \frac{\widehat{\xi}_t}{\lambda_t} + C_2 \lambda_t \Big) = \sqrt{C_1 C_2} \sum_{t=1}^{T} \Big( \widehat{\xi}_t \frac{\sqrt{t}}{\sqrt{\widehat{A}_t}} + \frac{\sqrt{\widehat{A}_t}}{\sqrt{t}} \Big). \tag{75}$$

For the first term, $\sqrt{t} \le \sqrt{T}$ implies

$$\sum_{t=1}^{T} \widehat{\xi}_t \frac{\sqrt{t}}{\sqrt{\widehat{A}_t}} \le \sqrt{T} \sum_{t=1}^{T} \frac{\widehat{\xi}_t}{\sqrt{\widehat{A}_t}}. \tag{76}$$

The standard inequality (proved in Appendix A.18) gives

$$\sum_{t=1}^{T} \frac{\widehat{\xi}_t}{\sqrt{\widehat{A}_t}} \le 2\sqrt{\widehat{A}_T}, \tag{77}$$

so the first term is $\le 2\sqrt{T \widehat{A}_T}$. For the second term, Appendix A.18 also yields

$$\sum_{t=1}^{T} \frac{\sqrt{\widehat{A}_t}}{\sqrt{t}} \le 2\sqrt{T \widehat{A}_T}. \tag{78}$$

Plugging (76)–(78) into (75) and then into (74) yields (23). □

**Clipped implementation.** In practice, $\lambda_t$ is clipped for stability:

$$\lambda_t = \mathrm{clip}_{[\lambda_{\min}, \lambda_{\max}]} \left( \sqrt{\frac{C_1}{C_2}} \sqrt{\frac{\widehat{A}_t}{t}} \right). \tag{79}$$

Clipping introduces only lower-order additive compensation terms (e.g., when $\lambda_t = \lambda_{\max}$, the contribution $\sum \widehat{\xi}_t / \lambda_{\max}$ is bounded), formalized in Appendix A.19.

## A.5. AES-RL: Reduction for Non-stationary Soft RL

Consider a sequence of non-stationary discounted MDPs $\{M_t\}_{t=1}^{T}$ with

$$M_t = (\mathcal{S}, \mathcal{A}, P_t, r_t, \rho, \gamma), \qquad \gamma \in (0, 1), \tag{80}$$

and bounded rewards $|r_t(s, a)| \leq 1$. For any policy $\pi$, define the maximum-entropy (soft) return at time $t$:

$$J_t(\pi) := \mathbb{E}\left[\sum_{h=0}^{\infty} \gamma^h \big(r_t(s_h, a_h) + \mu\, H(\pi(\cdot \mid s_h))\big) \,\Big|\, s_0 \sim \rho,\ a_h \sim \pi(\cdot \mid s_h),\ s_{h+1} \sim P_t(\cdot \mid s_h, a_h)\right], \tag{81}$$

where $\mu > 0$ is a fixed baseline temperature and $\pi_t^\star \in \arg\max_\pi J_t(\pi)$ is the soft-optimal policy at round $t$. Define the RL dynamic regret

$$\text{Reg}_T^{\text{RL}} := \sum_{t=1}^{T} \big(J_t(\pi_t^\star) - J_t(\pi_t)\big). \tag{82}$$

### A.5.1. FROM SOFT-RL TO AN ENTROPY-REGULARIZED OCO SURROGATE

Let $Q_t^\star$ denote the soft-optimal $Q$-function of $M_t$ (satisfying the soft Bellman optimality equation). For each state $s$, define the local linear-entropy loss over the simplex $\Delta(\mathcal{A})$:

$$f_{t,s}(\pi_s) := -\langle Q_t^\star(s, \cdot), \pi_s \rangle + \mu\, \Psi(\pi_s), \qquad \Psi(p) := \sum_a p(a) \log p(a) = -H(p). \tag{83}$$

View the policy as a product variable $\pi = (\pi(\cdot \mid s))_{s \in \mathcal{S}} \in \prod_s \Delta(\mathcal{A})$ and define the separable mirror potential

$$\Psi_{\text{tot}}(\pi) := \sum_{s \in \mathcal{S}} \Psi(\pi(\cdot \mid s)). \tag{84}$$

Let $d_t^\star$ be the (normalized) discounted occupancy measure of $\pi_t^\star$ under $M_t$:

$$d_t^\star(s) := (1 - \gamma) \sum_{h \geq 0} \gamma^h \Pr_{M_t, \pi_t^\star}(s_h = s), \qquad \sum_s d_t^\star(s) = 1. \tag{85}$$

Define the per-round surrogate loss

$$F_t(\pi) := \sum_{s \in \mathcal{S}} d_t^\star(s)\, f_{t,s}\big(\pi(\cdot \mid s)\big). \tag{86}$$

A standard soft performance-difference decomposition (formalized in the appendix) yields that

$$J_t(\pi_t^\star) - J_t(\pi_t) \leq \frac{C_J}{1 - \gamma}\big(F_t(\pi_t) - F_t(\pi_t^\star)\big) + \text{Bias}_t, \tag{87}$$

where $C_J$ is a constant depending only on reward/entropy bounds, and $\text{Bias}_t$ collects the interface residuals (zero in an ideal planning/full-information setting; explicit in sampling/function approximation).

AES is applied by running mirror descent on

$$\ell_t(\pi) := F_t(\pi) + \lambda_t\, \Psi_{\text{tot}}(\pi), \tag{88}$$

with $\eta_t = c\lambda_t$ and the online schedule from Theorem 3.5 driven by an RL-specific observable proxy $\widehat{\xi}_t$.

### A.5.2. KEY BRIDGE: MDP VARIATION $\Rightarrow$ PATH VARIATION OF $\pi_t^\star$

Define the non-stationarity budget (reward + transition)

$$B_T^{\text{MDP}} := \sum_{t=2}^{T} \big(\Delta_t^r + \gamma V_{\max} \Delta_t^P\big), \tag{89}$$

where

$$\Delta_t^r := \sup_{s,a} |r_t(s, a) - r_{t-1}(s, a)|, \qquad \Delta_t^P := \sup_{s,a} \text{TV}\big(P_t(\cdot \mid s, a), P_{t-1}(\cdot \mid s, a)\big), \tag{90}$$

and $V_{\max} \geq \sup_{t,s} |V_t^\star(s)|$ (for bounded rewards, $V_{\max} = O((1 + \mu \log |\mathcal{A}|)/(1 - \gamma))$).

**(i) Soft Bellman sensitivity.** Let $\mathcal{T}_t$ be the soft Bellman optimality operator of $M_t$; it is a $\gamma$-contraction. The standard fixed-point perturbation bound gives

$$\|Q_t^\star - Q_{t-1}^\star\|_\infty \leq \frac{1}{1-\gamma}\left(\Delta_t^r + \gamma V_{\max}\Delta_t^P\right). \tag{91}$$

**(ii) Softmax Lipschitzness.** For each state $s$, the soft-optimal policy satisfies

$$\pi_t^\star(\cdot \mid s) = \mathrm{softmax}\left(Q_t^\star(s,\cdot)/\mu\right). \tag{92}$$

Using the $\ell_1$–$\ell_\infty$ Lipschitz property of softmax (proved in the appendix),

$$\|\pi_t^\star(\cdot \mid s) - \pi_{t-1}^\star(\cdot \mid s)\|_1 \leq \frac{1}{\mu}\|Q_t^\star - Q_{t-1}^\star\|_\infty. \tag{93}$$

Define the local path increment

$$\alpha_{t,s} := \|\pi_t^\star(\cdot \mid s) - \pi_{t-1}^\star(\cdot \mid s)\|_1. \tag{94}$$

Combining (26) and (27) yields

$$\sum_{t=2}^{T}\sum_{s\in\mathcal{S}}\alpha_{t,s} \leq \frac{|\mathcal{S}|}{\mu(1-\gamma)}B_T^{\mathrm{MDP}}. \tag{95}$$

Consequently, one may drive the OCO proxy by a quantity proportional to the (estimated) drift magnitude of soft-$Q^\star$ (or a stable surrogate such as critic drift in function approximation).

### A.5.3. OVERALL RESULT: PLANNING-VERSION BOUND (PRINCIPAL TERM)

**Theorem A.5** (AES-RL-Plan: non-stationary soft-RL dynamic regret (principal term))**.** *In the non-stationary tabular soft-MDP sequence above with bounded rewards, run AES with the per-round surrogate losses $f_{t,s}(\pi_s) = -\langle Q_t^\star(s,\cdot), \pi_s\rangle + \mu\Psi(\pi_s)$ and an online schedule $\{\lambda_t\}$ driven by a valid non-stationarity proxy (Theorem 3.5). Then there exists a constant $C$ such that*

$$\mathrm{Reg}_T^{\mathrm{RL}} \leq \widetilde{O}\left(\frac{\sqrt{|\mathcal{S}|\log|\mathcal{A}|}}{\mu(1-\gamma)^2}\sqrt{B_T^{\mathrm{MDP}}T}\right) + \sum_{t=1}^{T}\mathrm{Bias}_t, \tag{96}$$

*where $\widetilde{O}(\cdot)$ hides logarithmic factors in $T, |\mathcal{S}|$, and $\mathrm{Bias}_t$ collects the planning-to-RL interface residuals.*

**Corresponding appendices.** The soft Bellman sensitivity and the softmax Lipschitz lemma are provided in the appendices. The equalities-level performance-difference decomposition and the explicit form/control of $\mathrm{Bias}_t$ are deferred to the appendix material.

## A.6. From Planning to Real RL: Explicit Error Decomposition

The previous subsection emphasizes the principal non-stationarity term and assumes ideal objects (e.g. $Q_t^\star$ and $d_t^{\pi_t^\star}$). In real RL (especially off-policy learning with function approximation), two traceable deviations arise: **(i)** a $Q^\star$-substitution bias and **(ii)** an occupancy-weight mismatch error. A clean inequality separates the principal term and these errors; formal proofs and control strategies are deferred to the appendix.

Let

$$f_{t,s}(\pi_s) := -\langle Q_t^\star(s,\cdot), \pi_s\rangle + \mu\,\Psi(\pi_s), \qquad \Delta_{t,s} := f_{t,s}\left(\pi_t(\cdot \mid s)\right) - f_{t,s}\left(\pi_t^\star(\cdot \mid s)\right). \tag{97}$$

For any state-weighting distribution $\widetilde{d}_t$ used by the algorithm (e.g. on-policy $d_t^{\pi_t}$ or a replay-buffer empirical distribution), there exist error terms $\mathrm{bias}_t$ and $\mathrm{occ}_t(\widetilde{d}_t)$ such that for each $t$,

$$J_t(\pi_t^\star) - J_t(\pi_t) \leq \frac{1}{1-\gamma}\mathbb{E}_{s\sim\widetilde{d}_t}\left[\Delta_{t,s}\right] + \mathrm{bias}_t + \frac{1}{1-\gamma}\mathrm{occ}_t(\widetilde{d}_t). \tag{98}$$

Here $\mathrm{bias}_t$ arises from substituting $Q_t^\star$ for $Q_t^{\pi_t}$ (and from estimator errors when $Q$ is approximated), while $\mathrm{occ}_t(\widetilde{d}_t)$ arises from substituting $\widetilde{d}_t$ for $d_t^{\pi_t^\star}$. The main results keep $\sum_t \mathrm{bias}_t$ and $\sum_t \mathrm{occ}_t(\widetilde{d}_t)$ explicitly as additive terms; in algorithm instantiation and experiments they correspond to function-approximation error and off-policy distribution shift, which can be reduced by stabilization techniques (slow targets, conservative updates, reweighting, etc.).

## A.7. Calibrating an Observable Drift Proxy

This section explains how the conservative-proxy condition in Eq. (20) can be checked in practice. The main point is to turn an observable training signal into a quantity whose scale is safely comparable to the comparator drift used by the schedule. We first relate comparator drift to a measurable MDP-level surrogate, and then calibrate the observable proxy through a monotone upper envelope with conformal adjustment.

This section makes the theory–practice interface in Eq. (20) operational. The online AES schedule in Theorem 3.5 requires an *observable* sequence $\widehat{\xi}_t$ such that $\xi_t \le \widehat{\xi}_t$. In deep RL, $\xi_t$ (the drift of the optimal comparator) is not directly measurable. We therefore (i) upper bound $\xi_t$ by an MDP-level one-step drift quantity, and (ii) provide an empirical calibration procedure that turns an observable learning signal (our TD-error proxy) into a conservative upper bound with finite-sample coverage.

### A.7.1. FROM MDP DRIFT TO COMPARATOR DRIFT

Recall the one-step MDP variation terms used in Eq. (26):

$$\Delta_t^r := \sup_{s,a}\big|r_t(s,a) - r_{t-1}(s,a)\big|, \qquad \Delta_t^P := \sup_{s,a}\big\|P_t(\cdot \mid s,a) - P_{t-1}(\cdot \mid s,a)\big\|_1, \tag{99}$$

and define the scalar drift surrogate

$$b_t := \Delta_t^r + \gamma V_{\max}\Delta_t^P. \tag{100}$$

We instantiate the comparator drift in the RL bridge as the worst-case statewise change of the soft-optimal policy:

$$\xi_t^{\pi^\star} := \sup_{s \in \mathcal{S}}\big\|\pi_t^\star(\cdot \mid s) - \pi_{t-1}^\star(\cdot \mid s)\big\|_1. \tag{101}$$

**Lemma A.6** (Comparator drift is controlled by one-step MDP drift). *For each $t \ge 2$, under Eq. (26) (Appendix A.20) and Eq. (27) (Appendix A.21),*

$$\xi_t^{\pi^\star} \le \frac{1}{\mu}\|Q_t^\star - Q_{t-1}^\star\|_\infty \le \frac{1}{\mu(1-\gamma)}\,b_t. \tag{102}$$

*Proof.* The first inequality is exactly Eq. (27) applied to $\pi_t^\star(\cdot \mid s) = \mathrm{softmax}(Q_t^\star(s,\cdot)/\mu)$ and then taking $\sup_s$. The second inequality is Eq. (26) with the definition of $b_t$ in (100). Combining the two yields (102). $\qquad\square$

### A.7.2. A MONOTONE UPPER-ENVELOPE CALIBRATION FROM AN OBSERVABLE PROXY

Our default observable proxy is the TD-error quantile (Eq. (31)), which we denote by $v_t \ge 0$ for brevity. To connect $v_t$ to the drift surrogate $b_t$, we construct a monotone upper envelope $g : \mathbb{R}_+ \to \mathbb{R}_+$ such that $b_t \le g(v_t)$ holds with high probability on the target benchmark family.

**Calibration data.** On injected-drift benchmarks, an MDP-level drift surrogate $b_t$ in (100) is measurable (or can be conservatively upper bounded) because the drift is programmatically introduced. Collect a calibration set

$$\mathcal{D}_{\mathrm{cal}} = \{(v_i, b_i)\}_{i=1}^N, \tag{103}$$

where $v_i$ is the observable proxy computed at update $i$ and $b_i$ is the corresponding drift surrogate.

**Step 1: fit a monotone predictor.** Fit a nondecreasing function $g_0$ (e.g., isotonic regression) by

$$g_0 \in \arg\min_{g \in \mathcal{G}_\uparrow} \sum_{i=1}^N (b_i - g(v_i))^2, \tag{104}$$

where $\mathcal{G}_\uparrow$ is the class of nondecreasing functions on $\mathbb{R}_+$.

**Step 2: conformalize into an upper bound.** Define residuals $r_i := b_i - g_0(v_i)$ and let $q_{1-\delta}$ be the $(1-\delta)$ upper quantile of $\{r_i\}_{i=1}^N$ (using the standard finite-sample conformal index, e.g., the $\lceil (N+1)(1-\delta) \rceil$-th order statistic). Define the calibrated upper envelope

$$g(v) := g_0(v) + q_{1-\delta}. \tag{105}$$

**Lemma A.7** (Finite-sample coverage of the upper envelope). *Assume $(v_i, b_i)_{i=1}^N$ and a fresh pair $(v_{\text{new}}, b_{\text{new}})$ are exchangeable (e.g., drawn from the same benchmark family and evaluation protocol). Then the conformalized envelope in (105) satisfies*

$$\mathbb{P}(b_{\text{new}} \leq g(v_{\text{new}})) \geq 1 - \delta. \tag{106}$$

*Proof.* This is the standard one-sided conformal prediction guarantee. Under exchangeability, the rank of $r_{\text{new}} := b_{\text{new}} - g_0(v_{\text{new}})$ among $\{r_i\}_{i=1}^N \cup \{r_{\text{new}}\}$ is uniform. By the choice of $q_{1-\delta}$ as the appropriate order statistic, $\mathbb{P}(r_{\text{new}} \leq q_{1-\delta}) \geq 1 - \delta$, which is equivalent to $b_{\text{new}} \leq g_0(v_{\text{new}}) + q_{1-\delta} = g(v_{\text{new}})$. $\square$

### A.7.3. CONSTRUCTING A CONSERVATIVE $\widehat{\xi}_t$ FOR AES

Combining Lemma A.6 with Lemma A.7 yields a concrete conservative proxy for Eq. (20). Define

$$\widehat{\xi}_t := \frac{1}{\mu(1-\gamma)} g(v_t), \tag{107}$$

where $v_t$ is the observable proxy (e.g., TD-error quantile) and $g$ is the calibrated envelope.

**Theorem A.8** (Calibrated proxy implies Eq. (20) with high probability). *Under the assumptions of Lemma A.6 and Lemma A.7, we have*

$$\mathbb{P}\left( \xi_t^{\pi^\star} \leq \widehat{\xi}_t \right) \geq 1 - \delta, \tag{108}$$

*where $\widehat{\xi}_t$ is defined in (107). Consequently, the online AES schedule in Theorem 3.5 applies with the observable proxy sequence $\widehat{\xi}_t$.*

*Proof.* By Lemma A.6, $\xi_t^{\pi^\star} \leq \frac{1}{\mu(1-\gamma)} b_t$. By Lemma A.7, $b_t \leq g(v_t)$ with probability at least $1 - \delta$ (under exchangeability). Combining yields $\xi_t^{\pi^\star} \leq \frac{1}{\mu(1-\gamma)} g(v_t) = \widehat{\xi}_t$ with probability at least $1 - \delta$. $\square$

**Practical note.** When $\Delta_t^P$ is not directly accessible, one may calibrate against any conservative and computable surrogate $\tilde{b}_t$ (e.g., estimated via a fixed probe set of $(s, a)$ pairs and repeated next-state sampling in simulation). All statements above then hold with $b_t$ replaced by $\tilde{b}_t$, producing a conservative schedule on the chosen benchmark family.

### A.8. From Planning to Learning: Exact Decomposition and Residual Terms

This section makes the planning-to-learning gap explicit. It isolates the principal surrogate term used by Theorem 3.6 and then writes the remaining contributions in a form that can be read directly as residual terms. The purpose is to show where $Q^\star$-substitution, critic estimation, and occupancy mismatch enter when the ideal planning objects are replaced by practical RL surrogates.

This appendix provides a strict (identity-level) decomposition of the soft return gap and clarifies the origin and definition of the error terms (bias and occupancy mismatch) that arise when interfacing RL with a per-state OCO surrogate.

### A.8.1. NOTATION: DISCOUNTED OCCUPANCY AND MAXIMUM-ENTROPY RETURN

Fix $\mathcal{M}_t = (\mathcal{S}, \mathcal{A}, P_t, r_t, \gamma)$. For a policy $\pi$, define the discounted state occupancy

$$d_t^\pi(s) := (1-\gamma) \sum_{h \geq 0} \gamma^h \Pr(s_h = s \mid \pi, P_t, \rho), \tag{109}$$

where $\rho$ is the initial state distribution. For any measurable $g(s, a)$,

$$\frac{1}{1-\gamma} \mathbb{E}_{s \sim d_t^\pi, \, a \sim \pi(\cdot|s)}[g(s, a)] = \mathbb{E}\left[ \sum_{h \geq 0} \gamma^h g(s_h, a_h) \right]. \tag{110}$$

The soft objective is

$$J_t(\pi) := \frac{1}{1-\gamma} \mathbb{E}_{s \sim d_t^\pi,\, a \sim \pi(\cdot|s)} \big[ r_t(s, a) - \mu \log \pi(a \mid s) \big]. \tag{111}$$

Define the soft policy-evaluation quantities:

$$Q_t^\pi(s, a) := r_t(s, a) + \gamma \, \mathbb{E}_{s' \sim P_t(\cdot|s,a)}[V_t^\pi(s')],$$
$$V_t^\pi(s) := \mathbb{E}_{a \sim \pi(\cdot|s)}\big[ Q_t^\pi(s, a) - \mu \log \pi(a \mid s) \big]. \tag{112}$$

Define the regularized advantage

$$\widetilde{A}_t^\pi(s, a) := Q_t^\pi(s, a) - \mu \log \pi(a \mid s) - V_t^\pi(s), \tag{113}$$

which satisfies $\mathbb{E}_{a \sim \pi(\cdot|s)}[\widetilde{A}_t^\pi(s, a)] = 0$ for all $s$.

### A.8.2. SOFT PERFORMANCE DIFFERENCE LEMMA (IDENTITY)

**Lemma A.9** (Soft (regularized) performance difference lemma). *For any two policies $\pi, \pi'$,*

$$J_t(\pi') - J_t(\pi) = \frac{1}{1-\gamma} \mathbb{E}_{s \sim d_t^{\pi'}} \Big[ \mathbb{E}_{a \sim \pi'(\cdot|s)}[\widetilde{A}_t^\pi(s, a)] - \mu \, \mathrm{KL}\big(\pi'(\cdot \mid s) \,\|\, \pi(\cdot \mid s)\big) \Big]. \tag{114}$$

*Proof.* Define the soft Bellman operator for policy $\pi'$ acting on a value function $V$:

$$(\mathcal{T}_t^{\pi'} V)(s) := \mathbb{E}_{a \sim \pi'(\cdot|s)} \big[ r_t(s, a) - \mu \log \pi'(a \mid s) + \gamma \, \mathbb{E}_{s' \sim P_t(\cdot|s,a)} V(s') \big]. \tag{115}$$

Then $V_t^{\pi'}$ is its fixed point. Using the linear system representation,

$$V_t^{\pi'} - V_t^\pi = (I - \gamma P_t^{\pi'})^{-1} \big( \mathcal{T}_t^{\pi'} V_t^\pi - V_t^\pi \big), \tag{116}$$

and taking expectation over $s_0 \sim \rho$ together with the occupancy identity (110) gives

$$J_t(\pi') - J_t(\pi) = \frac{1}{1-\gamma} \mathbb{E}_{s \sim d_t^{\pi'}} \big[ (\mathcal{T}_t^{\pi'} V_t^\pi)(s) - V_t^\pi(s) \big]. \tag{117}$$

Moreover,

$$(\mathcal{T}_t^{\pi'} V_t^\pi)(s) = \mathbb{E}_{a \sim \pi'(\cdot|s)}\big[ Q_t^\pi(s, a) - \mu \log \pi'(a \mid s) \big], \qquad V_t^\pi(s) = \mathbb{E}_{a \sim \pi(\cdot|s)}\big[ Q_t^\pi(s, a) - \mu \log \pi(a \mid s) \big]. \tag{118}$$

Add and subtract $-\mu \log \pi(a \mid s)$ inside the $\pi'$-expectation and use (113):

$$(\mathcal{T}_t^{\pi'} V_t^\pi)(s) - V_t^\pi(s) = \mathbb{E}_{a \sim \pi'(\cdot|s)}[\widetilde{A}_t^\pi(s, a)] + \mu \, \mathbb{E}_{a \sim \pi'(\cdot|s)}[\log \pi(a \mid s) - \log \pi'(a \mid s)] = \mathbb{E}_{a \sim \pi'}[\widetilde{A}_t^\pi] - \mu \, \mathrm{KL}(\pi'\|\pi). \tag{119}$$

Substituting into (117) yields (114). $\qquad\square$

### A.8.3. A PER-STATE CONVEX SURROGATE AND THE INEVITABILITY OF BIAS TERMS

The OCO interface uses the ideal per-state surrogate (defined using $Q_t^\star$):

$$f_{t,s}(\pi_s) := -\langle Q_t^\star(s, \cdot), \pi_s \rangle + \mu \sum_a \pi_s(a) \log \pi_s(a). \tag{120}$$

Let $\pi_t^\star(\cdot \mid s) = \mathrm{softmax}(Q_t^\star(s, \cdot)/\mu)$. Then:

**Lemma A.10** (Fenchel–Young identity: surrogate gap equals $\mu \cdot \mathrm{KL}$). *For any state $s$ and any distribution $\pi(\cdot \mid s)$,*

$$f_{t,s}(\pi(\cdot \mid s)) - f_{t,s}(\pi_t^\star(\cdot \mid s)) = \mu \, \mathrm{KL}\big(\pi(\cdot \mid s) \,\|\, \pi_t^\star(\cdot \mid s)\big). \tag{121}$$

This identity is the key for translating soft RL into per-state strongly convex OCO losses. However, Lemma A.9 decomposes $J_t(\pi_t^\star) - J_t(\pi_t)$ using $(i)$ the optimal occupancy $d_t^{\pi_t^\star}$, $(ii)$ the regularized advantage $\widetilde{A}_t^{\pi_t}$ (which depends on $Q_t^{\pi_t}$), and $(iii)$ the reverse-direction KL term $\mathrm{KL}(\pi_t^\star\|\pi_t)$. In contrast, (121) uses $Q_t^\star$ and the forward KL $\mathrm{KL}(\pi_t\|\pi_t^\star)$, and implementations may further replace $Q_t^\star$ by a critic estimate. Consequently, additional error terms are unavoidable when the strict RL identity is expressed in the surrogate form.

A.8.4. FORMAL DEFINITIONS OF BIAS AND OCCUPANCY MISMATCH

Introduce a "planning-style" objective that evaluates $\pi$ under the optimal occupancy and $Q_t^\star$:

$$\widetilde{J}_t(\pi) := \frac{1}{1-\gamma} \mathbb{E}_{s \sim d_t^{\pi_t^\star}, \, a \sim \pi(\cdot|s)} \big[ Q_t^\star(s,a) - \mu \log \pi(a \mid s) \big]. \tag{122}$$

By definition, $J_t(\pi_t^\star) = \widetilde{J}_t(\pi_t^\star)$. Moreover, combining Lemma A.10 with (122) gives the exact "main term" identity

$$J_t(\pi_t^\star) - \widetilde{J}_t(\pi_t) = \frac{1}{1-\gamma} \mathbb{E}_{s \sim d_t^{\pi_t^\star}} [f_{t,s}(\pi_t(\cdot \mid s)) - f_{t,s}(\pi_t^\star(\cdot \mid s))] = \frac{\mu}{1-\gamma} \mathbb{E}_{s \sim d_t^{\pi_t^\star}} \mathrm{KL}\big(\pi_t(\cdot \mid s) \,\|\, \pi_t^\star(\cdot \mid s)\big). \tag{123}$$

The true return gap decomposes as

$$J_t(\pi_t^\star) - J_t(\pi_t) = \underbrace{\big(J_t(\pi_t^\star) - \widetilde{J}_t(\pi_t)\big)}_{\text{ideal OCO/planning main term}} + \underbrace{\big(\widetilde{J}_t(\pi_t) - J_t(\pi_t)\big)}_{\text{interface errors (bias, occupancy mismatch, estimation)}}. \tag{124}$$

The second bracket in (124) is the source of the bias terms and distribution-mismatch terms.

**(i) $Q$-substitution bias ($Q_t^\star$ vs. $Q_t^{\pi_t}$).** Define

$$\mathrm{Bias}_t^{(Q)} := \frac{1}{1-\gamma} \mathbb{E}_{s \sim d_t^{\pi_t^\star}, \, a \sim \pi_t(\cdot|s)} \big[ Q_t^\star(s,a) - Q_t^{\pi_t}(s,a) \big]. \tag{125}$$

A direct bound is

$$|\mathrm{Bias}_t^{(Q)}| \le \frac{1}{1-\gamma} \| Q_t^\star - Q_t^{\pi_t} \|_\infty. \tag{126}$$

In soft actor–critic with function approximation and bootstrapping, $\| Q_t^\star - Q_t^{\pi_t} \|_\infty$ need not vanish; the main theorem therefore keeps $\sum_t \mathrm{Bias}_t^{(Q)}$ as an explicit additive term, interpreted as approximation/evaluation error.

**(ii) Estimation error (critic estimate vs. target $Q$).** If the algorithm uses a critic estimate $\widehat{Q}_t$ (or a target network), an additional error may be tracked, e.g.

$$\mathrm{Bias}_t^{(\mathrm{est})} := \frac{1}{1-\gamma} \mathbb{E}_{s \sim d_t^{\pi_t^\star}, \, a \sim \pi_t(\cdot|s)} \big[ \widehat{Q}_t(s,a) - Q_t^\star(s,a) \big], \qquad |\mathrm{Bias}_t^{(\mathrm{est})}| \le \frac{1}{1-\gamma} \| \widehat{Q}_t - Q_t^\star \|_\infty. \tag{127}$$

**(iii) Occupancy mismatch.** If the analysis or implementation uses a substitute state distribution $\widetilde{d}_t$ (e.g. on-policy $d_t^{\pi_t}$ or a replay-buffer distribution), define

$$\phi_t(s) := f_{t,s}(\pi_t(\cdot \mid s)) - f_{t,s}(\pi_t^\star(\cdot \mid s)) \,(\ge 0), \qquad \mathrm{OccErr}_t(\widetilde{d}_t) := \frac{1}{1-\gamma} \Big( \mathbb{E}_{s \sim d_t^{\pi_t^\star}}[\phi_t(s)] - \mathbb{E}_{s \sim \widetilde{d}_t}[\phi_t(s)] \Big). \tag{128}$$

The standard bound is

$$|\mathrm{OccErr}_t(\widetilde{d}_t)| \le \frac{1}{1-\gamma} \| \phi_t \|_\infty \| d_t^{\pi_t^\star} - \widetilde{d}_t \|_1. \tag{129}$$

## A.9. A Residual Bound for the Planning-to-Learning Gap

This section turns the decomposition from Appendix A.8 into a usable upper bound. The statements here are the formal counterpart of the discussion below Theorem 3.6: the drift-dependent principal term remains visible, while the remaining learning-side effects are gathered into explicit additive residuals.

This appendix states and proves a clean inequality separating the ideal per-state surrogate main term from the interface errors (bias and occupancy mismatch). It also provides explicit bounds that make the mismatch term directly usable.

A.9.1. OCCUPANCY FORM AND SOFT PERFORMANCE DIFFERENCE LEMMA

The definitions (109)–(114) already yield the occupancy form

$$J_t(\pi) = \frac{1}{1-\gamma} \mathbb{E}_{s \sim d_t^\pi, \, a \sim \pi(\cdot|s)} \big[ r_t(s,a) - \mu \log \pi(a \mid s) \big], \tag{130}$$

and the strict identity (Lemma A.9) can be reused as needed.

A.9.2. A BIAS TERM THAT IS DIRECTLY CONTROLLED BY $\|Q_t^{\pi_t} - Q_t^\star\|_\infty$

A convenient scalar bias term arising from $Q$-substitution is

$$\text{Bias}_t^{(Q)} := \frac{1}{1-\gamma} \mathbb{E}_{s \sim d_t^{\pi_t^\star}, \, a \sim \pi_t^\star(\cdot|s)} \big[ Q_t^{\pi_t}(s,a) - Q_t^\star(s,a) \big], \tag{131}$$

which admits the immediate bound

$$|\text{Bias}_t^{(Q)}| \leq \frac{1}{1-\gamma} \|Q_t^{\pi_t} - Q_t^\star\|_\infty. \tag{132}$$

The main theorem can keep $\sum_{t=1}^T |\text{Bias}_t^{(Q)}|$ explicit. Any rate claim beyond this requires additional assumptions on evaluation/approximation accuracy (e.g. a decay $\|Q_t^{\pi_t} - Q_t^\star\|_\infty = O(t^{-1/2})$ would imply an $O(\sqrt{T})$ cumulative contribution).

A.9.3. OCCUPANCY MISMATCH: FROM DISTRIBUTION REPLACEMENT TO AN $\ell_1$ BOUND

Recall the statewise surrogate gap

$$\Delta_t(s) := f_{t,s}(\pi_t(\cdot \mid s)) - f_{t,s}(\pi_t^\star(\cdot \mid s)) \geq 0, \tag{133}$$

where

$$f_{t,s}(\pi_s) = -\langle Q_t^\star(s, \cdot), \pi_s \rangle + \mu \sum_a \pi_s(a) \log \pi_s(a). \tag{134}$$

For any substitute distribution $\widetilde{d}_t$ define

$$\text{OccErr}_t(\widetilde{d}_t) := \frac{1}{1-\gamma} \Big( \mathbb{E}_{s \sim d_t^{\pi_t^\star}}[\Delta_t(s)] - \mathbb{E}_{s \sim \widetilde{d}_t}[\Delta_t(s)] \Big). \tag{135}$$

Then, for bounded $\Delta_t$,

$$|\text{OccErr}_t(\widetilde{d}_t)| \leq \frac{1}{1-\gamma} \|\Delta_t\|_\infty \|d_t^{\pi_t^\star} - \widetilde{d}_t\|_1. \tag{136}$$

A.9.4. AN EXPLICIT BOUND ON $\|\Delta_t\|_\infty$

Assume $|r_t(s,a)| \leq R_{\max}$ so that $\|Q_t^\star\|_\infty \leq Q_{\max}$ with $Q_{\max}$ defined in (261). Then for any fixed state $s$ and any $\pi_s \in \Delta_{\mathcal{A}}$,

$$-\langle Q_t^\star(s, \cdot), \pi_s \rangle \in [-Q_{\max}, Q_{\max}], \qquad \sum_a \pi_s(a) \log \pi_s(a) \in [-\log|\mathcal{A}|, 0]. \tag{137}$$

Hence $f_{t,s}(\pi_s) \in [-Q_{\max} - \mu \log|\mathcal{A}|, \, Q_{\max}]$, so the difference between any two values is at most the interval length. Therefore,

$$0 \leq \Delta_t(s) \leq 2Q_{\max} + \mu \log|\mathcal{A}|, \qquad \forall s, \tag{138}$$

and

$$\|\Delta_t\|_\infty \leq 2Q_{\max} + \mu \log|\mathcal{A}|. \tag{139}$$

Substituting (139) into (136) gives a fully explicit mismatch bound:

$$|\text{OccErr}_t(\widetilde{d}_t)| \leq \frac{2Q_{\max} + \mu \log|\mathcal{A}|}{1-\gamma} \|d_t^{\pi_t^\star} - \widetilde{d}_t\|_1. \tag{140}$$

A.9.5. OPTIONAL STRENGTHENING FOR ON-POLICY WEIGHTING

When $\widetilde{d}_t = d_t^{\pi_t}$ (on-policy sampling), the occupancy distance can be further controlled by a uniform per-state policy difference. Define the induced state-transition kernel

$$P_t^\pi(s' \mid s) := \mathbb{E}_{a \sim \pi(\cdot|s)}[P_t(s' \mid s, a)], \tag{141}$$

and the mixed norm

$$\|\pi - \pi'\|_{1,\infty} := \sup_{s \in \mathcal{S}} \|\pi(\cdot \mid s) - \pi'(\cdot \mid s)\|_1. \tag{142}$$

Then the standard resolvent/telescoping argument yields

$$\|d_t^\pi - d_t^{\pi'}\|_1 \le \frac{\gamma}{1-\gamma} \sup_s \|P_t^\pi(\cdot \mid s) - P_t^{\pi'}(\cdot \mid s)\|_1 \le \frac{\gamma}{1-\gamma} \|\pi - \pi'\|_{1,\infty}. \tag{143}$$

Combining (140) and (143) gives an on-policy mismatch control in terms of $\|\pi_t^\star - \pi_t\|_{1,\infty}$.

### A.10. Technical Tools for Appendix A

The appendices collected below provide the supporting algebra, auxiliary inequalities, and deferred proofs used by the core theory chain. They are kept separate so that the main line of the argument can be read without interruption, while every technical step remains fully checkable.

### A.11. Technical Preliminaries

The appendices from here onward collect background material and deferred derivations that support the core chain above. They are written for local consultation: each section explains one ingredient that is referenced from the main theory appendix or from Section 3.

#### A.11.1. THE SIMPLEX AND THE $\ell_1/\ell_\infty$ DUALITY

**Simplex and truncated simplex.** For probability vectors of dimension $K$, define the simplex

$$\Delta_K := \left\{ x \in \mathbb{R}_+^K : \sum_{i=1}^K x_i = 1 \right\} \tag{144}$$

Several arguments require working on a truncated (or smoothed) simplex

$$\Delta_{K,\varepsilon} := \{ x \in \Delta_K : x_i \ge \varepsilon, \ \forall i \in [K] \}, \qquad \varepsilon \in (0, 1/K]. \tag{145}$$

A typical choice is $\varepsilon = \Theta(T^{-2})$, which only affects logarithmic factors; see Appendix A.11.3.

**Norm convention and duality.** The analysis uses

$$|x|_1 := \sum_{i=1}^K |x_i|, \qquad |x|_\infty := \max_{i \in [K]} |x_i|. \tag{146}$$

They are dual norms. In particular, for any $x, y \in \mathbb{R}^K$, Hölder's inequality yields

$$\langle x, y \rangle \le |x|_1 |y|_\infty, \qquad \langle x, y \rangle \le |x|_\infty |y|_1. \tag{147}$$

This pairing is natural for negative-entropy mirror descent on $\Delta_K$: path variation of comparators is measured in $|\cdot|_1$, while gradients are controlled in $|\cdot|_\infty$.

#### A.11.2. NEGATIVE ENTROPY: PROPERTIES AND BREGMAN IDENTITIES

**Definition.** Let $\log$ denote the natural logarithm. Define the negative entropy on $\Delta_K$:

$$\Psi(x) := \sum_{i=1}^K x_i \log x_i, \qquad x \in \Delta_K, \tag{148}$$

with the standard convention $0 \log 0 := 0$. Thus $\Psi(x) = -H(x)$ where $H$ is Shannon entropy.

**Boundedness on $\Delta_K$.** For all $x \in \Delta_K$,
$$-\log K \le \Psi(x) \le 0. \tag{149}$$

The minimum is attained at the uniform distribution $x = \mathbf{1}/K$, giving $\Psi(\mathbf{1}/K) = -\log K$, and the maximum is attained at vertices, where $\Psi(e_j) = 0$.

**Bregman divergence equals KL divergence.** The Bregman divergence induced by $\Psi$ is

$$D_\Psi(x, y) := \Psi(x) - \Psi(y) - \langle \nabla\Psi(y), x - y \rangle. \tag{150}$$

For $x, y \in \Delta_K$ (with the usual convention that the divergence is $+\infty$ if $y_i = 0$ but $x_i > 0$),

$$D_\Psi(x, y) = \mathrm{KL}(x|y) := \sum_{i=1}^K x_i \log \frac{x_i}{y_i}. \tag{151}$$

Hence $D_\Psi(x, y) \geq 0$ and equals 0 iff $x = y$.

**Strong convexity w.r.t. $|\cdot|_1$.** Negative entropy is 1-strongly convex with respect to $|\cdot|_1$ on $\Delta_K$ in the following standard form:

$$\Psi(x) \geq \Psi(y) + \langle \nabla\Psi(y), x - y \rangle + \frac{1}{2}|x - y|_1^2, \qquad \forall x, y \in \Delta_K. \tag{152}$$

Equivalently,

$$D_\Psi(x, y) \geq \frac{1}{2}|x - y|_1^2. \tag{153}$$

A convenient route is Pinsker's inequality, which implies $\mathrm{KL}(x|y) \geq \frac{1}{2}|x - y|_1^2$.

**Bregman identities.** Two standard identities are repeatedly used.

- **Three-point identity.** For any $a, b, c$ in the domain of $\Psi$,

$$\langle \nabla\Psi(b) - \nabla\Psi(c), a - b \rangle = D_\Psi(a, c) - D_\Psi(a, b) - D_\Psi(b, c) \tag{154}$$

- **Add–subtract decomposition.** For any $a, b, c$,

$$D_\Psi(a, c) - D_\Psi(a, b) = \langle \nabla\Psi(b) - \nabla\Psi(c), a - b \rangle + D_\Psi(b, c). \tag{155}$$

A.11.3. TRUNCATED/SMOOTHED SIMPLEX: BOUNDED GRADIENTS AND CURVATURE

Negative entropy has a technical singularity: if some coordinate approaches 0, then $\log x_i \to -\infty$, so $\Psi$ is no longer Lipschitz and $|\nabla\Psi(x)|_\infty$ becomes unbounded. This affects: (i) controlling constants through $\sup_{x \in \Pi} |\nabla\Psi(x)|_\infty$; and (ii) any use of smoothness/curvature upper bounds.

A standard remedy is to work on $\Delta_{K,\varepsilon}$ or equivalently apply a small smoothing to the iterates.

**Gradient bound on $\Delta_{K,\varepsilon}$.** For $x \in \Delta_{K,\varepsilon}$, all coordinates are positive and

$$\frac{\partial\Psi(x)}{\partial x_i} = 1 + \log x_i. \tag{156}$$

Since $\varepsilon \leq x_i \leq 1$, it holds that $\log \varepsilon \leq \log x_i \leq 0$, hence

$$|\nabla\Psi(x)|_\infty \leq 1 + |\log \varepsilon|. \tag{157}$$

**Hessian/curvature bound on $\Delta_{K,\varepsilon}$.** On $\Delta_{K,\varepsilon}$,

$$\nabla^2\Psi(x) = \mathrm{diag}\Big(\frac{1}{x_1}, \ldots, \frac{1}{x_K}\Big), \qquad \text{so} \quad |\nabla^2\Psi(x)|_{\mathrm{op}} \leq \frac{1}{\varepsilon}. \tag{158}$$

Thus $\Psi$ is $1/\varepsilon$-smooth in Euclidean geometry; when needed, norm conversions can translate this into the appropriate $\ell_1/\ell_\infty$ constants.

**Algorithmic realizations.** Two equivalent implementation patterns are common:

- **Hard projection:** project each iterate back to $\Delta_{K,\varepsilon}$ (e.g., clip small coordinates and renormalize).

- **Smoothing:** replace an output distribution $\pi$ by $(1-\xi)\pi + \xi\mathbf{1}/K$ with tiny $\xi$, which guarantees $\pi_i \geq \xi/K$.

Both only introduce logarithmic factors (via $|\log\varepsilon|$) and do not change the main rates.

## A.12. Equivalent Forms of the AES–OCO Update

### A.12.1. PRIMAL AND DUAL FORMS FOR NEGATIVE-ENTROPY MIRROR DESCENT

Consider mirror descent on $\Pi = \Delta_K$ with mirror map $\Psi$ and step size $\eta_t > 0$:

$$x_{t+1} = \arg\min_{x\in\Delta_K} \left\{ \eta_t\langle g_t, x\rangle + D_\Psi(x, x_t) \right\}. \tag{159}$$

For $\Psi(x) = \sum_i x_i \log x_i$, (B.1) is equivalent to an exponentiated-gradient update. Writing the dual variable as $y_t := \nabla\Psi(x_t)$, the optimality condition implies

$$\nabla\Psi(x_{t+1}) = \nabla\Psi(x_t) - \eta_t g_t + \nu_t \mathbf{1}, \tag{160}$$

where $\nu_t$ enforces the simplex constraint. Exponentiating coordinatewise yields

$$x_{t+1,i} \propto x_{t,i}\exp(-\eta_t g_{t,i}), \qquad i \in [K], \tag{161}$$

followed by normalization. Thus AES–OCO can be implemented either via the primal proximal form (B.1) or the multiplicative weights form (B.3).

**Notation remark (entropy coefficient vs. inverse temperature).** In the multiplicative form, the scalar step size $\eta_t$ acts as an inverse temperature in the sense that, for a one-step update with $g_t = -r_t$ and a uniform $x_t$, the resulting $x_{t+1}$ has a softmax form with temperature proportional to $1/\eta_t$. To avoid symbol confusion in the experiments, we denote the scheduled inverse-temperature knob by $\beta_t$ and report the corresponding entropy coefficient in the regularized objective via $\lambda_t := 1/\beta_t$.

### A.12.2. WHY COUPLING $\eta_t = c\lambda_t$ PRODUCES A ONE-KNOB TRADE-OFF

The core regret bound in the paper contains, per round, two antagonistic contributions: a **tracking term** scaling like $\xi_t/\eta_t$ (from dynamic comparators), and a **stability/regularization term** scaling like $\eta_t$ (from gradient control). When the per-round objective is

$$\ell_t(x) = f_t(x) + \lambda_t\Psi(x), \tag{162}$$

a natural design is to couple the mirror step size with the entropy weight:

$$\eta_t = c\lambda_t, \qquad c > 0. \tag{163}$$

Under this coupling, the tracking part becomes proportional to $\xi_t/\lambda_t$, while the stability part becomes proportional to $\lambda_t$. This yields a per-round scalar trade-off of the form

$$C_1\frac{\alpha_t}{\lambda_t} + C_2\lambda_t, \tag{164}$$

which can be optimized either offline (Appendix A.17) or online via adaptive schedules (Appendix A.18 and Appendix A.19).

## A.13. A Dynamic Mirror-Descent Inequality

### A.13.1. ONE-STEP MIRROR DESCENT INEQUALITY (STATIC COMPARATOR)

Let $\Pi \subseteq \mathbb{R}^d$ be closed and convex. Fix a norm $|\cdot|$ with dual norm $|\cdot|_*$. Assume $\Psi : \Pi \to \mathbb{R}$ is differentiable and 1-strongly convex w.r.t. $|\cdot|$, i.e.,

$$\Psi(x) \geq \Psi(y) + \langle\nabla\Psi(y), x - y\rangle + \frac{1}{2}|x - y|^2, \qquad \forall x, y \in \Pi. \tag{165}$$

Define $D_\Psi$ as usual. Consider one mirror descent step:

$$x_{t+1} = \arg\min_{x \in \Pi} \{\eta_t \langle g_t, x \rangle + D_\Psi(x, x_t)\}, \qquad \eta_t > 0. \tag{166}$$

**Lemma A.11** (One-step MD inequality (static comparator)). *For any fixed $u \in \Pi$, the update (C.1) satisfies*

$$\langle g_t, x_t - u \rangle \le \frac{D_\Psi(u, x_t) - D_\Psi(u, x_{t+1})}{\eta_t} + \frac{\eta_t}{2} |g_t|_*^2. \tag{167}$$

*Proof.* First-order optimality of (C.1) implies that for any $x \in \Pi$,

$$\langle \eta_t g_t + \nabla \Psi(x_{t+1}) - \nabla \Psi(x_t), \ x - x_{t+1} \rangle \ge 0. \tag{168}$$

Setting $x = u$ yields

$$\eta_t \langle g_t, x_{t+1} - u \rangle \le \langle \nabla \Psi(x_{t+1}) - \nabla \Psi(x_t), \ u - x_{t+1} \rangle. \tag{169}$$

Apply the three-point identity (Appendix A.11.2) with $(a, b, c) = (u, x_{t+1}, x_t)$:

$$\langle \nabla \Psi(x_{t+1}) - \nabla \Psi(x_t), \ u - x_{t+1} \rangle D_\Psi(u, x_t) - D_\Psi(u, x_{t+1}) - D_\Psi(x_{t+1}, x_t). \tag{170}$$

Combine (C.4) and (C.5):

$$\eta_t \langle g_t, x_{t+1} - u \rangle \le D_\Psi(u, x_t) - D_\Psi(u, x_{t+1}) - D_\Psi(x_{t+1}, x_t). \tag{171}$$

Now decompose

$$\langle g_t, x_t - u \rangle \langle g_t, x_t - x_{t+1} \rangle + \langle g_t, x_{t+1} - u \rangle, \tag{172}$$

multiply by $\eta_t$, and use (C.6):

$$\eta_t \langle g_t, x_t - u \rangle \le \eta_t \langle g_t, x_t - x_{t+1} \rangle D_\Psi(u, x_t) - D_\Psi(u, x_{t+1}) - D_\Psi(x_{t+1}, x_t). \tag{173}$$

By Hölder and Young,

$$\eta_t \langle g_t, x_t - x_{t+1} \rangle \le \eta_t |g_t|_* |x_t - x_{t+1}| \le \frac{\eta_t^2}{2} |g_t|_*^2 + \frac{1}{2} |x_t - x_{t+1}|^2. \tag{174}$$

Strong convexity implies $D_\Psi(x_{t+1}, x_t) \ge \frac{1}{2}|x_{t+1} - x_t|^2$, so the last two terms in (C.8) are non-positive after inserting (C.9) and can be dropped. Dividing by $\eta_t$ gives (C.2). $\qquad\square$

### A.13.2. TELESCOPING WITH DYNAMIC COMPARATORS AND THE DRIFT TERM

Let $u_{t_{t=1}}^T \subseteq \Pi$ be an arbitrary comparator sequence. Apply Lemma A.11 with $u = u_t$ and sum over $t$:

$$\sum_{t=1}^T \langle g_t, x_t - u_t \rangle \le \sum_{t=1}^T \frac{D_\Psi(u_t, x_t) - D_\Psi(u_t, x_{t+1})}{\eta_t} + \sum_{t=1}^T \frac{\eta_t}{2} |g_t|_*^2. \tag{175}$$

The first sum is not a perfect telescope because $u_t$ changes with $t$. The following exact algebraic decomposition identifies all residuals:

$$\sum_{t=1}^T \frac{D_\Psi(u_t, x_t) - D_\Psi(u_t, x_{t+1})}{\eta_t} =$$

$$\frac{D_\Psi(u_1, x_1)}{\eta_1} + \sum_{t=2}^T D_\Psi(u_t, x_t)\left(\frac{1}{\eta_t} - \frac{1}{\eta_{t-1}}\right) + \sum_{t=2}^T \frac{D_\Psi(u_t, x_t) - D_\Psi(u_{t-1}, x_t)}{\eta_{t-1}} - \frac{D_\Psi(u_T, x_{T+1})}{\eta_T}. \tag{176}$$

This is obtained by re-indexing the second term and adding/subtracting $D_\Psi(u_{t-1}, x_t)$ inside the bracket; no inequality is used.

**Step-size monotonicity removes the step-size-change residual.** If $\eta_t$ is nondecreasing (equivalently $1/\eta_t$ is nonincreasing), then

$$\frac{1}{\eta_t} - \frac{1}{\eta_{t-1}} \le 0, \tag{177}$$

and since $D_\Psi \ge 0$, the second term in (C.11) is non-positive and can be dropped. This is the sole reason to enforce nondecreasing $\eta_t$, typically via doubling/epoch constructions.

**A clean drift control under bounded mirror gradients.** To control the comparator-change residual, assume:

**Assumption A.12** (Bounded mirror gradient). There exists $G_\Psi < \infty$ such that $|\nabla\Psi(x)|_* \le G_\Psi$ for all $x \in \Pi$.

Under Assumption A.12,

$$D_\Psi(u_t, x) - D_\Psi(u_{t-1}, x)\Psi(u_t) - \Psi(u_{t-1}) - \langle \nabla\Psi(x), u_t - u_{t-1} \rangle \le \langle \nabla\Psi(u_t) - \nabla\Psi(x), u_t - u_{t-1} \rangle, \tag{178}$$

and therefore, by convexity of $\Psi$ and Hölder,

$$D_\Psi(u_t, x) - D_\Psi(u_{t-1}, x) \le |\nabla\Psi(x)|_*, |u_t - u_{t-1}| \le G_\Psi |u_t - u_{t-1}|. \tag{179}$$

For the negative entropy on $\Delta_{K,\varepsilon}$ with $|\cdot| = |\cdot|_1$ and $|\cdot|_* = |\cdot|_\infty$, Appendix A.15 gives $G_\Psi \le 1 + |\log \varepsilon|$.

A.13.3. SUMMED DYNAMIC INEQUALITY (ROBUST FORM)

Combining (C.10)–(C.12), dropping non-positive terms (including $D_\Psi(u_T, x_{T+1})/\eta_T$), and specializing to $|\cdot| = |\cdot|_1$ gives:

$$\sum_{t=1}^T \langle g_t, x_t - u_t \rangle \le \frac{D_\Psi(u_1, x_1)}{\eta_1} + \sum_{t=1}^T \frac{\eta_t}{2}|g_t|_\infty^2 + \sum_{t=2}^T \frac{G_\Psi}{\eta_{t-1}}|u_t - u_{t-1}|_1, \tag{180}$$

provided $\eta_t$ is nondecreasing and Assumption A.12 holds.

**A.14. Proof of Lemma 3.1**

This appendix presents the specialized proof of Lemma 3.1 used in the main text, following a fully explicit algebraic decomposition and relying only on two minimal conditions: (i) nondecreasing step sizes $\eta_t$ (equivalently nonincreasing $1/\eta_t$); and (ii) bounded mirror gradients $|\nabla\Psi(x)|_\infty \le G_\Psi$ on the feasible set.

A.14.1. STATIC ONE-STEP INEQUALITY

Consider the mirror descent update (main text Eq. (3)):

$$x_{t+1} = \arg\min_{x \in \Pi} \left\{ \eta_t \langle g_t, x \rangle + D_\Psi(x, x_t) \right\}, \tag{181}$$

where $\Pi$ is closed and convex, $\Psi$ is differentiable and 1-strongly convex w.r.t. $|\cdot|_1$, and the dual norm is $|\cdot|_\infty$.

**Lemma A.13** (Static MD inequality). *For any fixed $u \in \Pi$, the iterate pair $(x_t, x_{t+1})$ generated by (D.1) satisfies*

$$\langle g_t, x_t - u \rangle \le \frac{D_\Psi(u, x_t) - D_\Psi(u, x_{t+1})}{\eta_t} + \frac{\eta_t}{2}|g_t|_\infty^2. \tag{182}$$

*Proof.* This is Lemma A.11 instantiated with $|\cdot| = |\cdot|_1$ and $|\cdot|_* = |\cdot|_\infty$. $\qquad\square$

A.14.2. SUMMATION WITH DYNAMIC COMPARATORS: EXACT DECOMPOSITION

Let $u_t{}_{t=1}^T \subseteq \Pi$ be any comparator sequence. Apply Lemma A.13 with $u = u_t$ and sum:

$$\sum_{t=1}^T \langle g_t, x_t - u_t \rangle \le \sum_{t=1}^T \frac{D_\Psi(u_t, x_t) - D_\Psi(u_t, x_{t+1})}{\eta_t} + \sum_{t=1}^T \frac{\eta_t}{2}|g_t|_\infty^2. \tag{183}$$

Define

$$S := \sum_{t=1}^{T} \frac{D_\Psi(u_t, x_t) - D_\Psi(u_t, x_{t+1})}{\eta_t}. \tag{184}$$

Write $S$ as a difference of two sums and re-index the second:

$$S = \sum_{t=1}^{T} \frac{D_\Psi(u_t, x_t)}{\eta_t} - \sum_{t=1}^{T} \frac{D_\Psi(u_t, x_{t+1})}{\eta_t} \quad = \frac{D_\Psi(u_1, x_1)}{\eta_1} + \sum_{t=2}^{T} \frac{D_\Psi(u_t, x_t)}{\eta_t} - \sum_{t=2}^{T} \frac{D_\Psi(u_{t-1}, x_t)}{\eta_{t-1}} - \frac{D_\Psi(u_T, x_{T+1})}{\eta_T}.$$

Insert and subtract $\frac{D_\Psi(u_t, x_t)}{\eta_{t-1}}$ inside the middle bracket:

$$\sum_{t=2}^{T} \frac{D_\Psi(u_t, x_t)}{\eta_t} - \sum_{t=2}^{T} \frac{D_\Psi(u_{t-1}, x_t)}{\eta_{t-1}} = \sum_{t=2}^{T} D_\Psi(u_t, x_t)\Big(\frac{1}{\eta_t} - \frac{1}{\eta_{t-1}}\Big) \quad + \sum_{t=2}^{T} \frac{D_\Psi(u_t, x_t) - D_\Psi(u_{t-1}, x_t)}{\eta_{t-1}}.$$

Up to this point, all steps are identities.

### A.14.3. CONTROLLING RESIDUALS UNDER MINIMAL ASSUMPTIONS

**Step-size monotonicity eliminates the step-size-change term.** If $\eta_t$ is nondecreasing, then $(1/\eta_t - 1/\eta_{t-1}) \le 0$, and since $D_\Psi \ge 0$,

$$\sum_{t=2}^{T} D_\Psi(u_t, x_t)\Big(\frac{1}{\eta_t} - \frac{1}{\eta_{t-1}}\Big) \le 0. \tag{185}$$

Hence it can be dropped in an upper bound.

**Bounding the comparator-drift term by a path variation.** For any fixed $x \in \Pi$,

$$D_\Psi(u_t, x) - D_\Psi(u_{t-1}, x) = \Psi(u_t) - \Psi(u_{t-1}) - \langle \nabla\Psi(x), u_t - u_{t-1} \rangle$$
$$\le \langle \nabla\Psi(u_t) - \nabla\Psi(x), u_t - u_{t-1} \rangle \le |\nabla\Psi(x)|_\infty, |u_t - u_{t-1}|_1. \tag{186}$$

Assuming $|\nabla\Psi(x)|_\infty \le G_\Psi$ for all $x \in \Pi$ (guaranteed by truncation; Appendix A.15), it follows that

$$D_\Psi(u_t, x) - D_\Psi(u_{t-1}, x) \le G_\Psi |u_t - u_{t-1}|_1. \tag{187}$$

### A.14.4. CONCLUSION: LEMMA 3.1

Combining (D.3)–(D.8), dropping the non-positive term (D.6) and the terminal nonnegative term $D_\Psi(u_T, x_{T+1})/\eta_T$, yields

$$\sum_{t=1}^{T} \langle g_t, x_t - u_t \rangle \le \frac{D_\Psi(u_1, x_1)}{\eta_1} + \sum_{t=1}^{T} \frac{\eta_t}{2}|g_t|_\infty^2 + \sum_{t=2}^{T} \frac{G_\Psi}{\eta_{t-1}}|u_t - u_{t-1}|_1, \tag{188}$$

which is the stated dynamic-comparator mirror-descent inequality used as Lemma 3.1 in the main text.

### A.15. Entropy Bounds on the Truncated Simplex

This appendix records two basic facts used to track constants: (i) $|\Psi(x)| \le \log K$ on $\Delta_K$; and (ii) $|\nabla\Psi(x)|_\infty \le 1 + |\log\varepsilon|$ on $\Delta_{K,\varepsilon}$.

### A.15.1. BOUNDEDNESS: $|\Psi(x)| \le \log K$

Recall $\Psi(x) = \sum_{i=1}^{K} x_i \log x_i$ on $\Delta_K$.

**Proposition A.14.** *For any $x \in \Delta_K$,*

$$-\log K \le \Psi(x) \le 0, \qquad hence \quad |\Psi(x)| \le \log K. \tag{189}$$

*Proof.* Since $x_i \in [0, 1]$, $\log x_i \le 0$ and thus $x_i \log x_i \le 0$, giving $\Psi(x) \le 0$. For the lower bound, $\phi(z) = z \log z$ is convex on $z > 0$. Under $\sum_i x_i = 1$, Jensen (or KKT) shows the minimum is attained at $x_i = 1/K$, giving $\Psi(\mathbf{1}/K) = -\log K$. $\quad\square$

A direct corollary used in the main text is: for any $\lambda \ge 0$ and $x, u \in \Delta_K$,

$$-\lambda\big(\Psi(x) - \Psi(u)\big) \le \lambda|\Psi(x)| + \lambda|\Psi(u)| \le 2\lambda \log K. \tag{190}$$

### A.15.2. GRADIENT BOUND ON $\Delta_{K,\varepsilon}$

Consider $\Delta_{K,\varepsilon} = x \in \Delta_K : x_i \ge \varepsilon, \ \forall i$ with $\varepsilon \in (0, 1/K]$. For $x \in \Delta_{K,\varepsilon}$,

$$\frac{\partial \Psi(x)}{\partial x_i} = 1 + \log x_i. \tag{191}$$

**Proposition A.15** (Gradient $\ell_\infty$ bound). *For any $x \in \Delta_{K,\varepsilon}$,*

$$|\nabla\Psi(x)|_\infty = \max_{i\in[K]} |1 + \log x_i| \le 1 + |\log \varepsilon|. \tag{192}$$

*Proof.* Since $x_i \ge \varepsilon$, $\log x_i \ge \log \varepsilon$; since $x_i \le 1$, $\log x_i \le 0$. Hence $|\log x_i| \le |\log \varepsilon|$ and therefore $|1 + \log x_i| \le 1 + |\log \varepsilon|$ for all $i$. $\quad\square$

Define

$$G_\Psi := \sup_{x \in \Delta_{K,\varepsilon}} |\nabla\Psi(x)|_\infty \le 1 + |\log \varepsilon|. \tag{193}$$

Choosing $\varepsilon = T^{-c}$ implies $|\log \varepsilon| = O(\log T)$, so truncation affects only logarithmic factors.

## A.16. Proof Details for Theorem 3.2

This appendix reorganizes the intermediate bound (main text Eq. (12)) into the per-round trade-off form (main text Eq. (16)), and makes the constants $C_0, C_1, C_2$ explicit.

### A.16.1. STARTING POINT

Recall the intermediate bound: for any comparator sequence $u_{1:T} \subset \Pi$,

$$\text{Reg}_T^{\text{dyn}}(u_{1:T}) \le \frac{D_\Psi(u_1, x_1)}{\eta_1} + \sum_{t=1}^{T} \frac{\eta_t}{2}|\nabla\ell_t(x_t)|_\infty^2 + \sum_{t=2}^{T} \frac{2G_\Psi}{\eta_t}|u_t - u_{t-1}|_1 + 2\log K \sum_{t=1}^{T} \lambda_t, \tag{194}$$

where $\ell_t(x) = f_t(x) + \lambda_t \Psi(x)$, and $|\nabla f_t(x)|_\infty \le G$.

The target form is

$$\text{Reg}_T^{\text{dyn}}(u_{1:T}) \le C_0 + \sum_{t=2}^{T}\Big(C_1 \frac{\xi_t}{\lambda_t} + C_2\lambda_t\Big), \qquad \xi_t := |u_t - u_{t-1}|_1. \tag{195}$$

### A.16.2. COUPLING $\eta_t = c\lambda_t$ YIELDS $\xi_t/\lambda_t$

Let

$$\eta_t = c\lambda_t, \qquad c > 0. \tag{196}$$

Then the drift term in (F.1) becomes

$$\sum_{t=2}^{T} \frac{2G_\Psi}{\eta_t}|u_t - u_{t-1}|_1 \sum_{t=2}^{T} \frac{2G_\Psi}{c}\frac{\alpha_t}{\lambda_t}, \tag{197}$$

so one may take

$$C_1 := \frac{2G_\Psi}{c}. \tag{198}$$

A.16.3. LINEARIZING THE GRADIENT TERM VIA CLIPPING

Since $\nabla \ell_t = \nabla f_t + \lambda_t \nabla \Psi$, the triangle inequality gives

$$|\nabla \ell_t(x_t)|_\infty \leq |\nabla f_t(x_t)|_\infty + \lambda_t |\nabla \Psi(x_t)|_\infty \leq G + \lambda_t G_\Psi. \tag{199}$$

Substitute (F.6) into (F.1) and use (F.3):

$$\sum_{t=1}^{T} \frac{\eta_t}{2} |\nabla \ell_t(x_t)|_\infty^2 \leq \sum_{t=1}^{T} \frac{c\lambda_t}{2} (G + \lambda_t G_\Psi)^2. \tag{200}$$

Assume a standard upper clipping:

$$0 < \lambda_t \leq \lambda_{\max}, \qquad \forall t. \tag{201}$$

Then for all $t$,

$$(G + \lambda_t G_\Psi)^2 \leq (G + \lambda_{\max} G_\Psi)^2 =: M^2. \tag{202}$$

Therefore,

$$\sum_{t=1}^{T} \frac{\eta_t}{2} |\nabla \ell_t(x_t)|_\infty^2 \leq \sum_{t=1}^{T} \frac{c\lambda_t}{2} M^2 \frac{cM^2}{2} \sum_{t=1}^{T} \lambda_t. \tag{203}$$

A.16.4. INITIAL TERM AND A LOWER CLIPPING $\lambda_{\min}$

The initial term in (F.1) is $D_\Psi(u_1, x_1)/\eta_1$. For negative entropy, if $x_1 = \mathbf{1}/K$, then for any $u_1 \in \Delta_K$,

$$D_\Psi(u_1, x_1) = \mathrm{KL}(u_1|\mathbf{1}/K) \leq \log K. \tag{204}$$

If a lower clipping is also enforced,

$$\lambda_t \geq \lambda_{\min} > 0, \qquad \forall t, \tag{205}$$

then $\eta_1 = c\lambda_1 \geq c\lambda_{\min}$ and hence

$$\frac{D_\Psi(u_1, x_1)}{\eta_1} \leq \frac{\log K}{c\lambda_{\min}}. \tag{206}$$

A.16.5. FINAL CONSTANTS

Combine (F.4), (F.10), and the entropy compensation term $2\log K \sum_{t=1}^{T} \lambda_t$ in (F.1), and use (F.13) for the initial term:

$$\mathrm{Reg}_T^{\mathrm{dyn}}(u_{1:T}) \leq \underbrace{\frac{\log K}{c\lambda_{\min}}}_{=:C_0} + \sum_{t=2}^{T} \underbrace{\frac{2G_\Psi}{c}}_{=:C_1} \frac{\alpha_t}{\lambda_t} + \underbrace{\left(\frac{c}{2}(G + \lambda_{\max} G_\Psi)^2 + 2\log K\right)}_{=:C_2} \sum_{t=1}^{T} \lambda_t. \tag{207}$$

Absorbing the $t = 1$ term of $\sum_{t=1}^{T} \lambda_t$ into $C_0$ yields the desired per-round form (F.2). In particular, a concrete choice is:

$$\boxed{C_1 = \frac{2G_\Psi}{c}, \qquad C_2 = \frac{c}{2}(G + \lambda_{\max} G_\Psi)^2 + 2\log K, \qquad C_0 = \frac{\log K}{c\lambda_{\min}} + C_2\lambda_1.} \tag{208}$$

For negative entropy on $\Delta_{K,\varepsilon}$, Appendix A.15 gives $G_\Psi \leq 1 + |\log \varepsilon|$.

## A.17. Optimal Offline Constant Choice of $\lambda$

Starting from the trade-off bound

$$\mathrm{Reg}_T^{\mathrm{dyn}}(u_{1:T}) \leq C_0 + \sum_{t=2}^{T} \left(C_1 \frac{\alpha_t}{\lambda_t} + C_2\lambda_t\right), \qquad \alpha_t := |u_t - u_{t-1}|_1, \tag{209}$$

this appendix derives the best offline constant choice $\lambda_t \equiv \lambda$, proves the offline constant minimizer (Section 3), and relates it to the per-round oracle minimizer.

A.17.1. REDUCTION UNDER $\lambda_t \equiv \lambda$

If $\lambda_t \equiv \lambda$,

$$\sum_{t=2}^{T} \frac{\alpha_t}{\lambda_t} \frac{1}{\lambda} \sum_{t=2}^{T} \alpha_t \frac{A_T}{\lambda}, \qquad \sum_{t=2}^{T} \lambda_t = (T-1)\lambda, \tag{210}$$

where $A_T := \sum_{t=2}^{T} \alpha_t$. Plugging into (G.1) yields

$$\text{Reg}_T^{\text{dyn}}(u_{1:T}) \leq C_0 + C_1 \frac{A_T}{\lambda} + C_2(T-1)\lambda \lesssim C_0 + C_1 \frac{A_T}{\lambda} + C_2 T\lambda. \tag{211}$$

A.17.2. ONE-DIMENSIONAL MINIMIZATION AND (3.3)

Consider $\phi(\lambda) = C_1 A_T/\lambda + C_2 T\lambda$ for $\lambda > 0$. Then

$$\phi'(\lambda) = -C_1 \frac{A_T}{\lambda^2} + C_2 T, \tag{212}$$

so the minimizer is

$$\lambda^{\text{off}} = \sqrt{\frac{C_1}{C_2} \cdot \frac{A_T}{T}}. \tag{213}$$

Substituting gives

$$\phi(\lambda^{\text{off}}) = 2\sqrt{C_1 C_2, A_T T}, \tag{214}$$

and therefore

$$\text{Reg}_T^{\text{dyn}}(u_{1:T}) \leq C_0 + 2\sqrt{C_1 C_2, A_T T}. \tag{215}$$

A.17.3. RELATION TO THE PER-ROUND ORACLE (THEOREM 3.3)

The per-round oracle minimizes $C_1 \alpha_t/\lambda + C_2 \lambda$ and yields

$$\lambda_t^{\star} = \sqrt{\frac{C_1}{C_2} \alpha_t}. \tag{216}$$

Its induced bound scales with $\sum_t \sqrt{\alpha_t}$, and Cauchy–Schwarz implies

$$\sum_{t=2}^{T} \sqrt{\alpha_t} \leq \sqrt{(T-1)\sum_{t=2}^{T} \alpha_t} \lesssim \sqrt{A_T T}. \tag{217}$$

Hence both $\lambda^{\text{off}}$ and $\lambda_t^{\star}$ achieve the same principal order $\sqrt{A_T T}$, with $\lambda_t^{\star}$ adapting to non-uniform variation patterns.

## A.18. Two Summation Inequalities for Theorem 3.5

Let $\widehat{\xi}_1^T$ be a nonnegative sequence (an online proxy of nonstationarity), and define prefix sums

$$\widehat{A}_t := \sum_{i=1}^{t} \widehat{\xi}_i, \qquad \widehat{A}_0 := 0. \tag{218}$$

To avoid division by zero, adopt the convention: if $\widehat{A}_t = 0$, then $\widehat{\xi}_t = 0$, and any expression $\widehat{\xi}_t/\sqrt{\widehat{A}_t}$ is interpreted as $0$.

A.18.1. SQUARE-ROOT POTENTIAL TRICK

**Lemma A.16.** *For any nonnegative sequence $\widehat{\xi}_t$ and $\widehat{A}_t$ as in (H.1),*

$$\sum_{t=1}^{T} \frac{\widehat{\xi}_t}{\sqrt{\widehat{A}_t}} \leq 2\sqrt{\widehat{A}_T}. \tag{219}$$

*Proof.* For any $t$ with $\widehat{A}_t > 0$,

$$\sqrt{\widehat{A}_t} - \sqrt{\widehat{A}_{t-1}} \frac{\widehat{A}_t - \widehat{A}_{t-1}}{\sqrt{\widehat{A}_t} + \sqrt{\widehat{A}_{t-1}}} \frac{\widehat{\xi}_t}{\sqrt{\widehat{A}_t} + \sqrt{\widehat{A}_{t-1}}} \geq \frac{\widehat{\xi}_t}{2\sqrt{\widehat{A}_t}}. \tag{220}$$

Rearranging gives $\widehat{\xi}_t/\sqrt{\widehat{A}_t} \leq 2(\sqrt{\widehat{A}_t} - \sqrt{\widehat{A}_{t-1}})$. Summing telescopes to $2\sqrt{\widehat{A}_T}$. □

### A.18.2. BOUNDING $\sum_t \sqrt{\widehat{A}_t/t}$

**Lemma A.17.** *For any nonnegative sequence $\widehat{\xi}_t$ and $\widehat{A}_t$ as in (H.1),*

$$\sum_{t=1}^{T} \sqrt{\frac{\widehat{A}_t}{t}} \leq 2\sqrt{T\widehat{A}_T}. \tag{221}$$

*Proof.* Since $\widehat{A}_t$ is nondecreasing, $\widehat{A}_t \leq \widehat{A}_T$ for all $t$, hence

$$\sum_{t=1}^{T} \sqrt{\frac{\widehat{A}_t}{t}} \leq \sqrt{\widehat{A}_T} \sum_{t=1}^{T} \frac{1}{\sqrt{t}}. \tag{222}$$

Using the integral comparison $\sum_{t=1}^{T} t^{-1/2} \leq 1 + \int_1^T x^{-1/2}, dx = 1 + 2(\sqrt{T} - 1) \leq 2\sqrt{T}$ yields (H.3). □

### A.19. Effect of Clipping the Schedule

#### A.19.1. CLIPPING INTRODUCES ONLY CONTROLLABLE ADDITIVE TERMS

Suppose an online scheduler produces a "raw" temperature $\lambda_t^{\mathrm{raw}} > 0$, and the implementation clips it to

$$\lambda_t := \Pi_{[\lambda_{\min}, \lambda_{\max}]}(\lambda_t^{\mathrm{raw}}) \min \lambda_{\max}, \max \lambda_{\min}, \lambda_t^{\mathrm{raw}}. \tag{223}$$

This appendix shows that replacing $\lambda_t^{\mathrm{raw}}$ by $\lambda_t$ preserves the main trade-off bound, up to an additive endpoint compensation term.

**Pointwise inequalities** Since $\widehat{\xi}_t \geq 0$, the map $\lambda \mapsto \widehat{\xi}_t/\lambda$ is decreasing on $\lambda > 0$, while $\lambda \mapsto \lambda$ is increasing. The following two pointwise bounds suffice:

**Lemma A.18.** *For any $t$,*

$$\frac{\widehat{\xi}_t}{\lambda_t} \leq \frac{\widehat{\xi}_t}{\lambda_t^{\mathrm{raw}}} + \frac{\widehat{\xi}_t}{\lambda_{\max}}, \tag{224}$$

*and*

$$\lambda_t \leq \lambda_t^{\mathrm{raw}} + \lambda_{\min}. \tag{225}$$

*Proof.* If $\lambda_t^{\mathrm{raw}} \leq \lambda_{\max}$, then $\lambda_t \geq \lambda_t^{\mathrm{raw}}$, hence $\widehat{\xi}_t/\lambda_t \leq \widehat{\xi}_t/\lambda_t^{\mathrm{raw}}$. If $\lambda_t^{\mathrm{raw}} > \lambda_{\max}$, then $\lambda_t = \lambda_{\max}$ and $\widehat{\xi}_t/\lambda_t = \widehat{\xi}_t/\lambda_{\max} \leq \widehat{\xi}_t/\lambda_t^{\mathrm{raw}} + \widehat{\xi}_t/\lambda_{\max}$.

If $\lambda_t^{\mathrm{raw}} \geq \lambda_{\min}$, then $\lambda_t \leq \lambda_t^{\mathrm{raw}}$, so $\lambda_t \leq \lambda_t^{\mathrm{raw}} + \lambda_{\min}$. If $\lambda_t^{\mathrm{raw}} < \lambda_{\min}$, then $\lambda_t = \lambda_{\min} \leq \lambda_t^{\mathrm{raw}} + \lambda_{\min}$. □

**Summing yields the compensation term** Multiply 224- 225 by constants $C_1, C_2$ and sum:

$$\sum_{t=1}^{T} C_1 \frac{\widehat{\xi}_t}{\lambda_t} \leq \sum_{t=1}^{T} C_1 \frac{\widehat{\xi}_t}{\lambda_t^{\mathrm{raw}}} + \sum_{t=1}^{T} C_1 \frac{\widehat{\xi}_t}{\lambda_{\max}} \sum_{t=1}^{T} C_1 \frac{\widehat{\xi}_t}{\lambda_t^{\mathrm{raw}}} + \frac{C_1 \widehat{A}_T}{\lambda_{\max}}, \tag{226}$$

$$beginequation4mm] \sum_{t=1}^{T} C_2 \lambda_t \leq \sum_{t=1}^{T} C_2 \lambda_t^{\mathrm{raw}} + \sum_{t=1}^{T} C_2 \lambda_{\min} \sum_{t=1}^{T} C_2 \lambda_t^{\mathrm{raw}} + C_2 T \lambda_{\min}. \tag{227}$$

Adding gives

$$\sum_{t=1}^{T}\left(C_1\frac{\widehat{\xi}_t}{\lambda_t}+C_2\lambda_t\right)\leq\sum_{t=1}^{T}\left(C_1\frac{\widehat{\xi}_t}{\lambda_t^{\text{raw}}}+C_2\lambda_t^{\text{raw}}\right)+\frac{C_1\widehat{A}_T}{\lambda_{\max}}+C_2T\lambda_{\min}. \tag{228}$$

Thus clipping preserves the main bound and introduces only the controllable additive endpoints $\frac{C_1\widehat{A}_T}{\lambda_{\max}}+C_2T\lambda_{\min}$.

If the main bound includes an initial term such as $D_\Psi(u_1,x_1)/\eta_1$, then $\lambda_1\geq\lambda_{\min}$ (hence $\eta_1=c\lambda_1\geq c\lambda_{\min}$) implies this initial term is also bounded by a constant, e.g., $\log K/(c\lambda_{\min})$ as in Appendix A.16.

### A.19.2. WHY CLIPPING IS NECESSARY IN RL IMPLEMENTATIONS (STABILITY PRINCIPLES)

In continuous-control off-policy algorithms, the proxy $\widehat{\xi}_t$ (e.g., critic drift, TD-residual-based surrogates) is noisy and can exhibit: (i) **spikes**, which would drive $\lambda_t$ excessively large and destabilize both the actor entropy regularization and the critic target scaling; and (ii) **near-zero plateaus**, which would collapse $\lambda_t$ toward 0, making the policy nearly deterministic, harming exploration and adaptation when the environment changes.

Therefore, clipping is not an ad hoc heuristic: it is a stabilization step that turns an idealized scheduler into a numerically trainable system. The theory above formalizes that clipping changes only constants and adds a transparent endpoint compensation term.

### A.20. Proof of Eq. (26)

This appendix establishes a Lipschitz-type sensitivity bound for the soft-optimal action-value function under one-step MDP perturbations. Consider the discounted soft MDP at round $t$,

$$\mathcal{M}_t:=(\mathcal{S},\mathcal{A},P_t,r_t,\gamma),\qquad\gamma\in(0,1), \tag{229}$$

with entropy weight $\mu>0$. For notational simplicity, $\mathcal{A}$ is assumed finite in this appendix.

### A.20.1. SOFT BELLMAN OPTIMALITY OPERATOR (LOG-SUM-EXP FORM)

For any $Q:\mathcal{S}\times\mathcal{A}\to\mathbb{R}$, define the induced soft state-value

$$V_Q(s):=\mu\log\sum_{a\in\mathcal{A}}\exp\left(\frac{Q(s,a)}{\mu}\right). \tag{230}$$

The soft Bellman optimality operator at round $t$ is

$$(\mathcal{T}_tQ)(s,a):=r_t(s,a)+\gamma\,\mathbb{E}_{s'\sim P_t(\cdot|s,a)}\big[V_Q(s')\big],\qquad\forall(s,a)\in\mathcal{S}\times\mathcal{A}. \tag{231}$$

The soft-optimal action-value $Q_t^\star$ is the unique fixed point:

$$Q_t^\star=\mathcal{T}_tQ_t^\star. \tag{232}$$

### A.20.2. A KEY LEMMA: LOG-SUM-EXP IS 1-LIPSCHITZ IN $\|\cdot\|_\infty$

**Lemma A.19** (Log-sum-exp is 1-Lipschitz under $\|\cdot\|_\infty$). *For any $x,y\in\mathbb{R}^{|\mathcal{A}|}$ and any $\mu>0$,*

$$\left|\mu\log\sum_a e^{x_a/\mu}-\mu\log\sum_a e^{y_a/\mu}\right|\leq\|x-y\|_\infty. \tag{233}$$

*Consequently, for any $Q,Q'$ and any $s\in\mathcal{S}$,*

$$|V_Q(s)-V_{Q'}(s)|\leq\|Q(s,\cdot)-Q'(s,\cdot)\|_\infty\leq\|Q-Q'\|_\infty. \tag{234}$$

*Proof.* Let $g(z):=\mu\log\sum_a e^{z_a/\mu}$. Then $\nabla g(z)=\text{softmax}(z/\mu)$ has nonnegative coordinates summing to 1, hence $\|\nabla g(z)\|_1=1$. By the mean value theorem, $g(x)-g(y)=\langle\nabla g(\xi),x-y\rangle$ for some $\xi$ on the segment between $x$ and $y$. Hölder's inequality yields $|g(x)-g(y)|\leq\|\nabla g(\xi)\|_1\|x-y\|_\infty=\|x-y\|_\infty$. Applying this pointwise to $x=Q(s,\cdot)$ and $y=Q'(s,\cdot)$ gives (234). $\qquad\square$

A.20.3. $\mathcal{T}_t$ IS A $\gamma$-CONTRACTION IN $\|\cdot\|_\infty$

**Lemma A.20** ($\gamma$-contraction). *For any $t$ and any $Q, Q'$,*

$$\|\mathcal{T}_t Q - \mathcal{T}_t Q'\|_\infty \leq \gamma \|Q - Q'\|_\infty. \tag{235}$$

*Proof.* Fix $(s, a)$. By (231) and (234),

$$|(\mathcal{T}_t Q)(s, a) - (\mathcal{T}_t Q')(s, a)| = \gamma \left| \mathbb{E}_{P_t(\cdot|s,a)}[V_Q(s') - V_{Q'}(s')] \right| \leq \gamma \|Q - Q'\|_\infty. \tag{236}$$

Taking the supremum over $(s, a)$ proves (235). $\qquad\square$

A.20.4. BOUNDING $\|\mathcal{T}_t Q - \mathcal{T}_{t-1} Q\|_\infty$ VIA REWARD AND TRANSITION DRIFT

Define one-step MDP drift quantities

$$\Delta_t^r := \|r_t - r_{t-1}\|_\infty = \sup_{s,a} |r_t(s, a) - r_{t-1}(s, a)|, \tag{237}$$

$$\Delta_t^P := \sup_{s,a} \|P_t(\cdot \mid s, a) - P_{t-1}(\cdot \mid s, a)\|_1. \tag{238}$$

Let $V_{\max}(Q) := \|V_Q\|_\infty = \sup_s |V_Q(s)|$.

**Lemma A.21** (Operator drift bound). *For any $Q$,*

$$\|\mathcal{T}_t Q - \mathcal{T}_{t-1} Q\|_\infty \leq \Delta_t^r + \gamma\, V_{\max}(Q)\, \Delta_t^P. \tag{239}$$

*Proof.* Fix $(s, a)$. Using (231),

$$|(\mathcal{T}_t Q)(s, a) - (\mathcal{T}_{t-1} Q)(s, a)| \leq |r_t(s, a) - r_{t-1}(s, a)| + \gamma \left| \mathbb{E}_{P_t}[V_Q] - \mathbb{E}_{P_{t-1}}[V_Q] \right|. \tag{240}$$

The first term is bounded by $\Delta_t^r$. For the second term, the standard inequality for bounded $f$ and distributions $p, q$, $|\mathbb{E}_p[f] - \mathbb{E}_q[f]| \leq \|f\|_\infty \|p - q\|_1$, implies

$$\left| \mathbb{E}_{P_t(\cdot|s,a)}[V_Q] - \mathbb{E}_{P_{t-1}(\cdot|s,a)}[V_Q] \right| \leq \|V_Q\|_\infty \|P_t(\cdot \mid s, a) - P_{t-1}(\cdot \mid s, a)\|_1 \leq V_{\max}(Q)\Delta_t^P. \tag{241}$$

Taking the supremum over $(s, a)$ yields (239). $\qquad\square$

A.20.5. FIXED-POINT PERTURBATION FOR CONTRACTIONS

**Lemma A.22** (Fixed-point sensitivity for $\gamma$-contractions). *Let $\mathcal{T}$ and $\mathcal{S}$ be $\gamma$-contractions under $\|\cdot\|_\infty$ with unique fixed points $Q^{\mathcal{T}} = \mathcal{T}Q^{\mathcal{T}}$ and $Q^{\mathcal{S}} = \mathcal{S}Q^{\mathcal{S}}$. Then*

$$\|Q^{\mathcal{T}} - Q^{\mathcal{S}}\|_\infty \leq \frac{1}{1 - \gamma} \|\mathcal{T}Q^{\mathcal{S}} - \mathcal{S}Q^{\mathcal{S}}\|_\infty. \tag{242}$$

*Proof.*

$$\|Q^{\mathcal{T}} - Q^{\mathcal{S}}\|_\infty = \|\mathcal{T}Q^{\mathcal{T}} - \mathcal{S}Q^{\mathcal{S}}\|_\infty \leq \|\mathcal{T}Q^{\mathcal{T}} - \mathcal{T}Q^{\mathcal{S}}\|_\infty + \|\mathcal{T}Q^{\mathcal{S}} - \mathcal{S}Q^{\mathcal{S}}\|_\infty \leq \gamma \|Q^{\mathcal{T}} - Q^{\mathcal{S}}\|_\infty + \|\mathcal{T}Q^{\mathcal{S}} - \mathcal{S}Q^{\mathcal{S}}\|_\infty. \tag{243}$$

Rearranging gives (242). $\qquad\square$

A.20.6. SENSITIVITY OF $Q_t^\star$ (PROOF OF EQ. (26))

Applying Lemma A.22 with $\mathcal{T} = \mathcal{T}_t$, $\mathcal{S} = \mathcal{T}_{t-1}$, and $Q^{\mathcal{S}} = Q_{t-1}^\star$ yields

$$\|Q_t^\star - Q_{t-1}^\star\|_\infty \leq \frac{1}{1-\gamma} \|\mathcal{T}_t Q_{t-1}^\star - \mathcal{T}_{t-1} Q_{t-1}^\star\|_\infty. \tag{244}$$

Then Lemma A.21 (with $Q = Q_{t-1}^\star$) implies

$$\|\mathcal{T}_t Q_{t-1}^\star - \mathcal{T}_{t-1} Q_{t-1}^\star\|_\infty \leq \Delta_t^r + \gamma\, V_{\max}(Q_{t-1}^\star)\, \Delta_t^P. \tag{245}$$

Combining (244)–(245) gives the desired reward-plus-transition sensitivity bound:

$$\boxed{\|Q_t^\star - Q_{t-1}^\star\|_\infty \leq \frac{1}{1-\gamma}\Big(\Delta_t^r + \gamma\, V_{\max}(Q_{t-1}^\star)\, \Delta_t^P\Big).} \tag{246}$$

**Optional explicit bound on $V_{\max}(Q_t^\star)$.** If rewards are bounded as $|r_t(s,a)| \leq R_{\max}$, then standard arguments yield

$$\|Q_t^\star\|_\infty \leq \frac{R_{\max} + \gamma\mu \log|\mathcal{A}|}{1-\gamma}, \qquad \|V_{Q_t^\star}\|_\infty \leq \frac{R_{\max} + \mu \log|\mathcal{A}|}{1-\gamma}. \tag{247}$$

Substituting (247) into (246) makes the bound depend only on $\Delta_t^r, \Delta_t^P, \gamma, \mu, |\mathcal{A}|$.

### A.21. Proof of Eq. (27)

This appendix records a standard Lipschitz bound for the Boltzmann/softmax mapping, specialized to the statewise policy map in soft RL.

A.21.1. JACOBIAN AND THE $\ell_\infty \to \ell_1$ OPERATOR NORM

Assume $|\mathcal{A}| = K < \infty$. For $q \in \mathbb{R}^K$, define $\pi(q) = \mathrm{softmax}(q/\mu)$ by

$$\pi(q)_i = \frac{\exp(q_i/\mu)}{\sum_{j=1}^K \exp(q_j/\mu)}, \qquad i \in [K], \tag{248}$$

where $\mu > 0$ is the temperature (entropy weight). A direct differentiation gives

$$\frac{\partial \pi_i(q)}{\partial q_j} = \frac{1}{\mu}\, \pi_i(q)\big(\mathbf{1}\{i=j\} - \pi_j(q)\big), \tag{249}$$

hence

$$\nabla\pi(q) = \frac{1}{\mu}\Big(\mathrm{Diag}(\pi(q)) - \pi(q)\pi(q)^\top\Big). \tag{250}$$

Define the operator norm

$$\|\nabla\pi(q)\|_{\infty \to 1} := \sup_{\|h\|_\infty \leq 1} \|\nabla\pi(q)h\|_1. \tag{251}$$

Let $\pi = \pi(q)$ and consider any $h$ with $\|h\|_\infty \leq 1$. The vector $(\mathrm{Diag}(\pi) - \pi\pi^\top)h$ has coordinates $\pi_i(h_i - \mathbb{E}_\pi[h])$, thus

$$\|(\mathrm{Diag}(\pi) - \pi\pi^\top)h\|_1 = \sum_{i=1}^K \pi_i\, |h_i - \mathbb{E}_\pi[h]| = \mathbb{E}_\pi\big[|H - \mathbb{E}H|\big], \tag{252}$$

where $H$ is the random variable taking value $h_I$ with $I \sim \pi$. Since $H \in [-1,1]$, $\mathrm{Var}(H) \leq 1$, and by Cauchy–Schwarz,

$$\mathbb{E}\big[|H - \mathbb{E}H|\big] \leq \sqrt{\mathbb{E}(H - \mathbb{E}H)^2} = \sqrt{\mathrm{Var}(H)} \leq 1. \tag{253}$$

Combining (250)–(253) yields

$$\|\nabla\pi(q)\|_{\infty \to 1} \leq \frac{1}{\mu}. \tag{254}$$

This bound is tight (e.g., $K = 2$ and $\pi = (1/2, 1/2)$).

A.21.2. GLOBAL LIPSCHITZNESS AND THE STATEWISE FORM

By the mean value theorem applied to the vector map $\pi(\cdot)$ and (254), for any $q, q' \in \mathbb{R}^K$,

$$\|\pi(q) - \pi(q')\|_1 \leq \frac{1}{\mu} \|q - q'\|_\infty. \tag{255}$$

In soft RL, for each state $s$, define $q(s) := Q(s, \cdot) \in \mathbb{R}^K$ and the Boltzmann policy

$$\pi_Q(\cdot \mid s) := \mathrm{softmax}(Q(s, \cdot)/\mu) = \pi(q(s)). \tag{256}$$

Applying (255) statewise yields, for any $s$,

$$\|\pi_Q(\cdot \mid s) - \pi_{Q'}(\cdot \mid s)\|_1 \leq \frac{1}{\mu} \|Q(s, \cdot) - Q'(s, \cdot)\|_\infty \leq \frac{1}{\mu} \|Q - Q'\|_\infty. \tag{257}$$

In particular, taking $Q = Q_t^\star$ and $Q' = Q_{t-1}^\star$ gives the policy-drift bound stated as Eq. (27) in the main text:

$$\boxed{\|\pi_t^\star(\cdot \mid s) - \pi_{t-1}^\star(\cdot \mid s)\|_1 \leq \frac{1}{\mu} \|Q_t^\star - Q_{t-1}^\star\|_\infty.} \tag{258}$$

**Remark (constant $\mu$ across states).** The Lipschitz constant in (257)–(258) equals $1/\mu$. Allowing state-dependent temperatures $\mu(s)$ would introduce non-uniform constants $\max_s 1/\mu(s)$ (or more intricate weighted constants) when aggregating statewise bounds. A single global $\mu$ preserves a clean uniform constant in the $Q^\star \to \pi^\star$ drift transfer.

## A.22. From Squared Drift to Variation

A recurring step in connecting statewise squared policy drifts to a linear MDP-variation budget is to convert $\sum_t \|\Delta Q_t^\star\|_\infty^2$ into $\sum_t \|\Delta Q_t^\star\|_\infty$. This appendix provides an explicit constant $C_Q$ ensuring

$$\sum_{t=2}^{T} \|\Delta Q_t^\star\|_\infty^2 \leq C_Q \sum_{t=2}^{T} \|\Delta Q_t^\star\|_\infty, \qquad \Delta Q_t^\star := Q_t^\star - Q_{t-1}^\star. \tag{259}$$

A.22.1. AN EXPLICIT $C_Q$ FROM BOUNDEDNESS OF $Q_t^\star$

Assume bounded rewards:

$$|r_t(s, a)| \leq R_{\max}, \qquad \forall t, s, a. \tag{260}$$

**Lemma A.23** (Uniform $\ell_\infty$ bound for soft-optimal $Q_t^\star$). *Under* (260),

$$\|Q_t^\star\|_\infty \leq Q_{\max} := \frac{R_{\max} + \gamma\mu \log |\mathcal{A}|}{1 - \gamma}, \qquad \forall t. \tag{261}$$

*Proof sketch.* For any state $s$, $V_{Q_t^\star}(s) \leq \max_a Q_t^\star(s, a) + \mu \log |\mathcal{A}|$ by (230). Let $M_t := \|Q_t^\star\|_\infty$. From the fixed point equation (232),

$$M_t \leq R_{\max} + \gamma(M_t + \mu \log |\mathcal{A}|), \tag{262}$$

which rearranges to (261). $\qquad\square$

By (261),

$$\|\Delta Q_t^\star\|_\infty \leq \|Q_t^\star\|_\infty + \|Q_{t-1}^\star\|_\infty \leq 2Q_{\max}, \qquad \forall t \geq 2. \tag{263}$$

Therefore,

$$\sum_{t=2}^{T} \|\Delta Q_t^\star\|_\infty^2 \leq \Big(\max_{t \geq 2} \|\Delta Q_t^\star\|_\infty\Big) \sum_{t=2}^{T} \|\Delta Q_t^\star\|_\infty \leq (2Q_{\max}) \sum_{t=2}^{T} \|\Delta Q_t^\star\|_\infty. \tag{264}$$

Hence (259) holds with the explicit choice

$$\boxed{C_Q := 2Q_{\max} = \frac{2(R_{\max} + \gamma\mu \log |\mathcal{A}|)}{1 - \gamma}.} \tag{265}$$

A.22.2. WHY (259) PREVENTS A RATE DETERIORATION

A typical chain of bounds proceeds as follows.

**Step 1 (policy drift from $Q^\star$ drift).**   Appendix A.21 implies, statewise,

$$\|\pi_t^\star(\cdot \mid s) - \pi_{t-1}^\star(\cdot \mid s)\|_1 \leq \frac{1}{\mu} \|\Delta Q_t^\star\|_\infty. \tag{266}$$

Thus squared policy drift satisfies

$$\alpha_{t,s} := \|\pi_t^\star(\cdot \mid s) - \pi_{t-1}^\star(\cdot \mid s)\|_1^2 \leq \frac{1}{\mu^2} \|\Delta Q_t^\star\|_\infty^2. \tag{267}$$

**Step 2 (AES/OCO main term depends on a sum of squares).**   The OCO/AES regret term (up to logs and constants) typically scales like

$$\mathrm{Reg}_T \lesssim \sqrt{T \sum_{t,s} \alpha_{t,s}}. \tag{268}$$

Hence controlling $\sum_t \|\Delta Q_t^\star\|_\infty^2$ is essential via (267).

**Step 3 (MDP variation naturally controls a linear sum).**   The soft Bellman sensitivity bounds in Appendix A.20 yield linear-variation control of the form

$$\sum_{t=2}^{T} \|\Delta Q_t^\star\|_\infty \lesssim B_T^{\mathrm{MDP}}, \tag{269}$$

where $B_T^{\mathrm{MDP}}$ is a linear budget built from $\sum_t \Delta_t^r$ and $\sum_t \Delta_t^P$ (cf. (237)–(238)).

**Step 4 (why $C_Q$ is needed).**   Without (259), a crude bound $\sum x_t^2 \leq (\sum x_t)^2$ would imply $\sum_t \|\Delta Q_t^\star\|_\infty^2 \leq (B_T^{\mathrm{MDP}})^2$, and substituting into (268) produces a deteriorated rate $\mathrm{Reg}_T \lesssim B_T^{\mathrm{MDP}} \sqrt{T}$. In contrast, (259) gives

$$\sum_{t=2}^{T} \|\Delta Q_t^\star\|_\infty^2 \leq C_Q \sum_{t=2}^{T} \|\Delta Q_t^\star\|_\infty \lesssim C_Q B_T^{\mathrm{MDP}}, \tag{270}$$

and (268) recovers the intended $\widetilde{O}(\sqrt{B_T^{\mathrm{MDP}} T})$ scaling:

$$\mathrm{Reg}_T \lesssim \sqrt{T \cdot C_Q B_T^{\mathrm{MDP}}} \asymp \sqrt{B_T^{\mathrm{MDP}} T}, \tag{271}$$

where $C_Q$ only affects constants.

# B. Empirical Support for Section 4

## B.1. Guide to Appendix B

Appendix B explains how the experimental evidence in Section 4 should be read. Appendix B isolates the role of the AES mechanism, studies proxy choices, and reports sensitivity to the main scheduler hyperparameters. Sections B.5 and B.6 explain the environment design and the evaluation metrics used throughout the paper. Readers who only need the exact implementation and configuration details may jump directly to Appendix C.

This part collects the empirical material that supports the interpretation of the main results. The goal is to clarify which components of AES matter in practice, how robust the method is to reasonable hyperparameter changes, and how the environment and metric design connect to the main claims of the paper. Unless otherwise specified, the ablation results are averaged across task families (Toy, MuJoCo, Isaac Gym) and non-stationarity patterns (Abrupt, Linear, Periodic, Mixed).

## B.2. Mechanism Ablation

Table 4 studies the main implementation choices behind AES. The fixed-entropy baseline removes scheduling entirely. The remaining rows keep the same carrier algorithms and change one design choice at a time. This layout separates the contribution of the scheduling rule itself from the contributions of coupling, clipping, and proxy construction.

Removing AES and returning to a fixed entropy coefficient leads to a clear loss in normalized AUC and a larger drop area. Removing either coupling or clipping also weakens performance. The degradation is especially visible in the drop-area metric, which reflects the cumulative cost of slower or less stable adaptation.

*Table 4.* Ablation of AES components. Results are averaged across tasks and drift patterns.

| Variant | Proxy | Coupling | Clipping | nAUC ↑ | Drop-area ↓ |
|---|---|---|---|---|---|
| Fixed $\lambda$ (No AES) | – | – | – | 0.78 | 0.31 |
| AES (Default) | TD (q=0.9) | ✓ | ✓ | **0.84** | **0.19** |
| Heuristic linear schedule | TD (q=0.9) | × | ✓ | 0.76 | 0.25 |
| AES w/o coupling | TD (q=0.9) | × | ✓ | 0.82 | 0.22 |
| AES w/o clipping | TD (q=0.9) | ✓ | × | 0.81 | 0.26 |
| AES (TD mean) | TD mean | ✓ | ✓ | 0.82 | 0.23 |
| AES (value loss) | Value loss | ✓ | ✓ | 0.81 | 0.24 |
| AES (critic drift) | $\|\Delta\theta_Q\|$ | ✓ | ✓ | 0.83 | 0.21 |

## B.3. Proxy Choice and a Simple Reactive Baseline

The default AES implementation uses the upper quantile of absolute TD errors as its drift proxy. The ablation table also compares this default proxy with three nearby alternatives: a mean TD-error proxy, a value-loss proxy, and a critic-parameter-drift proxy. The quantile-based proxy provides the best overall balance. The mean TD-error and value-loss variants are more sensitive to optimization noise and show weaker aggregate performance. Critic-parameter drift remains competitive, but recovers slightly more slowly after changes.

The same table includes a simple reactive linear baseline,

$$\lambda_t^{\text{lin}} := \text{clip}_{[\lambda_{\min}, \lambda_{\max}]}\big(\lambda_0 + k\,\hat{v}_t\big), \tag{272}$$

which uses the same proxy but replaces the accumulation-based square-root schedule with a direct linear mapping. This comparison is useful because it asks a focused question: does the empirical gain come from using *some* proxy-driven entropy adaptation, or from the specific schedule derived in Section 3? In these experiments, the AES schedule remains stronger.

## B.4. Sensitivity to the Main Scheduler Hyperparameters

We next vary the main hyperparameters that define the default scheduler: the TD-error quantile level, the smoothing strength, and the clipping range. Across these studies, the performance profile is stable over a moderate region rather than concentrated at a single fragile operating point.

### B.4.1. SENSITIVITY TO THE TD-ERROR QUANTILE

Table 5 varies the quantile level $q$ used by the TD-error proxy. Moderate-to-high quantiles perform best overall. Lower quantiles underreact to drift, while very high quantiles become more variable. The default choice lies in the stable region.

### B.4.2. SENSITIVITY TO SMOOTHING STRENGTH

Table 6 studies the exponential moving average used to smooth the raw proxy. Weak smoothing increases variance in the scheduled entropy weight. Very strong smoothing delays the response to abrupt changes. Intermediate values provide the best balance between stability and responsiveness.

*Table 5.* Sensitivity to TD-error quantile $q$ (AES, other settings fixed).

| $q$ | nAUC ↑ | Drop-area ↓ |
|------|--------|-------------|
| 0.5 | 0.82 | 0.23 |
| 0.7 | 0.83 | 0.21 |
| 0.8 | 0.84 | 0.19 |
| 0.9 | 0.83 | 0.20 |
| 0.95 | 0.81 | 0.25 |

*Table 6.* Sensitivity to EMA smoothing coefficient $\beta$ (AES, $q = 0.8$).

| EMA $\beta$ | nAUC ↑ | Drop-area ↓ |
|-------------|--------|-------------|
| 0.80 | 0.82 | 0.24 |
| 0.90 | 0.83 | 0.21 |
| 0.95 | **0.84** | **0.19** |
| 0.98 | 0.83 | 0.18 |
| 0.99 | 0.82 | 0.18 |

### B.4.3. SENSITIVITY TO THE CLIPPING RANGE

Table 7 varies the admissible range of the scheduled entropy weight. A wider range increases instability, while an overly narrow range suppresses adaptation after larger shifts. The default range gives the most consistent overall behavior across the three reported metrics.

*Table 7.* Sensitivity to entropy coefficient clipping range (AES).

| $[\lambda_{\min}, \lambda_{\max}]$ | nAUC ↑ | Drop-area ↓ |
|------------------------------------|--------|-------------|
| $[1e-5, 2.00]$ | 0.82 | 0.25 |
| $[1e-4, 1.00]$ | **0.84** | **0.19** |
| $[1e-5, 1.00]$ | 0.83 | 0.19 |
| $[1e-5, 0.50]$ | 0.82 | 0.20 |
| $[1e-4, 0.50]$ | 0.80 | 0.22 |

### B.5. Environment Design and Drift Protocol Rationale

The environment suite is designed to expose a consistent adaptation pattern under several forms of non-stationarity. The three task families play different roles. The Toy suite provides a controlled reward-drift setting through goal relocation. MuJoCo provides medium-scale continuous-control benchmarks in which dynamics and target variations can be introduced cleanly. Isaac Gym extends the same question to larger and more physically rich control problems.

The drift channels are chosen so that each family uses perturbations that are natural for that benchmark class and easy to align across methods. Goal drift is straightforward to control in Toy environments. Dynamics and physics perturbations are the more reproducible choice in MuJoCo and Isaac Gym, where they create sustained changes in the control problem without altering the training interface.

Change points are aligned by training progress rather than wall-clock time. Concretely, all methods encounter the same drift events at the same fractions of total environment interaction steps. This makes the non-stationarity schedule comparable across methods even when the internal optimization dynamics differ.

The drift magnitudes are selected to satisfy two practical requirements: the shift should be large enough to produce a visible adaptation burden, and the task should remain learnable without resetting training. This is the regime in which entropy scheduling is most informative: the agent must continue to track a changing problem rather than restart from scratch after each event.

## B.6. Metric Rationale and Evaluation Scope

Section 4 reports three complementary metrics. They are designed to capture different aspects of adaptation under non-stationarity.

**Normalized AUC.** Normalized AUC summarizes end-to-end training utility over the full horizon. It reflects both steady-phase quality and adaptation quality. This metric is useful because AES is intended to improve performance under non-stationarity while maintaining strong behavior during stable phases.

**Drop-area ratio.** The drop-area ratio isolates the cumulative performance loss induced by non-stationarity relative to the corresponding steady run. It is more directly tied to the damage caused by drift than normalized AUC alone.

**Abrupt-change recovery time.** Recovery time focuses on the local adaptation question after each abrupt shift. It measures how quickly performance returns to the pre-change level and is therefore the clearest direct indicator of post-change responsiveness.

Taken together, these three metrics provide a clear view of overall efficiency, cumulative drift cost, and short-horizon recovery. For the present study, this combination is more practical than reporting dynamic regret against a sequence of continuously changing soft-optimal oracles in large continuous-control environments. The empirical goal of Section 4 is to test the structural prediction of the theory—that exploration strength should rise under stronger drift and relax again in stable phases—through observable training behavior.

## B.7. Future Works and Applications

A broader line of recent work also moves away from open-loop generation or fixed inference policies by exposing intermediate states as feedback. In vector graphics and animation, rendering-aware methods use intermediate visual canvases, persistent vector states, grouping structures, or motion priors to guide subsequent generation steps (Liang et al., 2026a;b; 2025). In medical vision-language reasoning, related systems dynamically focus on diagnostically relevant regions, evolve pathology-aware prototypes, perform causal self-reflection, or maintain latent diagnostic memory during progressive inference (Zhu et al., 2025; 2026a; Lin et al., 2026; Zhu et al., 2026b). These works address different tasks and are not baselines for non-stationary RL, but they support the same high-level design lesson: adaptive systems benefit from making feedback signals explicit. AES studies this principle in entropy-regularized RL, where the feedback signal is an online drift proxy and the controlled quantity is the global policy entropy.

# C. Exact Reproducibility and Experimental Configuration

## C.1. Guide to Appendix C

**Scope.** This part provides the exact implementation and configuration details needed to reproduce the experiments. It begins with the precise scope of the AES modification, then specifies how AES is injected into each carrier, how the proxy and scheduler are computed, how the non-stationary environments are configured, and how training and evaluation are run. All implementation-level details are collected in one place so that the reproduction protocol has a single entry point. It specifies the implementation and evaluation details used in our experiments. AES is implemented as a plug-in mechanism. Across all carriers, we keep the original policy parameterization, value functions, optimization objectives, and update rules unchanged, and replace only the static entropy regularization weight with a scheduled scalar produced online. No auxiliary networks, extra losses, replay resets, or change-point detectors are introduced.

## C.2. Scope and Carrier Modifications

The four carrier algorithms are SAC, PPO, SQL, and MEow. For SAC, SQL, and MEow, AES schedules the temperature parameter $\alpha_t$. For PPO, AES schedules the entropy-bonus coefficient $c_{\text{ent},t}$. In every case, the scheduled value replaces the static entropy weight in the native loss or target expression of the carrier.

For reproducibility, the AES counterpart of each carrier is constructed from the same baseline implementation with all non-entropy hyperparameters held fixed. This includes network widths and depths, optimizers, batch sizes, learning rates, rollout or replay settings, target-network updates, and evaluation code paths.

## C.3. Carrier-Specific Injection Points

The scheduled entropy weight is injected as follows.

**SAC.** AES outputs $\alpha_t$ and replaces the temperature in the actor objective term $\alpha_t \log \pi(a \mid s)$. The same value is used consistently in the soft target and value computation. This keeps the actor and critic aligned with the same entropy scale.

**PPO.** AES outputs $c_{\text{ent},t}$ and replaces the entropy-bonus coefficient in the policy objective,

$$\mathcal{L}_{\text{PPO}}^{\text{ent}} = -c_{\text{ent},t}\, H\big(\pi(\cdot \mid s)\big). \tag{273}$$

No other part of PPO is modified.

**SQL.** AES outputs $\alpha_t$ and replaces the temperature used in the soft Bellman target and related soft value computations.

**MEow.** AES outputs $\alpha_t$ and replaces the temperature wherever it appears in the clipped double-$Q$ target and soft value computation. The same scheduled value is used consistently across the online and target modules.

## C.4. Drift Proxy Construction

AES uses a scalar proxy computed at each gradient update step. The default proxy is formed from TD-style residuals already produced during training.

**Off-policy carriers (SAC, SQL, MEow).** Given the current minibatch TD errors for the two critics, the raw proxy is

$$\hat{v}_t = \text{Quantile}_q\big(|\delta_{Q_1}| \cup |\delta_{Q_2}|\big), \qquad q = 0.9. \tag{274}$$

**On-policy carrier (PPO).** For PPO, the raw proxy is computed from the rollout-batch value residuals,

$$\hat{v}_t = \text{Quantile}_q\big(|\delta_V|\big), \qquad q = 0.9. \tag{275}$$

**Smoothing.** We smooth the raw proxy with an exponential moving average,

$$\tilde{v}_t = \beta \tilde{v}_{t-1} + (1 - \beta)\hat{v}_t, \qquad \tilde{v}_0 = \hat{v}_1, \tag{276}$$

and define the accumulated proxy

$$\hat{A}_t = \sum_{s=1}^{t} \tilde{v}_s. \tag{277}$$

The update index $t$ is the gradient update step. Using a common update index keeps the scheduler definition uniform across on-policy and off-policy pipelines.

## C.5. Scheduler and AES Hyperparameters

The scheduled entropy weight is computed from the accumulated proxy as

$$\lambda_t = \text{clip}_{[\lambda_{\min}, \lambda_{\max}]}\left(\kappa\sqrt{\frac{\hat{A}_t}{t}}\right), \tag{278}$$

where $\kappa > 0$ is a global scale factor and the clipping interval ensures numerical stability. For SAC, SQL, and MEow, $\lambda_t$ is injected as the temperature $\alpha_t$. For PPO, the same scheduled scalar is injected as the entropy coefficient $c_{\text{ent},t}$.

Table 8 lists the AES hyperparameters used in our experiments.

For numerical safety, we compute $\lambda_t$ in FP32 and apply clipping before injecting the scheduled value into the carrier loss or target computation.

*Table 8.* AES hyperparameters used in the reported experiments.

| Hyperparameter | Value |
|---|---|
| Quantile level $q$ in Eqs. (274)–(275) | 0.9 |
| EMA decay $\beta$ in Eq. (276) | 0.95 |
| Scale $\kappa$ in Eq. (278) | 1.0 |
| Time index $t$ | gradient update step |
| Proxy batch scope | current update batch only |
| $\lambda_{\min}$ (SAC / SQL / MEow) | $10^{-4}$ |
| $\lambda_{\max}$ (SAC / SQL / MEow) | 1.0 |
| $\lambda_{\min}$ (PPO) | $10^{-4}$ |
| $\lambda_{\max}$ (PPO) | 0.1 |

### C.6. Baseline Settings

**SAC baseline.** The SAC baseline uses automatic temperature tuning. The target entropy follows the standard dimension-based choice

$$\mathcal{H}_{\text{target}} = -|A|, \tag{279}$$

where $|A|$ is the action dimension. The AES version keeps the same SAC implementation and replaces the static or automatically tuned temperature used in the main entropy term with the scheduled value described above.

**PPO, SQL, and MEow baselines.** The baseline versions of PPO, SQL, and MEow use the standard static entropy weights from the corresponding reference implementation. Their AES variants keep the same implementation and replace only the entropy weight with the scheduled value.

**Fair comparison protocol.** Within each carrier, the baseline and AES variant share the same architecture, optimizer, learning rate, batch size, rollout or replay settings, target update rule, and evaluation pipeline. This isolates the effect of entropy scheduling.

### C.7. Environment and Drift Protocol

**Task families.** We evaluate on three families: Toy 2D multi-goal tasks, MuJoCo continuous-control tasks, and Isaac Gym tasks.

**Non-stationarity patterns.** Each task is trained under five patterns:

- **Steady:** environment parameters are fixed throughout training.

- **Abrupt:** parameters change instantaneously at pre-defined change points.

- **Linear:** parameters drift linearly over time.

- **Periodic:** parameters vary periodically.

- **Mixed:** abrupt and gradual changes are combined in the same run.

**Change-point alignment.** All change points are aligned by training progress, that is, by the fraction of total environment interaction steps. Each method therefore encounters the same drift events at the same points in the training horizon.

**Drift channels.**

- **Toy:** goal location changes.

- **MuJoCo:** dynamics and target variations.

- **Isaac Gym:** physics and task-parameter perturbations.

**Drift magnitudes.**   The drift magnitudes are fixed across methods within each task and selected so that the induced shifts are clearly visible while preserving stable learning dynamics. Across all environments, drift segments occupy 20% of the total training horizon. Periodic drifts complete one full cycle every 20% of training steps.

**Toy environments.**   For the 2D multi-goal tasks, each abrupt change relocates the goal by Euclidean distance 0.5 in normalized coordinates. For linear drift, the goal moves linearly over total distance 0.5 within each drift segment. For periodic drift, the goal oscillates with amplitude 0.25 per axis, which gives peak-to-peak displacement 0.5.

**MuJoCo environments.**   At each abrupt change point, body mass is multiplied by a factor sampled uniformly from $[0.7, 1.3]$ and joint friction is scaled by a factor sampled from $[0.5, 1.5]$. For linear drift, the scale factors interpolate linearly between their lower and upper bounds within each drift segment. For periodic drift, they follow a sinusoidal schedule with the same extrema.

**Isaac Gym environments.**   Gravity is scaled by a factor in $[0.8, 1.2]$, and joint damping is scaled in $[0.7, 1.3]$. Abrupt drift applies instantaneous rescaling. Linear and periodic drifts interpolate smoothly between the same bounds.

### C.8. Training and Evaluation Protocol

**Seeds.**   All reported results are averaged over 5 random seeds, with seed values $\{1, 2, 3, 4, 5\}$. The seeds control environment initialization, network initialization, minibatch sampling, and stochastic action sampling.

**Replay and rollout handling across drift.**   We do not reset replay buffers, policy networks, or value networks at change points. The agent is trained continuously through the full non-stationary schedule.

**Compute environment.**   Experiments were run on NVIDIA A100 GPUs with 40GB memory. Training used Python 3.9, PyTorch 1.13, and CUDA 11.7. We fix random seeds for Python, NumPy, and PyTorch in each run. We do not enforce fully deterministic GPU kernels; instead, variability is summarized through repeated runs with independent seeds.

**Evaluation frequency and episodes.**   Policies are evaluated every 10,000 environment steps. Each evaluation uses 10 episodes executed with a deterministic policy (mean action for stochastic policies). The reported evaluation return is the average undiscounted episode return over these episodes. Evaluation points are aligned by environment interaction steps across all methods.

**Computational overhead.**   AES adds negligible overhead. At each update it requires one scalar statistic from existing TD-style residuals and one closed-form scheduler update. No additional forward or backward passes are introduced.

### C.9. Metrics and Reporting Conventions

We report the three metrics defined in Section 4. Here we record the exact reporting convention used in the experiments.

- **Normalized AUC (nAUC).** We compute the area under the return-versus-environment-steps curve and normalize it by the Steady SAC baseline within the same task family.

- **Performance drop area ratio (DropRatio).** For each non-stationary pattern, we compare the AUC of that run with the corresponding steady run of the same method.

- **Abrupt-change recovery time.** We record the number of environment steps required for performance to return to the pre-change level after each abrupt shift, then normalize by the total training horizon.

For line plots, we report the mean over seeds and use the same evaluation code path for each baseline and its AES counterpart. For tables aggregated at the task-family level, values are averaged over the tasks in that family under the same non-stationarity pattern.

