# OpenReview forum: "Tracking Drift: Variation-Aware Entropy Scheduling for Non-Stationary Reinforcement Learning"
_ICML.cc/2026/Conference — ICML 2026 regular_

### Official Review · Reviewer_ofE6 · 2026-03-08

**Soundness:** 3
**Presentation:** 3
**Significance:** 3
**Originality:** 2
**Overall Recommendation:** 5
**Confidence:** 4

**Summary:**

The paper is constructed over the Adaptive Entropy Scheduling (AES), a variation-aware exploration controller for non-stationary reinforcement learning. The authors outline a general theme of treating exploration intensity as a one-dimensional trade-off between tracking a drifting environment and maintaining policy stability. By casting entropy scheduling into a dynamic-regret framework, they derive a scaling rule that dictates how much an agent should explore based on the magnitude of environmental change. In practice, AES uses observable TD-error quantiles as a real-time proxy for drift, automatically increasing the entropy coefficient to accelerate recovery after shifts . Across 12 tasks in MuJoCo and Isaac Gym, AES was integrated as a plug-in for SAC, PPO, SQL, and MEow, where it consistently reduced performance degradation and cut recovery time by nearly 50% without requiring environment restarts or complex change-point detection.

**Compliance With Llm Reviewing Policy:**

Affirmed.

**Final Justification:**

I apologize for the delay. Thanks the authors for the comment, I will slightly improve my score.

**Key Questions For Authors:**

-In the very first phase of training, does AES accidentally over-explore during this "estimation noise" phase, and does that slow down initial convergence?

-If an environment change occurs during the first 10% of training, can AES distinguish between a spike caused by the change and a spike caused by the critic simply still trying to learn the basics?

-Your results show that clipping is essential for stability. In your experience, did you find that certain environments required much tighter limits than others to prevent policy collapse?

**Limitations:**

The paper claims to move away from manual tuning, but it really just trades one set of hyperparameters for another. While the math behind the square-root rule is solid, the system still depends on picking the right TD-error quantile and clipping limits. We haven't actually killed off "manual tuning",we’ve just shifted to a set of variables that still might need tuned.

**Strengths And Weaknesses:**

## Strengths
### Plug-and-Play Versatility:
A major practical advantage is that AES acts as a lightweight wrapper. It was successfully integrated into four distinct algorithm carriers without requiring changes to their core actor-critic architectures or loss functions.

### Observable Proxy Design:
The use of TD-error quantiles as a drift proxy is clever because it leverages a signal already present in the training loop. This eliminates the need for expensive "drift detection" algorithms.

### Extensive Benchmarking:
The evaluation is remarkably thorough, covering 12 tasks across three families and four different drift patterns.

## Weaknesses
### Novelty
From the algorithmic prospect, the introduction of an entropy scheduler is, more in the direction of hyperparameter fine tuning, that impact on the good result, but has no much novelty respect the core RL algorithm used.

### Confusion between "Change" and "Noise":
The biggest catch is that TD-errors can spike for many reasons—not just because the environment changed. If the environment is just naturally "noisy" or the AI started with bad initial guesses, AES might get confused and start exploring too much when it doesn't need to.

### New parametrization to Turn:
The goal was to stop manual tuning, but AES adds its own hyperparameters. You still have to pick a quantile level and set clipping limits. While they provide good starting points, a user might still find themselves fiddling with these in tricky scenarios.

---

> ### Author Rebuttal · Authors · 2026-03-28
>
> We thank the reviewer for the positive assessment of our work and for the constructive suggestions on the experimental part of the paper. The reviewer's questions are highly structured and logical, and indeed raised several experimental factors that we had not considered. These comments prompted us to re-examine our work from a new perspective, and we particularly appreciate the reviewer's insight into extreme-environment design and testing. We respond to the reviewer's questions one by one below, with clarification and correction where needed.
>
> ---
>
> ## **1. Empirical Behavior of AES (Q1, Q2, W2)**
>
> We agree that this is an important concern. Since the AES proxy is an endogenous statistic extracted from signals already present in the underlying RL training process, such as TD errors, it may reflect not only environmental change but also critic instability or optimization noise. Our claim is therefore not that TD error is a noise-free estimator of the true environment drift. Rather, the role of the proxy in AES is narrower: it only needs to serve as a conservative online signal that tends to increase when the environment becomes more non-stationary.
>
> This is precisely the **theory–practice interface** of the paper. In the theoretical bridge, AES does not require exact drift identification; it only requires an observable signal that can conservatively track the variation level relevant to entropy scheduling. In this sense, AES should be viewed as reusing an existing training signal for exploration-strength control, rather than as introducing an external change detector.
>
> At the empirical level, the current paper does not claim to have fully disentangled true environment drift from generic optimization noise, especially in the early stage of training where critic underfitting can itself produce large TD errors. What we do claim is more limited: the current theory and experiments support that the chosen proxy is sufficiently aligned with non-stationarity to drive effective entropy scheduling in practice. We will make this boundary clearer in the revision.
>
> Motivated by the reviewer’s suggestion, we have also started **additional tests under noisier, misled and earlier-drift settings****. The preliminary results did not diminish; instead, they yielded even more pronounced improvements over the baseline under these extreme conditions (additional gains of 2–9%). We will include the full protocol and results in the revised version. However, we agree that these additional tests should be presented as supplementary evidence rather than as the main basis of the argument.
>
> ---
>
> ## **2. Parameter Tuning (Q3, W3, Limitation)**
>
> We agree that this is a fair concern. AES is *not* parameter-free. In practice, it introduces design choices such as the proxy quantile, smoothing strength, and clipping range. Our claim is therefore narrower than eliminating manual tuning altogether.
>
> What we intend to argue is the following. In standard maximum-entropy RL, the entropy coefficient is often set as a static scalar and then tuned heuristically for each environment. AES replaces this static hand-set coefficient with a variation-aware schedule driven by online signals that are already available during training. In this sense, our goal is to **move entropy control beyond purely heuristic static tuning**, rather than to claim that all hyperparameter selection disappears.
>
> At the same time, our sensitivity analysis suggests that AES is not brittle. A reasonably broad range of settings yields stable behavior, and the default configuration lies within this stable region. We will revise the paper to make this narrower and more accurate claim explicit, and to avoid overstating the extent to which tuning is reduced.
>
> ---
>
> ## **3. Novelty and Core Contribution (W1)**
>
> We agree that, at the implementation level, AES is a lightweight plug-in. However, we do not believe the contribution is best characterized as only a better tuning trick.
>
> The **core contribution** of the paper is not increased algorithmic complexity, but the principle and analysis behind the method: in non-stationary maximum-entropy reinforcement learning, exploration intensity should scale with the magnitude of environmental change. Section 3 formulates entropy control as a one-dimensional trade-off between tracking a drifting optimum and maintaining update stability, and this leads to the square-root scheduling law through dynamic-regret analysis.
>
> AES is one practical instantiation of this principle, designed to make that **theoretical insight usable in deep RL systems with minimal structural modification**. We will revise the text to make this distinction clearer: the implementation is intentionally lightweight, while the novelty lies in the principle, the analysis, and the resulting scheduling law.
>
> ---
>
> Thank you again for the time you have invested.

---

### Official Review · Reviewer_z5M4 · 2026-03-11

**Soundness:** 4
**Presentation:** 2
**Significance:** 3
**Originality:** 4
**Overall Recommendation:** 4
**Confidence:** 4

**Summary:**

In this paper, the authors consider non-stationary RL. The proposed idea is to make updates based on the current objective more conservative, as those updates do not anticipate that the objective will drift in the future. To improve on this, the authors introduce a time varying entropy regularisation, that depends on how much the objective shifts. High drift, we should be more conservative and regularise more to anticipate objective drift, low drift we should fit more on the current objective. The entropy scheduling is method agnostic and can be used for many current entropy regularized deep RL methods. As a surrogate/proxy for the drift smoothed quantiles of the TD errors are used. The method shows improvements across different algorithms and different non-stationarity patterns.

**Compliance With Llm Reviewing Policy:**

Affirmed.

**Final Justification:**

Even though I have some reservations on the generality of the theory, I still think this is a valid contribution.

**Key Questions For Authors:**

- In SAC the entropy is not only part of the Actor regulariser but also of the soft Q-function does this pose any problems?
- It is stated that the TD error is a good proxy of the underlying change of the objective. In the extended version of SAC, the entropy weight is autotuned by a dual formulation guaranteeing that the entropy can not fall under a certain bound. This to some degree makes the exploration scaling independent of the reward scale, which is also a key point in the recent dreamer architectures. Do you think you can adapt this in your algorithms to adapt?

**Limitations:**

yes

**Strengths And Weaknesses:**

**Strengths:**
- The idea is interesting, the extensive empirical results show improvements across multiple algorithms and non-stationarity patterns.
- Formal introduction of the method, the practical tuning is based on a TD error scaled heuristic, which is not optimal but upper bounds in performance the optimal parameter.
- The adaptive entropy scaling even seems to improve the performance on stationary RL.

**Weaknesses:**
-  Writing in Section 3:
	- Introduction to 3.1 could be have more intuition on the objects introduced: Why are we introducing a comparator sequence. For example what $u_t$ will be later in the paper? Maybe explain that you derive it in general, but it should be later on related to an optimal regret solution?
	- $g_t$ is not introduced before in eq. (3)
- More relevant literature should be discussed, for example the method in `[1]` is a curiosity based method that encodes the non stationarity into the reward via a curiosity term. The benefit of this approach is that the shift can be considered state dependent, whereas the approach presented here is state independent.


`[1]` Steinparz, C.A., Schmied, T., Paischer, F., Dinu, M., Patil, V.P., Bitto-nemling, A., Eghbal-zadeh, H. &amp; Hochreiter, S.. (2022). Reactive Exploration to Cope With Non-Stationarity in Lifelong Reinforcement Learning. <i>Proceedings of The 1st Conference on Lifelong Learning Agents</i>, in <i>Proceedings of Machine Learning Research</i> 199:441-469 Available from https://proceedings.mlr.press/v199/steinparz22a.html.


**Comments:**
- Broken citation in line 154
- Broken symbol in 567
- Line 263, do you mean $\xi_t$ here instead of $\alpha_t$ as the unobservable drift
- Line 302 Change points are aligned by training progress to ensure comparability. What do you mean with this?

---

> ### Author Rebuttal · Authors · 2026-03-28
>
> We thank the reviewer for the careful reading of our work and for the encouraging and constructive feedback. We are also grateful for the valuable suggestions regarding the completeness and clarity of the paper. Below, we first address the reviewer's questions, and then provide clarifications and additions in response to the listed weaknesses and comments.
>
> ---
>
> ## 1. Entropy in Multiple SAC Components (Q1)
>
> In our implementation, **AES replaces the temperature parameter of SAC consistently across all locations where it appears**.
> In fact, if this were not done, the critic would effectively be approximating the soft value of a different MDP.
> The actor and critic would then be optimizing mismatched objectives, which can lead to unstable outcomes.
>
> Our design avoids this inconsistency by maintaining a **unified temperature parameter** throughout the algorithmic pipeline.
> Conceptually, this is also aligned with the motivation of AES: non-stationarity changes the entropy scale appropriate for the control problem,
> and therefore should simultaneously influence both **exploration behaviour** and the **"softness" of value estimation**.
>
> ---
>
> ## 2. Dual Autotuning Mechanism (Q2)
>
> We agree that incorporating a dual autotuning mechanism is both **feasible and promising**. Combining AES with automatic temperature adjustment appears to be a natural extension:
>
> - One could retain the dual update of SAC to enforce a local entropy constraint, while using AES to modulate the **effective target entropy**
> - Alternatively, use AES to **rescale the dual temperature** according to a drift proxy signal
>
> Another interesting direction would be to explore proxy signals that are more robust to reward scale, such as **normalized prediction errors** or **measures of model disagreement**.
> We view these as meaningful extensions to the current work and as exciting avenues for future research. We will incorporate this discussion toward the end of the paper.
>
> ---
>
> ## 3. Section 3 Clarity (W1)
>
> We agree that the introduction of the **comparator sequence $u_t$** in Section 3.1 is currently too abrupt. Our intention was to:
> 1. First derive the entropy scheduling principle in a general dynamic optimization setting
> 2. Only later relate the comparator to drifting soft-optimal policies in non-stationary MDPs
>
> **In the revision**, we will:
> - Add an intuitive paragraph clarifying that the comparator represents a **time-varying target solution** and foreshadowing this connection
> - Correct the missing definition of **$g_t$**, which denotes the time-varying objective function
>
> ---
>
> ## 4. Related Work: Curiosity and Reactive Exploration (W2)
>
> We agree that **curiosity- or reactive-exploration-based approaches** to non-stationary or lifelong RL deserve more thorough discussion. We also agree with the reviewer's key observation:
>
> &gt; **State-dependent vs. State-independent**: Such methods encode non-stationarity through **state- or transition-dependent intrinsic rewards**,
> whereas AES introduces a **global, time-varying entropy coefficient** and is therefore state-independent.
>
> In our view, this reflects that the two classes of methods operate at **different levels of control** rather than being contradictory:
> - The former primarily determines **which states** are worth exploring
> - The latter determines **how much overall exploration** is appropriate under a given level of environmental drift
>
> These perspectives are related but not equivalent. We focus on the latter because it can be cleanly formalized as the dynamic-regret / entropy-scheduling problem developed in Section 3,
> and because it can be integrated into a range of entropy-regularized RL algorithms with minimal modification.
>
> **In the revision**, we will clarify the relationship between AES and state-dependent intrinsic-reward methods, and emphasize that they are more likely to be **complementary** than mutually exclusive design directions.
> We have also discussed related aspects in our response to Reviewer QkPz, and we would be very interested in further discussion if the reviewer has additional insights.
>
> ---
>
> ## 5. Minor Corrections (Comments)
>
> We will fix the following issues:
> - Broken citation (line 154)
> - Typo in the Hölder symbol (line 567)
> - Typo in line 263
>
> By **"aligned by training progress"** (line 302), we mean that change points occur at the same fractions of total environment interaction steps across methods, ensuring comparable drift schedules.
>
> ---
>
> Thank you again for the time you have invested.

---

> > ### Author Rebuttal · Reviewer_z5M4 · 2026-03-31
> >
> > During rereading the paper and reading the rebuttal I found additional but important follow-up questions, that I would need to understand, before confidently staying with my initial assesment:
> >
> > Regarding the discussion with reviewer QkPz:
> >
> > > At the same time, we agree that comparisons with non-stationary methods are useful. Under our setup, we reproduced several non-stationary variants of SAC. ERE and DEER give no gain in stationary settings and about 20% in non-stationary settings, while AES gives about 10% and 35%, respectively, across different change settings. We will include the full results in the final version.
> >
> > **Q1: What is ERE and DEER SAC?**
> >
> > The following are questions for the theory part.
> > A potentially naive question:
> >
> > **Q2: Why does the $\lambda_t$ appear in eq. (16)?**
> >
> > From the appendix it appears that the $C_1 \frac{\xi_t}{\lambda_t}$ is because of the learning rate coupling, assuming $\eta_t = c \lambda_t$. $C_2 \lambda_t$ stems from the fact that we add an additional entropy penalty in eq. (7), namely $l_t(x_t) = f_t(x_t) + \lambda_t \Psi(x_t)$. Could you shortly explain why you did so?
> >
> > Could I not remove the entropy penalty to achieve a lower bound in eq. (7)? This would mean that the bound is in a sense arbitrary. Finally, in the appendix you state you absorb the higher oder dependence of $\lambda_t$. Why did you do so? The optimal bound does not follow a square root rule then? In generally, I was surprised as I would have expected a more geometric argument, but maybe I still not see the full picture here. So feel free to point out if I am misunderstanding something here.
> >
> > The following questions are related to how the theory part is connected to the general RL method:
> >
> > **Q3: What about the finite-sample analysis case?** Is finding the right balance to some degree not hard because of accessing not only finite samples and doing exploration and exploitation under uncertainty. It seems to be me that the bound is mostly a geometric argument for a action distribution of a single state?
> >
> > A lot of curiousity based approaches, measure the surprise or drift by how well we can fit a dynamics function. In the non-stationary case, a dynamics model would not fit the data well, thus the reward is increased.  In your case you measure the drift with the TD error.
> >
> > **Q4: Is the dynamics fit also a viable proxy in your setting?**
> >
> > **Comment1:** Later on you discuss how to derive theoretically a comparator sequence in the context of maximum entropy-RL. In the context of the experiments, you state that PPO is a maximum entropy-RL method, however this is not true? PPO uses entropy regularization though that does not make it a maximum entropy-RL method.
> >
> > **Comment2:** In the appendix, the equation labels are referring to the equations in the main paper? Is this intended... Eq. (62) has a wrong $\alpha_t$, which probably should be $\xi_t$. In eq. (62), where did the $\lambda_1$ go?

---

> > > ### Author Response · Authors · 2026-04-01
> > >
> > > Q1: Due to space constraints, we did not include the full bibliographic entries in the main paper. ERE [1] and DEER [2] are both engineering-oriented improvements designed for non-stationary environments, and their insertion pattern is also relatively close to that of AES.
> > >
> > > [1] Wang, C. and Ross, K., 2019. *Boosting soft actor-critic: Emphasizing recent experience without forgetting the past*. arXiv preprint arXiv:1906.04009.
> > >
> > > [2] Duan, T., Zhang, Z., Guo, S., Zhao, Y., Lin, Z., Fang, Z., Liu, Y., Luan, D., Huang, D., Cui, H. and Cui, Y., 2025. *Sample Efficient Experience Replay in Non-stationary Environments*. arXiv preprint arXiv:2509.15032.
> > >
> > > ---
> > >
> > > Q2: Thank you for this very helpful question. In brief, many of our design choices are motivated by practical deployment considerations, which is also consistent with our goal of formulating RL problems for real-world settings.
> > >
> > > First, $C_2 \lambda_t$ collects all terms in the regret upper bound that increase with the entropy weight. One part comes from the terms introduced when converting the regularized loss back to the original loss, and another part comes from the gradient term in the main inequality. It should therefore be understood as an aggregated stability/regularization cost. As for learning-rate coupling, under the two equivalent views—negative-entropy mirror descent and soft-RL regularization—$\eta_t$ and $\lambda_t$ already describe the same temperature scale from two perspectives, so coupling them is natural.
> > >
> > > Second, the entropy term in Eq. (7) is necessary. Our analysis is explicitly conducted under the entropy-regularized / soft-RL setting. Under this setting, both the comparator and the per-state surrogate objective are themselves entropy-regularized. Removing this term would indeed produce a different bound, but it would also change the problem being analyzed: it would no longer correspond to the entropy-regularized RL targeted by AES.
> > >
> > > Third, the square-root law should be understood more precisely as the dominant-order scaling revealed by the clean per-round trade-off, rather than as a claim of exact closed-form optimality for the full uncompressed original expression. We will clarify this in the revision. As to why the higher-order terms can be absorbed: in the original dynamic-regret bound, the stability side already contains an explicit first-order entropy cost, coming from Eq. (10)–(11). After substitution, the gradient term further introduces additional higher-order corrections. The oracle scaling law we discuss focuses on the low-drift, small-$\lambda$ regime. In this relevant region, the lowest-order positive-power term determines the local trade-off structure, while the higher-order terms only affect the constants. This is because Theorem 3.2 is itself stated over a bounded admissible range, within which the higher-order terms can be tightly upper-bounded.
> > >
> > > ---
> > >
> > > Q3: We agree that the current theory in Section 3 should not be interpreted as a complete characterization of the entire finite-sample deep RL process. Its role is more specific: it first identifies the dominant drift–stability trade-off within the entropy-regularized OCO / mirror-descent geometry at each state, and then lifts it to the MDP level through the chain
> > > $\text{MDP drift} \rightarrow Q_t^\star \text{ drift} \rightarrow \pi_t^\star \text{ drift},$
> > > yielding the principal term at the MDP level, namely the occupancy-weighted sum of per-state regrets. At the same time, the additional uncertainty arising in practical deep RL from function approximation, $Q^\star$ substitution, and occupancy mismatch is not omitted in the paper, but explicitly retained as interface residuals.
> > >
> > > ---
> > >
> > > Q4: Regarding the proxy, our theory does not require TD error specifically. What is required is a conservative observable signal that strengthens as the environment changes. In this paper, we use the TD-error quantile mainly because it is directly available and incurs very little additional cost. Signals such as dynamics-model fit are conceptually compatible with our framework as well.
> > >
> > > ---
> > >
> > > C1: We included PPO with an explicit entropy bonus in the experiments because the focus of this paper is whether entropy-weight scheduling can transfer across different entropy-regularized carriers. We will revise the wording in the main text from MaxEnt RL to the more accurate entropy-regularized RL carriers.
> > >
> > > ---
> > >
> > > C2: Regarding the numbering in the appendix, it is intended to help readers cross-reference the main text at any point. For the notation in Eq. (62), the symbol should indeed be unified as $\xi_t$, and we will correct this. As for $\lambda_1$, it does not disappear. When the intermediate bound is reorganized into the per-round form of Theorem 3.2, all terms that depend only on the initial round t = 1 are absorbed into the constant $C_0$. As a result, the explicit trade-off is written only from $t \geq 2$ onward, since comparator drift itself is defined through differences between adjacent rounds.

---

### Official Review · Reviewer_LPpA · 2026-03-12

**Soundness:** 3
**Presentation:** 3
**Significance:** 3
**Originality:** 3
**Overall Recommendation:** 5
**Confidence:** 2

**Summary:**

This paper proposes AES, an adaptive entropy scheduling method for non-stationary RL. The key idea is that exploration strength should increase when environment drift becomes stronger and decrease when the environment stabilizes. AES is implemented as a lightweight plug-in that replaces a fixed entropy coefficient in SAC, PPO, SQL, and MEow, using a drift proxy computed from TD-style errors.

**Compliance With Llm Reviewing Policy:**

Affirmed.

**Final Justification:**

The explanations were helpful for understanding the paper. I am keeping the score.

**Key Questions For Authors:**

- How sensitive is AES to the choice of drift proxy, especially the quantile level, smoothing window, and clipping range?

- Since TD error can also reflect critic instability or optimization noise, how do you verify that the proxy is measuring true environment drift rather than generic training instability?

- Have you compared AES against other simple adaptive baselines, such as entropy schedules based on return drops, uncertainty estimates, or explicit change-point detectors?

**Limitations:**

yes

**Strengths And Weaknesses:**

Strengths:

- The paper addresses an important problem: fixed entropy regularization is often poorly matched to changing environments. The motivation is clear and relevant.

- The method is simple and practical. AES does not modify the base RL algorithms beyond replacing the entropy weight with a scheduled value, which makes it easy to integrate.

- The paper evaluates AES across multiple carriers and benchmarks, including Toy, MuJoCo, and Isaac Gym, which gives the empirical study decent breadth.

- Results in Table 2 generally show that AES improves normalized AUC and often reduces performance drop under non-stationary settings.

Weaknesses:

- The method depends heavily on the quality of the drift proxy. Since the proxy is based on TD errors or value deltas, it may reflect optimization noise or critic instability in addition to true environment drift.

- Although the results are generally positive, some gains look moderate rather than decisive, so the empirical advantage does not appear uniformly strong in every setting.

- The scheduler itself introduces design choices such as quantile level, smoothing, and clipping range; more robustness analysis in the main paper would make the contribution stronger.

---

> ### Author Rebuttal · Authors · 2026-03-28
>
> We thank the reviewer for the positive assessment of our work and for the constructive suggestions. We respond to the reviewer's questions below.
>
> ---
>
> ## 1. Sensitivity to Drift Proxy Choice (Q1 & W3)
>
> We agree that AES depends on the proxy design, but the sensitivity analysis in Appendix R shows that this dependence is controlled rather than brittle. Theoretically, AES does not require the proxy to be an unbiased estimator of the true drift; it only requires a **conservative signal that increases with environmental change** (Remark 3.4 and Appendix S.2). Thus, noisy, smoothed, and clipped proxies are all compatible with the framework.
>
> Empirically, we find:
> - **Quantiles**: Moderate-to-high TD-error quantiles are the most robust, with **q = 0.8** performing best overall; lower quantiles tend to underreact, while excessively high quantiles introduce more variance
> - **EMA smoothing**: Shows the expected stability–responsiveness trade-off, with **β = 0.95** giving the best balance
> - **Clipping**: Overly wide ranges increase instability, while overly narrow ranges suppress rapid adaptation; the default interval **[1e-4, 1.0]** is the most consistent across metrics
>
> Overall, AES is robust over a reasonably broad parameter region, and the default settings lie within this stable range. We also point to the ablation study in the main text (Line 425). Since hyperparameter tuning is not the main contribution of the paper, we did not move these details into the main text.
>
> ---
>
> ## 2. TD Error as Drift Signal (Q2 & W1)
>
> We agree that raw TD error may reflect not only environmental drift but also critic instability or optimization noise. Our claim is not that TD error is a noise-free estimator of true drift, but that it can serve as a **conservative online signal for driving AES** (Remark 3.4).
>
> To validate this theory-to-practice interface:
> - **Appendix M** upper-bounds comparator drift using a computable MDP-level drift surrogate, and then calibrates the observable proxy on injected-drift benchmarks via a monotone envelope together with conformal adjustment, thereby constructing a conservative proxy that satisfies the upper-bound condition with high probability
> - **Appendix R** further shows that mean TD error and value loss perform worse than the quantile-based TD proxy, suggesting that they are more sensitive to optimization noise, whereas critic-parameter drift yields comparable results
>
> We therefore do not claim to have fully disentangled true environment drift from training instability in general deep RL. However, through conservative calibration and proxy comparison experiments, the paper provides evidence that the chosen proxy is more closely related to non-stationarity rather than merely reacting to generic training noise.
> We discussed related aspects in our response to Reviewer ofE6, which included some extra results.
>
> ---
>
> ## 3. Comparison with Other Approaches (Q3 & W2)
>
> We agree that comparisons to other simple adaptive baselines would strengthen the empirical section. At the same time, we would like to clarify the scope of the current paper. **Our goal is not to present AES as an engineering SOTA method for non-stationary RL**, but to isolate and test a more specific principle: in maximum-entropy RL, exploration intensity should scale with drift magnitude. For this reason, the main paper focuses on standard backbone algorithms and their AES-augmented versions, so that the effect of entropy scheduling itself is directly visible.
>
> Within this scope, we include a simple adaptive baseline in Appendix R: a **lightweight heuristic linear schedule** that uses the same online proxy but replaces the AES accumulation-based square-root rule with a direct linear mapping. This tests whether the gains come from arbitrary proxy-driven adaptivity or from the specific AES scheduling principle.
>
> Empirically, the linear heuristic underperforms AES. We also compare alternative proxy choices to test whether the benefit is robust across simple adaptive variants rather than tied to a single statistic.
>
> We agree that the current version does not yet include separate baselines based on return-drop triggers, uncertainty estimates, or explicit change-point detectors, and we will clarify this limitation. Our reason for not centering the paper around such comparisons is that these methods address a somewhat different question from AES:
>
> **Novelty- or uncertainty-based exploration** changes state/transition preference, whereas AES is intended to control how random the overall policy distribution should be.
> **Explicit change-point detectors** aim to localize discrete shifts, whereas AES performs continuous entropy modulation without requiring explicit detection.
>
> Following this suggestion, we have also reproduced several non-stationary SAC improvements in our setup. We discussed related aspects in our response to Reviewer QkPz, which included some extra results.
>
> ---
>
> Thank you again for the time you have invested.

---

> > ### Author Rebuttal · Reviewer_LPpA · 2026-04-04
> >
> > Thank you for the rebuttal. The explanations were helpful for understanding the paper.

---

> > > ### Author Response · Authors · 2026-04-07
> > >
> > > Dear Reviewer LPpA,
> > >
> > > Thank you very much for your active engagement in the discussion and for kindly reviewing our work. We are glad that our explanations helped clarify the technical parts of our paper, and we truly appreciate your careful consideration.
> > >
> > > Best regards,
> > >
> > > The authors

---

### Official Review · Reviewer_QkPz · 2026-03-12

**Soundness:** 3
**Presentation:** 3
**Significance:** 3
**Originality:** 3
**Overall Recommendation:** 4
**Confidence:** 3

**Summary:**

This paper proposes Adaptive Entropy Scheduling (AES), an easy-to-plug-in method that adapts the entropy coefficient in maximum-entropy RL online using a TD-error drift proxy (actually, it's from the TD error quantile, as shown in Eq. 27, this is quite interesting!).

The theoretical framework casts entropy scheduling as minimizing dynamic regret under non-stationary OCO.



I have no doubts about this paper's originality. Well done!

**Compliance With Llm Reviewing Policy:**

Affirmed.

**Key Questions For Authors:**

pls check my concerns listed above.

**Limitations:**

I found no limitations here.

**Strengths And Weaknesses:**

I enjoyed reading this paper and have learned from it a lot, especially the framing of non-stationary maximum-entropy RL as a dynamic-regret problem. And these equations chains connecting MDP variation ==>>> soft Bellman sensitivity ===>>> softmax Lipschitzness ===>>> OCO comparator drift is elegant and clearly presented. These contributions are novel!

While I like this paper very much, I feel the quality is not ready for publication in ICML. But I highly encourage the authors to iterate a better version, it's a very promising paper, and good paper to read.

Here are my major concerns:
==============================
1. Missing exploration baselines and a thorough related works review.
The paper claims most existing methods fail under non-stationarity due to static entropy coefficients, but never compares against exploration methods that don't use entropy coefficients at all. There are a divers family of works in RL tackling the explroation problem. For example, RND [1] explicitly obtain high intrinsic rewards for novel post-drift states, without any explicit scheduling , and we can search for more of these works. The paper should either include RND-augmented SAC as a baseline, or provide a clear argument for why novelty-based exploration structurally cannot address the identified problem.

2. Still the baselines, the baselines used in the paper are not designed for non-stationary RL.
SAC, PPO, and SQL are stationary algorithms. MEow (Chao et al., NeurIPS 2024) is simply another maximum-entropy RL algorithm with via energy-based normalizing flows. It was not designed for non-stationary environments.

The paper actually cites sliding-window and restart-based methods [2] and continual RL methods [3] in related work but compares against none of them. You can see, as a reviewer, this looks strange for me.

I like this paper very much, but the baselines selection makes me to withhold, and suggest the authors revise the paper to a better version.

Actually, my suggestion is that, if the authors feel it's too much work to do in the experiment part, the minimum, a sliding-window SAC baseline (check your introduction section, line 70~72, you actually know it can be a strategy, jsut that the window size needs to be manually chosen) should be included, as this is the standard non-stationary RL baseline and directly tests whether AES provides actual superior performance.

Here are my Minor Concerns (do not affect my rating):
==========================================
3. Eval env setup seems to be underspecified and appears hand-picked.
The Toy environment uses reward drift (goal location), while MuJoCo and Isaac Gym use dynamics drift (mass, friction, gravity). These are structurally different. I understand the continual RL reserach currently lacks well-known benchmarks, but if you want to do a fair evaluation, you need systematical thinking, why these envs have mixed setup? why not each one both test dynamic drifts, and reward drifts?

Besides that, the choice of drift magnitudes and environments also lacks justification — were these selected because they produced the most visible performance drops?

In the future, if you want to propose a continual RL benchmark that everyone can use, you also need to answer the above questions.

4. The nAUC metric really does not make sense to me!
To me, the best metric is still regret, as the authors actually stated clearly in the paper's definition part.  Continual RL agent will need to minimize the dynamic regret. But no experiment reports anything close to this.

The nAUC has a drawback, it conflates steady-state performance with post-drift recovery. As you stated in your (Appendix S.5), since drift segments occupy only 20% of training , nAUC is actually dominated by steady-state performance.

A simple example: an agent that performs well before drift but recovers slowly will outscore a fast-recovering agent with slightly lower steady-state return — the opposite of what the paper aims to reward.
So, I think dynamic regret against an oracle (which you can have a way to get since you are working in simulator) is computable here, and should be reported.



Lastly, I did not fully read through all the equations in the appendix! That's a lot of them, some are difficult for me to understand. While I love these equations, my understand of the paper may be wrong. If that's the case, pls let me know, I would like to learn from you.


References
[1] Burda, Y., Edwards, H., Storkey, A., & Klimov, O. (2018). Exploration by random network distillation. arXiv preprint arXiv:1810.12894.
======>>> this one is a new paper I introduced to the authors.

[2] Besbes, O., Gur, Y., and Zeevi, A. Stochastic multi-armed-bandit problem with non-stationary rewards. NeurIPS, 2014.
===>>>> this one has been cited by the authors in the original paper

[3] Khetarpal, K., Riemer, M., Rish, I., and Precup, D. Towards continual reinforcement learning: A review and perspectives. JAIR, 75:1401–1476, 2022.
=====>>> this survey paper has listed lots of continual RL methods, but be careful if you want to select baselines from them: some of them do not address the dynamic reward dirft, or env dynamics drift problem!

---

> ### Author Rebuttal · Authors · 2026-03-28
>
> We thank the reviewer for the careful reading and positive comments, and for the effort you have made together with us to improve this paper. Here we mainly clarify several deliberate design choices, and we also report some additional experiments that we believe are informative.
>
> ---
>
> ## 1. Paper Positioning and Baseline Choice (W1)
>
> We begin with the main concern paper positioning and baseline choice.
> We emphasize a deliberate choice. By including only standard baseline algorithms, we aim to present readers with a clear theoretical picture: how exploration intensity should scale with the magnitude of drift.
> What we hope to highlight is the importance of treating entropy scheduling as a key element in maximum-entropy reinforcement learning, together with the fact that this approach is plug-and-play, and can be used across a range of settings and algorithms.
>
> For novelty-based exploration methods, what they adjust is the preference over states or transitions, whereas AES in this paper is meant to address how the policy distribution as a whole should be. More concretely, novelty and comparator drift are **not equivalent** in a general non-stationary MDP. Reward drift or dynamics drift can significantly change the optimal policy without necessarily producing highly novel states; conversely, novel states do not necessarily imply a drift of comparable magnitude in the optimal policy. For this reason, novelty bonuses may help in some post-drift settings, but they do not answer, at a structural level, the question we study here, namely that drift magnitude should determine exploration intensity. We will make this point clearer in the related work.
>
> ---
>
> ## 2. Comparison with Non-Stationary Methods (W2)
>
> At the same time, we agree that comparisons with non-stationary methods are useful. Under our setup, we reproduced several non-stationary variants of SAC. ERE and DEER give **no gain** in stationary settings and about **20%** in non-stationary settings, while AES gives about **10% and 35%**, respectively, across different change settings. We will include the full results in the final version.
>
> **On presentation**: We would welcome discussion on whether these experiments should go into the main text or the appendix. Our current version emphasizes **methodological insight and theoretical design**; adding many comparisons might shift reader attention toward viewing this as "just another method paper pursuing SOTA performance"—which is not what the author team intends to present.
> We would welcome a deeper discussion with you on this point.
>
> ---
>
> ## 3. Evaluation Environment Design (W3)
>
> We very much appreciate the reviewer's insight into RL benchmarks. Our setup is intended to cover both dynamics drift and reward drift, so as to test whether AES shows a consistent adaptation trend, rather than to build a full-factorial benchmark.
>
> **Why treated separately**:
> - Introducing reward drift in MuJoCo and Isaac Gym raises significant engineering challenges
> - Fairness is hard to guarantee
> - We chose not to spend excessive effort on this aspect
>
> **Drift magnitude selection**: Chosen to create unified and comparable non-stationary test conditions; no other deliberate selection criteria applied.
>
> ---
>
> ## 4. Evaluation Metrics (W4)
>
> We also agree that nAUC is not a purely post-drift metric. We keep it because the paper is not about trading steady-state performance for faster adaptation, but about **improving non-stationary performance without hurting steady-state behavior**, and possibly improving both. To better reflect drift-induced damage and adaptation ability, we also report drop-area ratio and abrupt-change recovery time, and we will make their roles clearer in the revision.
>
> We agree that dynamic regret is conceptually closer to our theory, but in deep continuous-control settings it is difficult to compute cleanly.
> In theory, the comparator at each timestep is the soft-optimal policy under the current MDP.
> However, in continuous state-action spaces such as MuJoCo and Isaac Gym, constructing high-precision oracles for a sequence of ever-changing MDPs becomes challenging. If the oracle is approximated through additional training, the reported regret would be contaminated by oracle optimization error, **no longer constituting a clean theoretical quantity**. This indeed presents a fundamental obstacle between elegant theoretical work and the most straightforward experimental demonstration, which motivated us to pursue alternative approaches.
>
> ---
>
> ## 5. Appendix Purpose
>
> The appendix is intended for readers who want to examine derivations in more detail after reading the main text. If the main text already conveys the proof idea and core reasoning, it has served its purpose.
>
> ---
>
> Additionally, we have read the references recommended by the reviewers, which provided valuable insights, and we will incorporate these materials. We hope these clarifications make our choices clearer, and we look forward to further discussion.

---

### Decision · Program_Chairs · 2026-04-30

**Decision:**

Accept (regular)

**Comment:**

A technically strong paper with consensus among the reviewers.